# Riemannian Neural Optimal Transport

**Alessandro Micheli** [1]   **Yueqi Cao** [2]   **Anthea Monod** [1]   **Samir Bhatt** [1 3]

## Abstract

Computational optimal transport (OT) offers a principled framework for generative modeling. Neural OT methods, which use neural networks to learn an OT map (or potential) from data in an amortized way, can be evaluated out of sample after training, but existing approaches are tailored to Euclidean geometry. Extending neural OT to high-dimensional Riemannian manifolds remains an open challenge. In this paper, we prove that any method for OT on manifolds that produces discrete approximations of transport maps necessarily suffers from the curse of dimensionality: achieving a fixed accuracy requires a number of parameters that grows exponentially with the manifold dimension. Motivated by this limitation, we introduce Riemannian Neural OT (RNOT) maps, which are continuous neural-network parameterizations of OT maps on manifolds that avoid discretization and incorporate geometric structure by construction. Under mild regularity assumptions, we prove that RNOT maps approximate Riemannian OT maps with sub-exponential complexity in the dimension. Experiments on synthetic and real datasets demonstrate improved scalability and competitive performance relative to discretization-based baselines.

## 1. Introduction

Optimal Transport (OT) casts generative modeling as a transport problem: one seeks a map (or coupling) that pushes a simple reference distribution (e.g., a Gaussian) onto the data distribution while minimizing an expected cost (Kantorovitch, 1958; Villani, 2016). In Euclidean space with quadratic cost, the optimal Monge map admits a particularly tractable representation: under mild conditions it is the gradient of a convex potential (Brenier, 1991). Neural OT methods exploit this structure by parameterizing the map or its potential with neural networks and training them by optimizing an OT objective (Makkuva et al., 2020; Korotin et al., 2023; Geuter et al., 2025). Once trained, sampling is amortized: samples are drawn from the reference distribution and pushed forward through the learned transport map (or the map induced by the learned potentials) to generate a data sample.

Extending computational OT beyond Euclidean spaces remains challenging. Many datasets are naturally supported on Riemannian manifolds, such as spheres and tori, especially when observations represent angles, axes, or directions (Mardia & Jupp, 1999), with applications in protein modelling (Hamelryck et al., 2006; Mardia et al., 2006; Boomsma et al., 2008), geology (Peel et al., 2001), and robotics (Feiten et al., 2013; Senanayake & Ramos, 2018).In such settings, intrinsic geometry is not merely a modeling preference: Euclidean embeddings can distort geodesic distances, periodic structure, and rotational symmetries, leading to transport costs and maps that are misaligned with the data geometry. There have been promising attempts to bring OT-based generative models to manifolds; for instance, Riemannian Convex Potential Maps (RCPMs) build manifold transport layers from $c$-concave potentials represented as minima over finitely many squared-distance templates anchored at a finite set of sites (Cohen et al., 2021). Methods of this form fall into a broad discretization-based paradigm: the learned transport is mediated by a finite discrete representation, yielding a map with inherently discrete complexity. In this work, we show that this paradigm faces a *fundamental barrier*: any method on compact Riemannian manifolds that constructs a discrete approximation of the OT map necessarily suffers from the curse of dimensionality (CoD), requiring exponentially many parameters in the manifold dimension to achieve fixed approximation accuracy.

This negative result does not rule out the possibility of a genuinely continuous neural OT framework on manifolds that escapes the CoD. On the contrary, recent approximation theory in geometric deep learning shows that neural architectures can avoid dimension-driven blow-ups when learning structured functions between manifolds under suitable conditions (Kratsios & Papon, 2022). Yet, despite the existence of theoretically grounded neural OT frameworks

[1]Imperial College London, London, UK [2]KTH Royal Institute of Technology, Stockholm, Sweden [3]University Of Copenhagen, Copenhagen, Denmark. Correspondence to: Alessandro Micheli <a.micheli19@imperial.ac.uk>.

*Proceedings of the $43^{rd}$ International Conference on Machine Learning*, Seoul, South Korea. PMLR 306, 2026. Copyright 2026 by the author(s).

in Euclidean settings (Makkuva et al., 2020; Korotin et al., 2023; Geuter et al., 2025), there is, to the best of our knowledge, no comparably principled neural OT framework that learns transport potentials and maps *intrinsically* on general Riemannian manifolds, let alone one with guarantees that escape the CoD.

**Our main contribution** is to bridge this gap by introducing *Riemannian Neural OT* (RNOT), a theoretically grounded neural OT framework that learns amortized transport maps directly on Riemannian manifolds with *provable polynomial complexity guarantees*. Our theory focuses on compact manifolds without boundary, a setting that avoids boundary effects and allows the $c$-transform, exp/log maps, Monge solutions, and regularity theory to be developed cleanly. RNOT avoids discretizing the manifold. Instead, it represents transport potentials as continuous functions on $\mathcal{M}$ and enforces the structural OT constraint of $c$-concavity for the quadratic cost by construction via the $c$-transform. This yields a practical family of manifold-valued transport maps through the exponential formula $T(x) = \exp_x(-\nabla\phi(x))$, enabling out-of-sample generation by pushing forward samples from a reference distribution. We establish a universality result showing that approximating potentials within our implicit $c$-concave class induces transport maps that converge to the true Riemannian OT map. For neural instantiations, we derive explicit polynomial bounds on the neural network parameter count and depth required to approximate the optimal transport map to prescribed pointwise accuracy. Finally, experiments on synthetic and real manifold-valued datasets demonstrate improved scalability and competitive performance compared with discretization-based baselines.

The code to reproduce our experiments is available on the GitHub repository https://github.com/MLGlobalHealth/riemannian-neural-optimal-transport under the MIT License.

## 2. Background

Throughout, $(\mathcal{M}, g)$ denotes a connected, compact, smooth $p$-dimensional Riemannian manifold without boundary, with geodesic distance $d(\cdot, \cdot)$. We assume that the intrinsic geometry of $\mathcal{M}$ is available, in the sense that geodesic distances and the required local Riemannian operations, such as exponential/logarithm maps and gradients, can be evaluated, which is a common setting in manifold generative models. We let $\mathcal{P}(\mathcal{M})$ be the set of Borel probability measures on $\mathcal{M}$ and write $\mathrm{vol}_{\mathcal{M}}$ for the Riemannian volume measure. A brief review of the required notions from Riemannian geometry and OT, together with the relevant notation, is provided in Appendix A and B.

### 2.1. Background on Optimal Transport

A central object in OT is the *c-transform*, which generalizes the Legendre–Fenchel transform to Riemannian manifolds. Let $c : \mathcal{M} \times \mathcal{M} \to \mathbb{R}$ be a cost function.

**Definition 2.1** (Def. 3.1 from Cordero-Erausquin et al. (2001))**.** The set $\Psi_c(\mathcal{M})$ of $c$-concave functions is the set of functions $\phi : \mathcal{M} \to \mathbb{R} \cup \{-\infty\}$ not identically $-\infty$, for which there exists a function $\psi : \mathcal{M} \to \mathbb{R} \cup \{-\infty\}$ such that

$$\phi(x) = \inf_{y \in \mathcal{M}} (c(x,y) - \psi(y)) \quad \forall x \in \mathcal{M}. \quad (1)$$

We refer to $\phi$ as the $c$-transform of $\psi$ and abbreviate (1) by writing $\phi = \psi^c$. Similarly, given $\phi \in \Psi_c(\mathcal{M})$, we define its $c$-transform $\phi^c \in \Psi_c(\mathcal{M})$ by

$$\phi^c(y) := \inf_{x \in \mathcal{M}} (c(x,y) - \phi(x)), \quad \forall y \in \mathcal{M}. \quad (2)$$

For $\phi \in \Psi_c(\mathcal{M})$, from (2) it is straightforward to show that $\phi^{cc} = \phi$ (3). As in McCann (2001), compactness of $\mathcal{M}$ and local Lipschitz regularity of $c(x,y)$ imply that $\phi^c$ is Lipschitz, regardless of whether $\phi : \mathcal{M} \to \mathbb{R} \cup \{-\infty\}$ is continuous. Consequently, using (3), we may assume without loss of generality that the functions $\psi$ and $\phi$ in (1) lie in $C(\mathcal{M}, \mathbb{R})$.

OT studies efficient ways to move mass from a source measure $\mu \in \mathcal{P}(\mathcal{M})$ to a target measure $\nu \in \mathcal{P}(\mathcal{M})$. In the Monge Problem (MP) formulation, the problem is to find a measurable map $T : \mathcal{M} \to \mathcal{M}$ such that $T_{\#}\mu = \nu$ and minimizes the transportation cost

$$\inf_{T_{\#}\mu = \nu} \int_{\mathcal{M}} c(x, T(x)) \, \mathrm{d}\mu(x). \quad \text{(MP)}$$

In this work, we focus on the squared-distance cost

$$c(x, y) := \tfrac{1}{2} d(x, y)^2.$$

When $\mu$ is absolutely continuous with respect to $\mathrm{vol}_{\mathcal{M}}$, McCann (McCann, 2001, Thm. 9) showed that there exists an OT map $T : \mathcal{M} \to \mathcal{M}$ that is $\mu$-a.e. unique, pushes $\mu$ forward to $\nu$ (i.e., $T_{\#}\mu = \nu$), and minimizes (MP). Moreover, the optimal map is induced by a $c$-concave potential: there exists a $c$-concave function $\phi$ associated with the optimal transport problem such that

$$T(x) = \exp_x(-\nabla\phi(x)),$$

where $\exp_x : T_x\mathcal{M} \to \mathcal{M}$ is the Riemannian exponential map and $\nabla$ denotes the Riemannian gradient.

### 2.2. Background on Geometric Deep Learning

A recurring strategy in geometric deep learning is to reduce learning on a manifold to learning in a Euclidean space via

a suitable *feature map* (or embedding). Since our goal is to establish universal approximation results for OT potentials and maps on a Riemannian manifold, we recall the notion of uniform convergence for standard Euclidean approximation theorems and overview how it adapts to compact domains.

We work with the topology of *uniform convergence on compact sets* (ucc) for spaces of continuous functions. For a topological space $X$, a sequence $(f_k)_{k \in \mathbb{N}} \subset C(X, \mathbb{R})$ converges to $f \in C(X, \mathbb{R})$ in the ucc topology if and only if

$$\forall K \subset X \text{ compact}, \quad \sup_{x \in K} |f_k(x) - f(x)| \xrightarrow[k \to \infty]{} 0. \quad (4)$$

When $X = \mathbb{R}^n$, this is the standard mode of convergence used for approximation on non-compact domains; equivalently, it is the topology induced by the usual ucc metric (see Kratsios & Bilokopytov (2020, Eq. (2), Sec. 2.2)). When $X$ is compact, (4) reduces to ordinary uniform convergence, since one may take $K = X$. In particular, because $\mathcal{M}$ is compact in our setting, the ucc topology on $C(\mathcal{M}, \mathbb{R})$ coincides with the topology induced by the uniform norm

$$\|g\|_\infty := \sup_{x \in \mathcal{M}} |g(x)|. \quad (5)$$

Following Kratsios & Bilokopytov (2020), we introduce a continuous feature map $\varphi : \mathcal{M} \to \mathbb{R}^n$ and consider function classes on $\mathcal{M}$ induced by composition. Throughout this paper we denote a dense subset of $C(\mathbb{R}^n, \mathbb{R})$ under ucc by $\mathcal{F}$, such as the neural network architectures studied by Leshno et al. (1993); Lu et al. (2017); Zhou (2020) or the posterior means of a Gaussian process with universal kernel as in Micchelli et al. (2006). We then define the following $\varphi$-*pullback* (or *feature-induced*) class:

$$\varphi^* \mathcal{F} := \{ f \circ \varphi \ : \ f \in \mathcal{F} \}.$$

The approximation power of the composed class $\varphi^* \mathcal{F}$ depends not only on $\mathcal{F}$, but crucially also on the geometry of the feature map $\varphi$. In particular, $\varphi$ must preserve enough information on points on $\mathcal{M}$ so that distinct points remain distinguishable after embedding.

**Assumption 2.2** (Feature Map Regularity)**.** The feature map $\varphi : \mathcal{M} \to \mathbb{R}^n$ is continuous and injective.

Assumption 2.2 is precisely the condition that allows for transferability of Euclidean approximation to the manifold. Informally, Kratsios & Bilokopytov (2020) show:

- If $\mathcal{F}$ is universal on $\mathbb{R}^n$ (dense in $C(\mathbb{R}^n, \mathbb{R})$ under ucc) and $\varphi$ is continuous and injective, then $\varphi^* \mathcal{F}$ is universal on $\mathcal{M}$ (dense in $C(\mathcal{M}, \mathbb{R})$ under $\| \cdot \|_\infty$); see Kratsios & Bilokopytov (2020, Theorem 3.3).

- Conversely, injectivity is effectively *necessary*: if $\varphi$ is not injective, then there exist continuous functions

on $\mathcal{M}$ that cannot be represented (or uniformly approximated) by compositions $f \circ \varphi$; see Kratsios & Bilokopytov (2020, Theorem 3.4).

For our purposes, a particularly relevant class of feature maps is obtained by encoding points through their distances to a set of landmarks. In the compact Riemannian setting, a classical construction due to Gromov (1983) shows that such maps can be chosen to be injective. Distance-to-landmark features provide an intrinsic alternative to ambient-coordinate MLPs: they are defined through the geodesic metric rather than a particular embedding, making the resulting potential parameterization better aligned with manifold OT geometry.

**Proposition 2.3** (Distance-to-Landmarks Embedding (Gromov, 1983))**.** *For sufficiently small $\delta > 0$, let $\{x_i\}_{i \in I}$ be a maximal $\delta$-separated subset of $\mathcal{M}$ (which is finite by compactness), and define*

$$\varphi(x) := (d(x, x_i))_{i \in I} \in \mathbb{R}^I.$$

*Then $\varphi$ is continuous and injective (hence satisfies Assumption 2.2).*

### 2.3. Background on (Un)Cursed Approximation Rates for Deep ReLU Networks

A convenient way to formalize the CoD in approximation theory is to ask how the model size—measured here by the number of parameters $W$—must grow in order to guarantee a prescribed uniform accuracy $\varepsilon$ on an $n$-dimensional domain. As a canonical benchmark, consider the Hölder unit ball $\mathcal{H}_{r,n}$ of $r$-smooth functions on the cube $[0,1]^n$ (see (16)). In this setting, the aim is to obtain estimates of the form

$$\sup_{f \in \mathcal{H}_{r,n}} \|f - \widehat{f}_W\|_\infty \ \lesssim \ W^{-\beta},$$

where $\widehat{f}_W$ denotes an approximation produced by a model with $W$ parameters, and $\beta > 0$ is the approximation exponent, which we refer to as a *rate*.

A sharp view of attainable rates is given by the phase diagram of Yarotsky & Zhevnerchuk (2020) for uniform approximation of the Hölder ball $\mathcal{H}_{r,N}$ by deep ReLU networks. In their framework, a rate $\beta$ is *achievable* if there exist network architectures with $W$ weights and weight-assignment maps such that the worst-case error over $\mathcal{H}_{r,n}$ decays as $\mathcal{O}(W^{-\beta})$ as $W \to \infty$. For Hölder classes, the classical reference rate is $r/n$: in the *continuous* regime, where the weights must depend continuously on the target function, Yarotsky & Zhevnerchuk (2020, Theorems 3.1 and 3.2) show that $r/n$ is the sharp threshold—achievable, and unimprovable under the continuity constraint. Allowing *discontinuous* weight assignment and sufficiently deep networks unlocks a faster *deep-discontinuous* phase: for

any $r > 0$ and any exponent $\frac{r}{n} < \beta < \frac{2r}{n}$, there exist deep ReLU constructions achieving rate $\beta$ (Yarotsky & Zhevnerchuk, 2020, Theorem 3.3 and Fig. 3).

Kratsios & Papon (2022) exploit the deep–discontinuous regime to obtain approximation guarantees on manifolds that avoid exponential dependence on the dimension. They do not aim for uniform approximation over $\mathcal{M}$; instead, they approximate the target only on a fixed dataset $\mathcal{D} \subset \mathcal{M}$ rather than uniformly over $\mathcal{M}$. In particular, for $f \in C^{kp,1}(\mathcal{M})$ with $k \in \mathbb{N}$ (equivalently, Hölder smoothness $r = kp + 1$), they show that $f$ can be approximated to accuracy $\varepsilon$ uniformly on $\mathcal{D}$ by a deep neural network with sub-exponential (in fact, polynomial) complexity in $\varepsilon^{-1}$ (Kratsios & Papon, 2022, Section 3.3).

# 3. Why Discrete Optimal Transport Suffers in High Dimensions

In this section, we establish a general CoD barrier for OT on Riemannian manifolds when the learned transport map has *finite support*, meaning it can send samples to only finitely many target locations. Our results apply broadly to any method whose discretization is introduced either explicitly (e.g., via meshes) or implicitly through the model parameterization, and they hold irrespective of the particular algorithm or optimization procedure. As a concrete instance, we show that RCPMs, which, to the best of our knowledge, are the only existing approach that directly constructs Riemannian OT maps, produce discrete-output maps at any finite model size and therefore inherit the same exponential dependence on dimension.

**Discrete-Output Maps and Approximation Error.** We fix $\mu, \nu \in \mathcal{P}(\mathcal{M})$ and let $T_\star$ denote the ($\mu$-a.e. defined) OT map pushing $\mu$ to $\nu$ for the quadratic cost $c(x, y) = \frac{1}{2}d(x, y)^2$. To compare measurable maps $T, S : \mathcal{M} \to \mathcal{M}$ under a source measure $\mu$, we use the $\mu$-root-mean-square error

$$\text{RMSE}_\mu(T, S) := \left( \int_{\mathcal{M}} d\big(T(x), S(x)\big)^2 \, \mathrm{d}\mu(x) \right)^{1/2}.$$

A common structural restriction in *discrete* OT parameterizations is that the learned map can output only finitely many points; for instance, it may induce a partition of $\mathcal{M}$ into cells that are mapped to a finite set of sites. We capture this discretization effect via the class of *discrete-output maps*. For $m \in \mathbb{N}$, define $\mathsf{D}_m(\mu)$ to be the set of measurable maps $T : \mathcal{M} \to \mathcal{M}$ such that the pushforward $T_\# \mu$ is supported on at most $m$ points, i.e., $\#\text{supp}(T_\# \mu) \leq m$. Equivalently, $T \in \mathsf{D}_m(\mu)$ if and only if there exist $y_1, \ldots, y_m \in \mathcal{M}$ such that $T(x) \in \{y_1, \ldots, y_m\}$ for $\mu$-a.e. $x$.

**Discrete-Output Maps Induce a CoD Barrier.** The next theorem isolates the core bottleneck of discrete-output parameterizations: any approximation of $T_\star$ within $\mathsf{D}_m(\mu)$ necessarily induces an $m$-atomic approximation of $\nu$. When $\nu \ll \text{vol}_\mathcal{M}$ (as in Theorem 3.1), this quantization step yields an unavoidable lower bound with exponential dependence on the intrinsic manifold dimension $p$.

**Theorem 3.1** (CoD Barrier for Discrete-Output Maps). *Assume $\mu, \nu \ll \text{vol}_\mathcal{M}$ and let $T_\star$ be the optimal transport map from $\mu$ to $\nu$ for $c(x, y) = \frac{1}{2}d(x, y)^2$. Then there exists $C > 0$ (depending on $\mathcal{M}$ and $\nu$) such that for all $m \in \mathbb{N}$,*

$$\inf_{T \in \mathsf{D}_m(\mu)} \text{RMSE}_\mu(T, T_\star) \geq C \, m^{-1/p}. \tag{6}$$

*In particular, achieving $\text{RMSE}_\mu(T, T_\star) \leq \delta$ requires $m \geq (C/\delta)^p$.*

The proof of Theorem 3.1 is deferred to Appendix H.1. Theorem 3.1 shows that this CoD barrier is inherent to discretization-based Riemannian OT methods, independent of architecture and optimization.

We now focus our discussion on RCPMs. For $m \in \mathbb{N}$, RCPMs consider the class $\mathcal{C}_m(\mathcal{M})$ of *discrete $c$-concave* potentials of the form

$$\phi_m(x) = \min_{i \in [m]} \big(c(x, y_i) + \alpha_i\big), \quad y_i \in \mathcal{M}, \ \alpha_i \in \mathbb{R}.$$

A central theorem in Cohen et al. (2021) shows that on any compact, smooth, boundaryless manifold, the family $\{\mathcal{C}_m(\mathcal{M})\}_{m \in \mathbb{N}}$ is dense (in an appropriate sense) on the set of $c$-concave potentials. In particular, the associated transport maps can approximate OT maps arbitrarily well as $m \to \infty$.

Despite this asymptotic expressivity, RCPMs are nevertheless subject to a CoD at any *finite* $m$. Indeed, for $\phi \in \mathcal{C}_m(\mathcal{M})$ the associated map

$$T_\phi(x) := \exp_x\big(-\nabla\phi(x)\big),$$

takes values in the finite site set $\{y_1, \ldots, y_m\}$. Consequently, $(T_\phi)_\# \mu$ is an $m$-atomic measure, and Theorem 3.1 yields a dimension-dependent lower bound on the approximation error.

**Corollary 3.2** (CoD Barrier for RCPMs). *Under the assumptions of Theorem 3.1, for every $m \in \mathbb{N}$ and every $\phi \in \mathcal{C}_m(\mathcal{M})$, the associated map $T_\phi$ satisfies $T_\phi \in \mathsf{D}_m(\mu)$. Consequently, there exists $C > 0$ such that for all $m \in \mathbb{N}$,*

$$\inf_{\phi \in \mathcal{C}_m(\mathcal{M})} \text{RMSE}_\mu(T_\phi, T_\star) \geq C \, m^{-1/p}, \tag{7}$$

*and achieving $\text{RMSE}_\mu(T_\phi, T_\star) \leq \delta$ requires $m \geq (C/\delta)^p$.*

The proof of Corollary 3.2 is deferred to Appendix H.2.

# 4. Riemannian Neural Optimal Transport

We have shown that discretization is a fundamental obstacle to scalable OT on Riemannian manifolds. We therefore pursue a complementary approach based on continuous parameterizations of OT maps that avoid discrete representations altogether. Concretely, we introduce *Riemannian Neural Optimal Transport (RNOT) maps*, which parameterize a transport potential with a neural network and recover the associated transport map via the Riemannian exponential map. This yields a manifold-valued generator whose pushforward is not restricted to finitely many locations, and which preserves the geometric structure of OT by construction.

**Implicit Continuous Representations.** To define RNOT, we start with a general framework for constructing continuous parameterizations of OT maps on Riemannian manifolds. The key idea is to approximate transport potentials within a class that is expressive, while enforcing the structural requirement of $c$-concavity by construction through the $c$-transform.

Recall from Section 2.2 that $\mathcal{F}$ is dense in $C(\mathbb{R}^n, \mathbb{R})$ under the ucc topology. Given a feature map $\varphi : \mathcal{M} \to \mathbb{R}^n$, this induces the pullback class

$$\varphi^* \mathcal{F} = \{ f \circ \varphi : f \in \mathcal{F} \} \subset C(\mathcal{M}, \mathbb{R}).$$

Since OT potentials are $c$-concave, we impose this property by passing to the $c$-transform. For any $\mathcal{G} \subset C(\mathcal{M}, \mathbb{R})$, define the associated implicit $c$-concave class

$$\mathfrak{C}(\mathcal{G}) := \{ \psi^c : \psi \in \mathcal{G} \}.$$

In particular, we approximate OT potentials in $\mathfrak{C}(\varphi^* \mathcal{F})$, so that $c$-concavity is built in rather than enforced via an external constraint. The following theorem shows that approximating the potential within this implicit $c$-concave class induces transport maps that converge to the true Riemannian OT map.

**Theorem 4.1** (Universality of Implicit $c$-Concave Potentials.). *Assume $\mu, \nu \in \mathcal{P}(\mathcal{M})$ with $\mu \ll \mathrm{vol}_{\mathcal{M}}$ and let $T_\star$ be the OT map from $\mu$ to $\nu$ for $c(x, y) = \frac{1}{2} d(x, y)^2$. Assume that $\mathcal{F}$ is a dense subset of $C(\mathbb{R}^n, \mathbb{R})$ under the ucc topology and that the feature map $\varphi : \mathcal{M} \to \mathbb{R}^n$ satisfies Assumption 2.2. Then there exists a sequence of potentials $\{\phi_k\}_{k \geq 1} \subset \mathfrak{C}(\varphi^* \mathcal{F})$ with associated maps*

$$T_k(x) := \exp_x(-\nabla \phi_k(x))$$

*such that*

$$T_k(x) \longrightarrow T_\star(x) \qquad \text{for } \mu\text{-almost every } x \in \mathcal{M}.$$

*In particular, $T_k \to T_\star$ in probability under $\mu$.*

The proof of Theorem 4.1 is deferred to Appendix H.3.

**RNOT Potentials and Maps.** We now instantiate the framework with neural networks, yielding a trainable class of Riemannian OT potentials and maps. Fix an activation $\sigma : \mathbb{R} \to \mathbb{R}$ and let $\mathcal{NN}_n^\sigma \subset C(\mathbb{R}^n, \mathbb{R})$ denote a feedforward neural network realization class that is dense in $C(\mathbb{R}^n, \mathbb{R})$ under the ucc topology. For standard choices of $\sigma$ (e.g., non-polynomial activations), ucc-density follows from universal approximation theorems; see Leshno et al. (1993); Lu et al. (2017); Zhou (2020). This leads to the following definition.

**Definition 4.2** (Riemannian Neural OT Potentials and Maps). Let $\varphi : \mathcal{M} \to \mathbb{R}^n$ be a feature map satisfying Assumption 2.2 and let $\mathcal{NN}_n^\sigma$ be as above. Define the class of *RNOT potentials* by

$$\mathfrak{C}(\varphi^* \mathcal{NN}_n^\sigma) = \{ (f \circ \varphi)^c : f \in \mathcal{NN}_n^\sigma \}.$$

The associated *RNOT maps* are obtained by the exponential-formula

$$T(x) = \exp_x(-\nabla \phi(x)), \qquad \phi \in \mathfrak{C}(\varphi^* \mathcal{NN}_n^\sigma).$$

**Training via the Kantorovich Semi-Dual.** Given source and target measures $\mu, \nu \in \mathcal{P}(\mathcal{M})$, we train RNOT potentials using the Kantorovich semi-dual formulation. For the quadratic cost $c(x, y) = \frac{1}{2} d(x, y)^2$, Kantorovich duality and the $c$-transform give

$$\mathrm{KP}(\mu, \nu) = \sup_{\psi \in C(\mathcal{M})} \left\{ \mathbb{E}_{x \sim \mu}[\psi^c(x)] + \mathbb{E}_{y \sim \nu}[\psi(y)] \right\},$$

$$\psi^c(x) = \inf_{y \in \mathcal{M}} (c(x, y) - \psi(y)).$$

RNOT parameterizes the prepotential as

$$\psi_\theta = f_\theta \circ \varphi,$$

where $f_\theta$ is a neural network acting on the intrinsic landmark-distance features. We therefore optimize the negative semi-dual loss

$$\mathcal{L}(\theta) = -\mathbb{E}_{x \sim \mu}[\psi_\theta^c(x)] - \mathbb{E}_{y \sim \nu}[\psi_\theta(y)],$$

estimated from mini-batches of samples from $\mu$ and $\nu$. Evaluating $\psi_\theta^c(x)$ requires the inner minimization

$$\psi_\theta^c(x) = \min_{y \in \mathcal{M}} \left\{ \tfrac{1}{2} d(x, y)^2 - \psi_\theta(y) \right\},$$

which is solved by Riemannian gradient descent in our implementation. This objective trains the $c$-concave potential directly from samples, without discretizing the transport map or evaluating Jacobian determinants; full optimization details are deferred to Appendix E.2.

This specialization yields a practical neural hypothesis class of RNOT potentials and maps which can be implemented via implicit layers and trained end-to-end from samples using

the Kantorovich semi-dual objective, thereby avoiding the expensive Jacobian-determinant evaluations that arise in likelihood/KL-based training of normalizing-flow models (e.g., Cohen et al. (2021); Rezende et al. (2020); Mathieu & Nickel (2020)). Training and implementation details are deferred to Appendix E.

# 5. Riemannian Neural Optimal Transport Breaks the CoD

Having introduced RNOT in Section 4, we now prove that *RNOT breaks the CoD*: for sufficiently regular problems, both the Kantorovich potential and the resulting transport map can be approximated to accuracy $\varepsilon$ by RNOT models whose parameter count and depth scale polynomially in $\varepsilon^{-1}$ (and do not grow exponentially with the manifold dimension). Throughout, we work with non-affine piecewise-linear activations and the architecture class $\mathcal{NN}_{n,W,L}^{\sigma}$, with input dimension $n$, number of trainable parameters $W$, and $L$ layers (Definitions H.7–H.8). We split the argument into three steps.

**Step 1: Function Approximation on $\mathcal{M}$ with Polynomial $\varepsilon^{-1}$-Complexity.** Our first result establishes uniform approximation of sufficiently smooth functions on $\mathcal{M}$ with explicit bounds on the parameter count $W_\varepsilon$ and depth $L_\varepsilon$.

**Theorem 5.1** (Polynomial $\varepsilon$-Complexity for Functions on Manifolds)**.** *Let $\sigma : \mathbb{R} \to \mathbb{R}$ be any non-affine piecewise-linear activation function. Fix $k \in \mathbb{N}$ and let $\psi \in C^{kp,1}(\mathcal{M}, \mathbb{R})$. Then, for every $\varepsilon \in (0,1)$, there exists a feature map $\varphi_\star : \mathcal{M} \to \mathbb{R}^{2p}$ satisfying Assumption 2.2, integers $W_\varepsilon, L_\varepsilon \in \mathbb{N}$, and a neural network $\hat{g}_\varepsilon \in \mathcal{NN}_{2p,W_\varepsilon,L_\varepsilon}^{\sigma}$, such that the pullback network*

$$\hat{\psi}_\varepsilon := \hat{g}_\varepsilon \circ \varphi_\star \in \varphi_\star^* \mathcal{NN}_{2p,W_\varepsilon,L_\varepsilon}^{\sigma}$$

*approximates $\psi$ uniformly:*

$$\|\psi - \hat{\psi}_\varepsilon\|_\infty < \varepsilon.$$

*Moreover, $\hat{g}_\varepsilon$ satisfies the following complexity estimates:*

- $W_\varepsilon = \mathcal{O}\left(\varepsilon^{-\frac{4p}{3(kp+1)}}\right),$

- $L_\varepsilon = \mathcal{O}\left(\varepsilon^{-\frac{2p}{3(kp+1)}}\right).$

The proof of Theorem 5.1 is postponed to Appendix H.4.

Theorem 5.1 *strengthens* the approximation guarantee of Kratsios & Papon (2022): While they control the error uniformly on a prescribed finite dataset $\mathcal{D} \subset \mathcal{M}$, we obtain a uniform approximation bound over the entire manifold $\mathcal{M}$ (under the stated smoothness assumptions).

**Step 2: Uniform Approximation of Kantorovich Potentials via the $c$-Transform.** We now lift the function-approximation guarantee of Theorem 5.1 to OT potentials. RNOT restricts the search for dual potentials to the hypothesis class $\mathfrak{C}(\varphi_\star^* \mathcal{NN}_{2p,W,L}^{\sigma})$, that is, $c$-transforms of pullback networks. Accordingly, to approximate an OT potential $\phi_\star$ it suffices to approximate its prepotential $\psi_\star := \phi_\star^c$ as an ordinary scalar function on $\mathcal{M}$. We show that if $\hat{\psi}_\varepsilon$ uniformly approximates $\psi_\star$, then its $c$-transform $\hat{\phi}_\varepsilon = \hat{\psi}_\varepsilon^c$ belongs to $\mathfrak{C}(\varphi_\star^* \mathcal{NN}_{2p,W_\varepsilon,L_\varepsilon}^{\sigma})$ and uniformly approximates $\phi_\star$ with the same $\varepsilon$-complexity. Consequently, OT potentials in the RNOT class admit polynomial $\varepsilon^{-1}$-scaling in parameter count and depth, avoiding the discretization-induced CoD identified in Section 3.

**Corollary 5.2** (Polynomial $\varepsilon^{-1}$-Complexity for RNOT Potentials)**.** *Assume $\mu, \nu \in \mathcal{P}(\mathcal{M})$ with $\mu \ll \mathrm{vol}_\mathcal{M}$, and let $\phi_\star$ be an OT potential for $c(x,y) = \frac{1}{2}d(x,y)^2$. Fix $k \in \mathbb{N}$ and assume that its prepotential $\psi_\star := \phi_\star^c$ belongs to $C^{kp,1}(\mathcal{M}, \mathbb{R})$. Let $\sigma$ be any non-affine piecewise-linear activation.*

*For any $\varepsilon \in (0,1)$, let $\hat{\psi}_\varepsilon$ be the pullback network provided by Theorem 5.1 applied to $\psi_\star$, so that $\|\psi_\star - \hat{\psi}_\varepsilon\|_\infty < \varepsilon$. Define the induced c-concave potential $\hat{\phi}_\varepsilon := \hat{\psi}_\varepsilon^c$. Then*

$$\|\phi_\star - \hat{\phi}_\varepsilon\|_\infty \leq \varepsilon,$$

*and the network implementing $\hat{\psi}_\varepsilon$ has parameter count $W_\varepsilon$ and depth $L_\varepsilon$ satisfying the same bounds as in Theorem 5.1.*

The proof of Corollary 5.2 is postponed to Appendix H.5.

The assumption $\psi_\star \in C^{kp,1}(\mathcal{M})$ should be viewed as the standard smoothness regime needed for polynomial neural approximation rates. Under classical OT regularity conditions, smooth positive densities yield smooth Kantorovich potentials, and away from the cut locus this regularity is inherited by the prepotential through the $c$-transform.

**Step 3: Pointwise Approximation of OT Maps via Stability of the Minimizer.** The transport map is defined implicitly through the inner problem $T(x) \in \arg\min_{y \in \mathcal{M}} \{\frac{1}{2}d(x,y)^2 - \psi(y)\}$, so a priori small perturbations of the potential could lead to large changes in the selected minimizer. We show that this does not happen under mild regularity assumptions: away from singular geometric configurations, a uniform $\varepsilon$-approximation of the potential yields pointwise control of the induced map on a full-$\mu$-measure subset of $\mathcal{M}$, with error bounded by $O(\sqrt{\varepsilon})$. The restriction to pointwise control is unavoidable in this formulation: the map is obtained by a minimizer selection rather than a stable linear operation on the potential, and minimizer degeneracies or cut locus effects can preclude global uniform control.

**Theorem 5.3** (Pointwise Stability of RNOT Maps)**.** *Let $\mu, \nu \in \mathcal{P}(\mathcal{M})$ satisfy $\mu, \nu \ll \mathrm{vol}_\mathcal{M}$, and consider the*

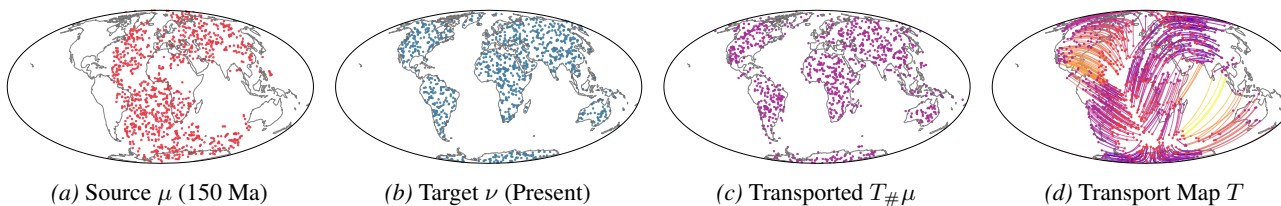

*(a)* Source $\mu$ (150 Ma)     *(b)* Target $\nu$ (Present)     *(c)* Transported $T_{\#}\mu$     *(d)* Transport Map $T$

*Figure 1.* Continental drift optimal transport on $\mathbb{S}^2$. Left to right: source mass distribution $\mu$ ($\sim 150$ million years ago), target distribution $\nu$ (present day), transported distribution $T_{\#}\mu$, and geodesic trajectories induced by the learned transport map $T$.

*quadratic cost $c(x,y) = \frac{1}{2}d(x,y)^2$. Let $\phi_\star$ be an optimal transport potential from $\mu$ to $\nu$, and set its prepotential $\psi_\star := \phi_\star^c$. Fix $k \in \mathbb{N}$ and assume $\psi_\star \in C^{kp,1}(\mathcal{M})$.*

*Then there exists a measurable set $\Omega \subset \mathcal{M}$ with $\mu(\Omega) = 1$ such that for every $x \in \Omega$ there are constants $C(x) > 0$ and $\varepsilon_x > 0$ with the following property.*

*For any $\varepsilon \in (0, \min\{1, \varepsilon_x\})$ and any continuous function $\psi_\varepsilon \in C(\mathcal{M})$ satisfying*

$$\|\psi_\varepsilon - \psi_\star\|_\infty \leq \varepsilon,$$

*define a (possibly set-valued) map $T_\varepsilon$ by selecting any minimizer*

$$T_\varepsilon(x) \in \arg\min_{y \in \mathcal{M}} \left\{ \frac{1}{2}d(x,y)^2 - \psi_\varepsilon(y) \right\}.$$

*Let $T_\star$ be the optimal transport map induced by $\phi_\star$. Then every such selection satisfies*

$$d(T_\varepsilon(x), T_\star(x)) \ \leq \ 2\sqrt{\frac{\varepsilon}{C(x)}}.$$

The proof of Theorem 5.3 is deferred to Appendix H.6.

Combining Theorem 5.3 with Corollary 5.2 yields polynomial $\varepsilon^{-1}$-complexity for pointwise approximation of the OT map $T_\star$ on a full-$\mu$-measure set, with explicit pointwise error $O(\sqrt{\varepsilon})$.

## 6. Experiments

In this section, we benchmark the empirical performance of RNOT maps against manifold normalizing-flow baselines. We first evaluate modeling fidelity on a real-world geological dataset arising from continental drift on $\mathbb{S}^2$. We then study scalability with respect to manifold dimension using synthetic experiments on high-dimensional spheres $\mathbb{S}^n$ and tori $\mathbb{T}^n$. For the RNOT implementation, we use the Gromov embedding from Proposition 2.3 and select its landmark set using either farthest-point sampling (FPS) or uniform random sampling (RND), allowing us to assess the impact of landmark geometry on performance; see Appendix E for details. Experimental details are provided in Appendix F. Code is available upon request.

### 6.1. Real-World Case Study: Continental Drift

Following Cohen et al. (2021), we study a geophysical application of manifold generative modeling: continental drift on the sphere $\mathbb{S}^2$, using paleogeographic reconstructions from Müller et al. (2018). We consider two distributions of terrestrial mass: one corresponding to the Earth $\sim 150$ million years ago (Fig. 1(a)) and one corresponding to the present-day Earth (Fig. 1(b)). Our goal is to learn an amortized map $T$ that transports the source distribution to the target.

We train an RNOT map on $\mathbb{S}^2$ using the FPS landmark embedding and the sample-based Kantorovich semi-dual objective (Appendices E and F). As shown in Fig. 1(c), the learned pushforward $T_{\#}\mu$ closely matches the present-day mass distribution. Beyond density matching, the learned transport $T$ also induces an explicit correspondence between locations on the "old" and "current" Earth.

To visualize the induced transport dynamics, we consider the one-parameter family of maps

$$T_t(x) \ = \ \exp_x\big(-t\,\nabla\phi(x)\big), \qquad t \in [0,1],$$

which interpolates between the identity ($t = 0$) and the learned transport ($t = 1$). Fig. 1(d) visualizes the induced trajectories for starting points on $\mathbb{S}^2$ sampled from $\mu$. These curves provide an interpretable picture of mass displacement consistent with plate-tectonic motion: points near the Eurasia–North America junction split into opposite directions, reflecting the separation of the two plates over geological time (cf. Wilson, 1963).

### 6.2. Synthetic Experiments on Spheres and Tori

We complement the real-world case study with controlled synthetic experiments on spheres and tori. We first compare methods in low dimensions on $\mathbb{S}^2$ and $\mathbb{T}^2$, and then repeat the same experiment while sweeping the dimension on $\mathbb{S}^n$ and $\mathbb{T}^n$ to assess scalability.

In both families, the source distribution is uniform on the manifold, and the target is a wrapped normal with scale $\sigma = 0.3$, centered at the south pole $(-1, 0, \ldots, 0)$ on $\mathbb{S}^n$ and at the analogous point $(\pi, \ldots, \pi)$ on $\mathbb{T}^n$. We compare against RCPMs (Cohen et al., 2021), RCNFs (Mathieu & Nickel,

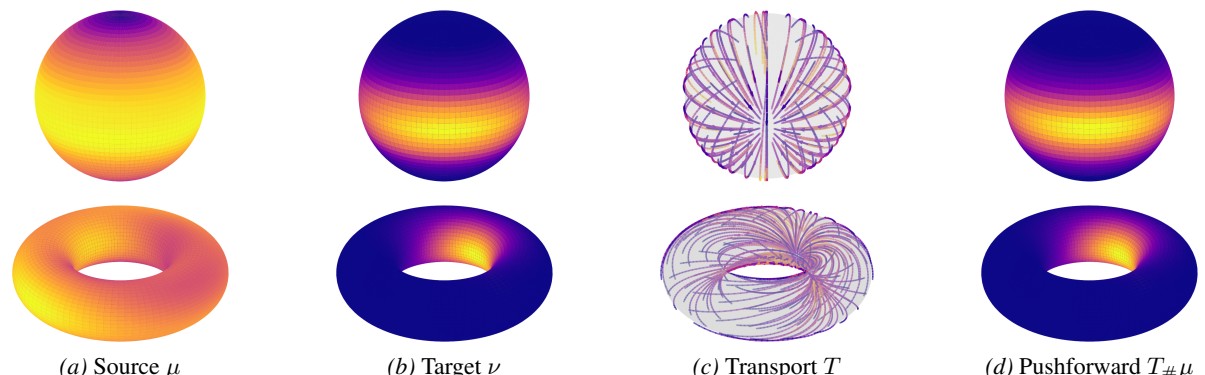

*(a) Source $\mu$*  *(b) Target $\nu$*  *(c) Transport $T$*  *(d) Pushforward $T_{\#}\mu$*

*Figure 2.* Optimal transport on $\mathbb{S}^2$ (top) and $\mathbb{T}^2$ (bottom) from a representative run corresponding to Tables 1 and 2, respectively. From left to right: uniform source $\mu$, wrapped normal target $\nu$, geodesic trajectories induced by the learned transport $T$, and pushforward $T_{\#}\mu$.

2020), and Moser Flows (Rozen et al., 2021) (see Appendix F for the rationale and implementation details). As in Cohen et al. (2021), we introduce a regularization parameter $\gamma$ and sweep $\gamma \in \{1.0, 0.1, 0.05, 0.01, 0.005, 0.001\}$, where the limit $\gamma \to 0$ corresponds to the hard (discrete) minimum. Performance is measured by the forward KL divergence $D_{\mathrm{KL}}(T_{\#}\mu \,\|\, \nu)$ and effective sample size (ESS), reported as mean $\pm$ confidence interval over 5 independent runs (1024 samples each). Following prior manifold normalizing-flow and RCPM evaluations, KL is computed using the manifold change-of-variables formula and is interpreted under the evaluation-time assumption that the learned map is locally invertible on the test samples. We also report the average wall-clock time per run (in seconds).

For all RCPM results reported here (on $\mathbb{S}^2$ and $\mathbb{T}^2$), we use $\gamma = 1$, which yields their best performance. On $\mathbb{S}^2$, RCPM achieves the lowest KL divergence (0.0037) and highest ESS (0.996), with RNOT maps with FPS landmarks performing comparably.

*Table 1.* KL divergence and ESS on $\mathbb{S}^2$ across methods. Bold denotes the best result (up to statistical significance), and underlining denotes the second best. Reported values are means with confidence intervals over 5 independent runs.

| Model | KL ↓ | ESS ↑ | Time (s) ↓ |
|---|---|---|---|
| Ours (FPS) | $\underline{0.03 \pm 0.00}$ | $\underline{0.97 \pm 0.00}$ | $986 \pm 2$ |
| Ours (RND) | $0.04 \pm 0.00$ | $0.95 \pm 0.00$ | $1100 \pm 1$ |
| RCPM | $\mathbf{0.0037 \pm 0.0008}$ | $\mathbf{0.996 \pm 0.000}$ | $\mathbf{37.3 \pm 0.6}$ |
| RCNF | $2.38 \pm 0.10$ | $0.48 \pm 0.02$ | $\underline{352 \pm 6}$ |
| Moser Flow | $1.16 \pm 0.03$ | $0.82 \pm 0.00$ | $758 \pm 16$ |

On $\mathbb{T}^2$, RNOT maps with FPS landmarks substantially outperform RCPM, attaining KL divergence 0.13 versus 0.93 and ESS 0.93 versus 0.55. RCNFs and Moser Flows perform considerably worse than both RCPM and RNOT on both $\mathbb{S}^2$ and $\mathbb{T}^2$. Figure 2 visualizes the learned RNOT transport maps on $\mathbb{S}^2$ (top row) and $\mathbb{T}^2$ (bottom row).

*Table 2.* KL divergence and ESS on $\mathbb{T}^2$ across methods. Bold denotes the best result (up to statistical significance), and underlining denotes the second best. Reported values are means with confidence intervals over 5 independent runs.

| Model | KL ↓ | ESS ↑ | Time (s) ↓ |
|---|---|---|---|
| Ours (FPS) | $\mathbf{0.13 \pm 0.01}$ | $\mathbf{0.93 \pm 0.01}$ | $1022 \pm 10$ |
| Ours (RND) | $\underline{0.22 \pm 0.04}$ | $\underline{0.85 \pm 0.02}$ | $1057 \pm 1$ |
| RCPM | $0.93 \pm 0.02$ | $0.55 \pm 0.03$ | $\mathbf{62.3 \pm 0.5}$ |
| RCNF | $6.31 \pm 0.56$ | $0.36 \pm 0.04$ | $\underline{265 \pm 6}$ |
| Moser Flow | $6.84 \pm 0.21$ | $0.56 \pm 0.01$ | $769 \pm 5$ |

This competitive performance comes with longer training time, since RNOT evaluates the implicit $c$-transform via an iterative inner minimization. Thus, the low-dimensional experiments should be interpreted as showing that RNOT remains competitive despite its higher wall-clock cost, while the main advantage of the method is its improved scalability in higher dimensions.

We next sweep the dimension. These experiments are not intended as a direct measurement of the full approximation-rate constants in Corollary 5.2. The purpose is to test its main qualitative implication under a fixed parameterization budget. As dimension increases, the continuous RNOT class remains stable while discretization-based parameterizations degrade. In Fig. 3, RCPM breaks down as $n$ grows — especially for small $\gamma$, where LogSumExp is closest to the hard minimum, as predicted by Corollary 3.2. Larger $\gamma$ smooths the objective and helps, but not enough to avoid the CoD in practice. RNOT, by contrast, stays essentially flat across all tested dimensions. Appendix G.2 provides additional controls showing that RCPM still degrades when trained with the same semi-dual loss as RNOT, and reports further experiments on $SO(3)$ and compactly supported $SE(3)$.

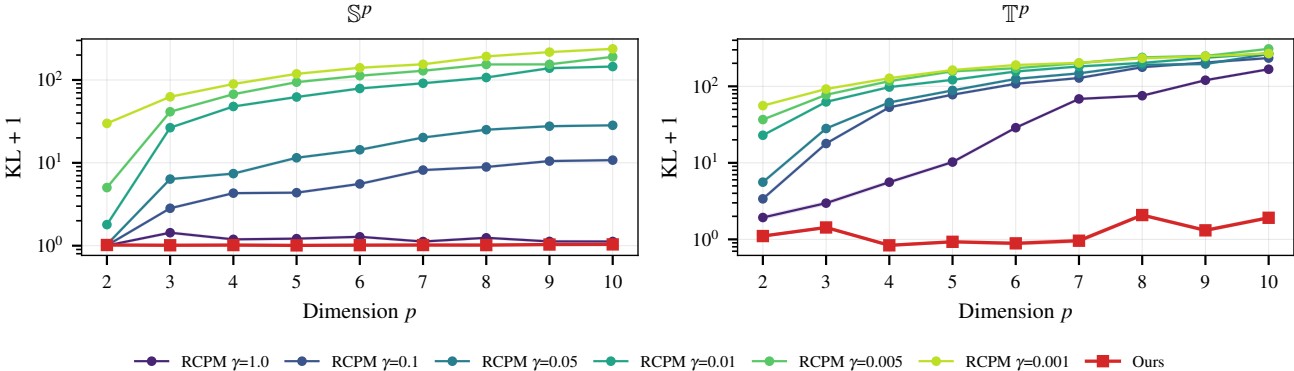

*Figure 3.* KL divergence (log scale) versus dimension $p \in \{2, \ldots, 10\}$ for transport from a uniform source to a wrapped normal target on $\mathbb{S}^p$ (left panel) and $\mathbb{T}^p$ (right panel). Curves show RNOT (ours) and RCPM under a sweep of regularization values $\gamma$. RCPM performance degrades as $p$ increases — most notably for small $\gamma$, where the map is sharp — whereas RNOT remains stable across dimensions.

Due to computational intractability, Figure 3 is restricted to a maximum dimension of 10, for which the KL divergence already saturates for small $\gamma$. Extended results up to dimension 40 are shown in Appendix Figure 5. Finally, ablation studies (Appendix G.1) show that FPS for landmark selection (vs. RND) and LogSumExp-based initialization have the largest impact on performance, highlighting the importance of landmark geometry and initialization, suggesting clear directions for future improvements.

## 7. Discussion and Limitations

This paper develops the *first* theoretically grounded neural OT framework on Riemannian manifolds. Our starting point is a negative result: any manifold OT method that outputs a *discrete* approximation of the transport map necessarily suffers from the CoD, requiring exponentially many parameters to reach a fixed accuracy. Riemannian Neural Optimal Transport (RNOT) circumvents this barrier by avoiding discretization altogether: we parameterize a continuous prepotential on $\mathcal{M}$ and enforce $c$-concavity by construction via the $c$-transform, yielding intrinsic manifold-valued maps.

For non-affine piecewise-linear activations, we are the *first* to prove explicit polynomial bounds in $\varepsilon^{-1}$ on the neural network parameter count and depth needed to approximate OT potentials and, via a stability argument, the induced OT map pointwise on a full-$\mu$-measure subset. These guarantees are reflected empirically in the dimension sweep: in Fig. 3, RCPM performance degrades sharply as $n$ increases, while RNOT remains essentially flat across all tested dimensions. Empirically, RNOT is competitive in low dimensions and remains stable under the dimension sweep: in Fig. 3, RCPM performance degrades sharply as $n$ increases, while RNOT stays essentially flat across all tested dimensions; the continental drift case study further highlights the interpretability of the learned transport trajectories.

Limitations include our focus on compact manifolds, the regularity assumptions behind the rates, the assumption that the intrinsic geometry of $\mathcal{M}$ is available, and the computational cost of the inner $c$-transform solve. Extending the framework to manifolds with boundary is an important direction, since many practical domains are naturally constrained; however, boundaries introduce additional difficulties for $c$-transforms, exp/log maps, and the regularity of both Kantorovich potentials and transport maps. Similarly, when the manifold is given only through samples, graphs, or point clouds, RNOT would require an additional geometry-estimation layer to approximate distances and local differential operators; extending our guarantees to this approximate-geometry setting is left for future work. Finally, RNOT is computationally more expensive than discrete baselines because evaluating the implicit $c$-transform requires an iterative inner minimization. This inner solve is the price of enforcing intrinsic $c$-concavity by construction, and accelerating it through warm starts, amortized $c$-transform solvers, adaptive inner-loop schedules, stronger Riemannian optimizers, or parallelization is an important direction for future work.

## Impact Statement

We develop a deep learning framework for optimal transport on Riemannian manifolds. This enables more principled modeling of geometrically structured data in domains such as biology, medicine, and the physical sciences.

## Acknowledgements

S.B. acknowledges support from the Novo Nordisk Foundation via The Novo Nordisk Young Investigator Award (NNF20OC0059309). S.B. acknowledges support from The Eric and Wendy Schmidt Fund For Strategic Innovation via the Schmidt Polymath Award (G-22-63345) which

also supports A.M. S.B. acknowledges support from the Novo Nordisk Foundation via the Global Pathogen Analysis Platform (GPAP) (NNF26SA0109818) which also supports A.M. Y.C. is supported by Digital Futures Postdoctoral Fellowship. A.Monod is supported by the EPSRC AI Hub on Mathematical Foundations of Intelligence: An "Erlangen Programme" for AI No. EP/Y028872/1. We acknowledged the use of the HX6 server at Imperial College London as well as the support of Chris De La Force.

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

# A. Review on Riemannian Geometry

This appendix collects the differential-geometric notation and facts used throughout the paper. Standard references include Lee (2018); Petersen (2016); Sakai (1996).

## A.1. Smooth Manifolds

Fix $p \in \mathbb{N}$. A *p-dimensional topological manifold* is a topological space $\mathcal{M}$ that is *Hausdorff* and *second countable*, and such that every point $x \in \mathcal{M}$ has a neighborhood homeomorphic to an open subset of $\mathbb{R}^p$. Equivalently, $\mathcal{M}$ admits an atlas: a family of pairs $\{(U_\alpha, \phi_\alpha)\}_{\alpha \in A}$ such that

$$U_\alpha \subset \mathcal{M} \text{ are open}, \qquad \bigcup_{\alpha \in A} U_\alpha = \mathcal{M}, \qquad \phi_\alpha : U_\alpha \to \phi_\alpha(U_\alpha) \subset \mathbb{R}^p \text{ are homeomorphisms.}$$

A *smooth ($C^\infty$) manifold* is a topological manifold endowed with an atlas whose transition maps

$$\phi_\beta \circ \phi_\alpha^{-1} : \phi_\alpha(U_\alpha \cap U_\beta) \to \phi_\beta(U_\alpha \cap U_\beta)$$

are $C^\infty$ diffeomorphisms; two such atlases are equivalent if their union is again smooth.

A *manifold with boundary* is defined similarly, replacing $\mathbb{R}^p$ by the half-space $\mathbb{H}^p = \{(x_1, \ldots, x_p) \in \mathbb{R}^p : x_p \geq 0\}$ in the chart ranges. Unless stated otherwise, "manifold" means *without boundary*.

**Tangent Bundles.** For $x \in \mathcal{M}$, the tangent space $T_x\mathcal{M}$ is a $p$-dimensional real vector space consisting of tangent vectors at $x$. Concretely, if $(U, x_1, \ldots, x_p)$ is a coordinate chart around $x$, then the tangent space can be identified as the real vector space spanned by the partial derivatives $\partial/\partial x_1, \ldots, \partial/\partial x_p$ at $x$. The *tangent bundle* is defined as the disjoint union $T\mathcal{M} := \bigsqcup_{x \in \mathcal{M}} T_x\mathcal{M}$. There is a canonical smooth manifold structure on $T\mathcal{M}$ which makes the natural projection $\mathrm{pr} : T\mathcal{M} \to \mathcal{M}$ a smooth map. Specifically, under the coordinate chart $(U, x_1, \ldots, x_p)$ of $\mathcal{M}$, each tangent vector can be uniquely written as

$$v = \sum_{i=1}^p v_i \frac{\partial}{\partial x_i},$$

which gives rise to induced coordinates $(\mathrm{pr}^{-1}(U), x_1, \ldots, x_p, v_1, \ldots, v_p)$ of $T\mathcal{M}$.

## A.2. Riemannian Metrics

A *Riemannian metric* on $\mathcal{M}$ is a smoothly varying inner product

$$g_x : T_x\mathcal{M} \times T_x\mathcal{M} \to \mathbb{R}, \qquad x \in \mathcal{M}.$$

It induces a norm $\|v\|_x := \sqrt{g_x(v, v)}$ on each $T_x\mathcal{M}$. The pair $(\mathcal{M}, g)$ is a *Riemannian manifold*.

Throughout all the appendices, we suppress the subscript and denote the norm of $v \in T_x\mathcal{M}$ simply by $\|v\|$ whenever the meaning is clear.

Given a piecewise $C^1$ curve $\gamma : [0, 1] \to \mathcal{M}$, its length is defined as

$$L(\gamma) := \int_0^1 \|\dot{\gamma}(t)\| \, dt.$$

The *Riemannian distance* is the induced length metric

$$d(x, y) := \inf\{L(\gamma) : \gamma(0) = x, \ \gamma(1) = y\}.$$

This turns $\mathcal{M}$ into a metric space $(\mathcal{M}, d)$. If $\mathcal{M}$ is compact, then $\mathrm{diam}(\mathcal{M}) < \infty$ and $(\mathcal{M}, d)$ is complete.

**Levi–Civita Connection.** A Riemannian metric determines a unique affine connection $\nabla$ on $T\mathcal{M}$ that is torsion-free and metric-compatible, known as the *Levi–Civita connection*. For a smooth function $f \in C^\infty(\mathcal{M})$, the *(Riemannian) gradient* $\nabla f$ is the vector field defined by $g(\nabla f, X) = df(X)$ for all vector fields $X$.

**Submanifolds and Normal Bundles.** Let $(\mathcal{M}, g)$ be a Riemannian manifold and let $\mathcal{V} \subseteq \mathcal{M}$ be a smooth submanifold. At each $x \in \mathcal{V}$, the tangent space $T_x\mathcal{V}$ is a linear subspace of $T_x\mathcal{M}$. Since $g_x$ is an inner product on $T_x\mathcal{M}$, there exists an orthogonal decomposition

$$T_x\mathcal{M} = T_x\mathcal{V} \bigoplus (T_x\mathcal{V})^\perp,$$

where $(T_x\mathcal{V})^\perp$ is the orthogonal complement of $T_x\mathcal{V}$ in $T_x\mathcal{M}$. The space $(T_x\mathcal{V})^\perp$ is called the normal space of $\mathcal{V}$ at $x$, denoted by $N_x\mathcal{V}$. The normal bundle is defined as the disjoint union $N\mathcal{V} := \bigsqcup_{x \in \mathcal{V}} N_x\mathcal{V}$. There is a canonical smooth manifold structure on $N\mathcal{V}$ which makes the natural projection $\mathrm{pr} : N\mathcal{V} \to \mathcal{V}$ a smooth map.

### A.3. Geodesics and the Exponential Map

A smooth curve $\gamma : [0, 1] \to \mathcal{M}$ is a *(affinely parametrized) geodesic* if it satisfies the geodesic equation

$$\nabla_{\dot\gamma}\dot\gamma = 0.$$

Given $x \in \mathcal{M}$ and $v \in T_x\mathcal{M}$, there exists a unique geodesic $\gamma_{x,v} : (-\varepsilon, \varepsilon) \to \mathcal{M}$ with $\gamma_{x,v}(0) = x$ and $\dot\gamma_{x,v}(0) = v$.

**Exponential map.** When $\gamma_{x,v}$ is defined at time $t = 1$, the *Riemannian exponential map* at $x$ is

$$\exp_x(v) := \gamma_{x,v}(1).$$

The exponential map is always locally well-defined due to the existence of normal neighborhoods.

**Proposition A.1** (Normal neighborhoods). *For any $x \in \mathcal{M}$ there exist neighborhoods $V \subset T_x\mathcal{M}$ of $0$ and $U \subset \mathcal{M}$ of $x$ such that $\exp_x : V \to U$ is a $C^\infty$-diffeomorphism.*

The inverse map on $U$ is the *logarithm map* $\log_x : U \to T_x\mathcal{M}$, defined by $\log_x(y) = (\exp_x)^{-1}(y)$.

**Injectivity Radius.** The *injectivity radius* at $x$ is

$$\mathrm{inj}(x) := \sup\{r > 0 : \exp_x \text{ is a diffeomorphism on } B_{T_x\mathcal{M}}(0, r)\},$$

where $B_{T_x\mathcal{M}}(0, r) = \{v \in T_x\mathcal{M} : \|v\|_x < r\}$. If $\mathcal{M}$ is compact, then $\mathrm{inj}(\mathcal{M}) := \inf_{x \in \mathcal{M}} \mathrm{inj}(x) > 0$.

**Completeness.** We always assume $\mathcal{M}$ is connected and complete. The following theorem is fundamental.

**Theorem A.2** (Hopf–Rinow). *Let $(\mathcal{M}, g)$ be a connected Riemannian manifold. The following are equivalent:*

1. *$(\mathcal{M}, d)$ is a complete metric space;*

2. *$(\mathcal{M}, g)$ is geodesically complete, i.e. $\exp_x$ is defined on all of $T_x\mathcal{M}$ for every $x \in \mathcal{M}$;*

3. *closed and bounded subsets of $(\mathcal{M}, d)$ are compact.*

*In particular, any compact Riemannian manifold is complete and any two points can be joined by a minimizing geodesic.*

### A.4. Cut Locus

For $x \in \mathcal{M}$ define the unit sphere in $T_x\mathcal{M}$ by

$$U_x\mathcal{M} := \{u \in T_x\mathcal{M} : \|u\|_x = 1\}, \qquad U\mathcal{M} := \bigsqcup_{x \in \mathcal{M}} U_x\mathcal{M}.$$

The set $U\mathcal{M}$ is a smooth submanifold of $T\mathcal{M}$ of codimension 1 (cf. Sakai (1996, Chapter III)).

**Definition A.3** (Cut time and cut locus (Sakai, 1996, Definition. 4.3)). Assume $(\mathcal{M}, g)$ is complete and fix $x \in \mathcal{M}$. For $u \in U_x\mathcal{M}$, let $\gamma_u(t) := \exp_x(tu)$ be the unit-speed geodesic with $\gamma_u(0) = x$. Define the *cut time* $t(u) \in (0, \infty]$ by

$$t(u) := \sup\left\{t > 0 : d(x, \gamma_u(s)) = s \text{ for all } s \in [0, t]\right\}.$$

If $t(u) < \infty$, the vector $t(u)u \in T_x\mathcal{M}$ is the *tangent cut point* along $\gamma_u$ and $\exp_x(t(u)u)$ is the corresponding *cut point*. The *tangent cut locus* and *cut locus* of $x$ are

$$\widetilde{\mathrm{Cut}}(x) := \{t(u)u : u \in U_x\mathcal{M}, t(u) < \infty\}, \qquad \mathrm{Cut}(x) := \exp_x(\widetilde{\mathrm{Cut}}(x)).$$

Define also the *interior domain* in $T_x\mathcal{M}$ and its image:

$$\widetilde{\mathcal{I}}_x := \{tu : u \in U_x\mathcal{M}, 0 < t < t(u)\}, \qquad \mathcal{I}_x := \exp_x(\widetilde{\mathcal{I}}_x).$$

We have the following characterizations about tangent cut points.

**Lemma A.4.** *(Petersen, 2016, Lemma 5.7.9) If $u \in T_x\mathcal{M}$ is a tangent cut point of $x$, then either*

1. *there exists $w \neq u \in T_x\mathcal{M}$ such that $\exp_x(u) = \exp_x(w)$, or*

2. *the differential $d(\exp_x)$ is singular at $u$.*

In the second case of Lemma A.4, the point $\exp_x(u) \in \mathrm{Cut}(x)$ is also called the *first conjugate point*.

The following summarizes key properties of $\mathrm{Cut}(x)$ (see Sakai (1996, Lem. 4.4)).

**Lemma A.5** (Basic properties of $\mathrm{Cut}(x)$ (Sakai, 1996, Lem. 4.4)). *Let $(\mathcal{M}, g)$ be complete and $x \in \mathcal{M}$. Then:*

1. $\mathcal{I}_x \cap \mathrm{Cut}(x) = \varnothing$, $\mathcal{M} = \mathcal{I}_x \cup \mathrm{Cut}(x)$, *and* $\overline{\mathcal{I}}_x = \mathcal{M}$;

2. $\widetilde{\mathcal{I}}_x$ *is a maximal domain containing $0 \in T_x\mathcal{M}$ on which $\exp_x$ is a diffeomorphism;*

3. $\mathrm{Cut}(x)$ *is $\mathrm{vol}_\mathcal{M}$-negligible and $\dim \mathrm{Cut}(x) \leq p - 1$, hence*

$$\mathrm{vol}_\mathcal{M}(\mathrm{Cut}(x)) = 0, \qquad \forall x \in \mathcal{M}.$$

**Squared Distance Function.** Fix $y \in \mathcal{M}$. The function $x \mapsto \frac{1}{2}d(x,y)^2$ is $C^\infty$ on $\mathcal{M} \setminus \mathrm{Cut}(y)$. Moreover, for $x \notin \mathrm{Cut}(y)$ the logarithm $\log_x(y)$ is well-defined and

$$\nabla_x\left(\tfrac{1}{2}d(x,y)^2\right) = -\log_x(y). \tag{8}$$

Symmetrically, for $y \notin \mathrm{Cut}(x)$ one has $\nabla_y(\frac{1}{2}d(x,y)^2) = -\log_y(x)$. These identities are frequently used when differentiating OT objectives involving the quadratic cost.

# B. Review on Riemannian Optimal Transport

## B.1. Formulations on Riemannian Manifolds

We briefly recall the optimal transport (OT) formalism on a compact Riemannian manifold (cf. Ambrosio et al. (2024, Chapter 7.3)). Let $(\mathcal{M}, g)$ be a smooth, connected, $n$-dimensional compact Riemannian manifold without boundary, and let $d$ denote its Riemannian distance. We write $\mathcal{B}(\mathcal{M})$ for the Borel $\sigma$-algebra of $\mathcal{M}$ and $\mathrm{vol}_\mathcal{M} \in \mathcal{M}_+(\mathcal{M})$ for the Riemannian volume measure, where $\mathcal{M}_+(\mathcal{M})$ denotes the set of finite, nonnegative Borel measures on $\mathcal{M}$. Let

$$\mathcal{P}(\mathcal{M}) := \{\mu \in \mathcal{M}_+(\mathcal{M}) : \mu(\mathcal{M}) = 1\}$$

be the set of Borel probability measures on $\mathcal{M}$. Given a Borel map $T : \mathcal{M} \to \mathcal{M}$ and $\mu \in \mathcal{P}(\mathcal{M})$, the pushforward (image) measure $T_\#\mu \in \mathcal{P}(\mathcal{M})$ is defined by

$$T_\#\mu(A) := \mu(T^{-1}(A)), \qquad \forall A \in \mathcal{B}(\mathcal{M}).$$

**The 2-Wasserstein Distance.** For $\mu, \nu \in \mathcal{P}(\mathcal{M})$, we denote by $\Pi(\mu, \nu)$ the set of *couplings* (or transport plans), i.e. probability measures $\pi \in \mathcal{P}(\mathcal{M} \times \mathcal{M})$ whose marginals are $\mu$ and $\nu$:

$$(\mathrm{pr}_1)_\#\pi = \mu, \qquad (\mathrm{pr}_2)_\#\pi = \nu,$$

where $\mathrm{pr}_i$ are the coordinate projections on $\mathcal{M} \times \mathcal{M}$. The 2-Wasserstein distance $W_2$ on $\mathcal{P}(\mathcal{M})$ is defined by

$$W_2(\mu, \nu) := \inf_{\pi \in \Pi(\mu, \nu)} \left( \int_{\mathcal{M} \times \mathcal{M}} d^2(x, y) \, \mathrm{d}\pi(x, y) \right)^{\frac{1}{2}}.$$

Since $\mathcal{M}$ is compact, the infimum is finite and is attained by at least one optimal plan $\pi^\star \in \Pi(\mu, \nu)$.

**Monge and Kantorovich Formulations.** Throughout the paper, we consider the quadratic cost

$$c(x, y) := \tfrac{1}{2} d^2(x, y).$$

The corresponding *Kantorovich problem* is

$$\mathrm{KP}(\mu, \nu) := \inf_{\pi \in \Pi(\mu, \nu)} \int_{\mathcal{M} \times \mathcal{M}} c(x, y) \, \mathrm{d}\pi(x, y) = \tfrac{1}{2} W_2^2(\mu, \nu). \tag{9}$$

The *Monge problem* is the restriction of (9) to couplings induced by maps:

$$\mathrm{MP}(\mu, \nu) := \inf_{T_\#\mu = \nu} \int_{\mathcal{M}} c\big(x, T(x)\big) \, \mathrm{d}\mu(x). \tag{10}$$

In general MP may fail to admit a minimizer (or even to be well-posed), while KP always admits at least one optimal plan on compact metric spaces.

## B.2. Duality, $c$-transform, and $c$-concavity

The dual formulation of Kantorovich problem (9) is given by

$$\mathrm{KP}(\mu, \nu) = \sup_{\substack{\varphi, \psi \in C(\mathcal{M}, \mathbb{R}) \\ \varphi(x) + \psi(y) \le c(x, y)}} \left\{ \int_{\mathcal{M}} \varphi(x) \, \mathrm{d}\mu(x) + \int_{\mathcal{M}} \psi(y) \, \mathrm{d}\nu(y) \right\}. \tag{11}$$

Given $\psi : \mathcal{M} \to \mathbb{R} \cup \{-\infty\}$, its $c$-transform is defined as

$$\psi^c(x) := \inf_{y \in \mathcal{M}} \big( c(x, y) - \psi(y) \big). \tag{12}$$

With this convention, formulation (11) can be written in the following *semi-dual* form

$$\mathrm{KP}(\mu, \nu) = \sup_{\psi \in C(\mathcal{M})} \left\{ \int_{\mathcal{M}} \psi^c(x) \, \mathrm{d}\mu(x) + \int_{\mathcal{M}} \psi(y) \, \mathrm{d}\nu(y) \right\}. \tag{13}$$

*Remark* B.1 (Sign convention). Many OT-on-manifolds constructions (and several learning objectives) prefer the equivalent reparametrization $\tilde{\psi} := -\psi$, in which case

$$\tilde{\psi}^c(x) = \inf_{y \in \mathcal{M}} \big( c(x,y) + \tilde{\psi}(y) \big), \qquad \mathrm{KP}(\mu, \nu) = \sup_{\tilde{\psi} \in C(\mathcal{M}, \mathbb{R})} \left\{ \int \tilde{\psi}^c \, \mathrm{d}\mu - \int \tilde{\psi} \, \mathrm{d}\nu \right\}. \tag{14}$$

Both conventions are identical up to the substitution $\tilde{\psi} = -\psi$.

A function $\varphi : \mathcal{M} \to \mathbb{R} \cup \{-\infty\}$ is called *c-concave* if $\varphi = \psi^c$ for some $\psi : \mathcal{M} \to \mathbb{R} \cup \{-\infty\}$. It is known that $\varphi$ is $c$-concave if and only $\varphi = \varphi^{cc}$, where $\varphi^{cc} := (\varphi^c)^c$. For the quadratic cost $c(x,y) = \frac{1}{2}d^2(x,y)$ on $\mathcal{M}$, the supremum in the semi-dual formulation (13) is always attained at some $c$-concave function.

## B.3. Optimality conditions and McCann's Theorem

**Definition B.2** (*c-subdifferential*). Let $\varphi : \mathcal{M} \to \mathbb{R} \cup \{-\infty\}$ and let $x \in \{\varphi > -\infty\}$. The $c$-subdifferential of $\varphi$ at $x$ is

$$\partial^c \varphi(x) := \Big\{ y \in \mathcal{M} : \ \varphi(x) + \varphi^c(y) = c(x,y) \Big\}.$$

Equivalently, if $\varphi = \psi^c$ as in (12), then

$$y \in \partial^c \varphi(x) \quad \Longleftrightarrow \quad y \in \arg \min_{z \in \mathcal{M}} \big( c(x,z) - \psi(z) \big).$$

Under the alternative sign convention (14), the same set is described as $\arg \min_z (c(x,z) + \tilde{\psi}(z))$.

If $(\varphi, \psi)$ is an optimal pair in (11) (with $\varphi = \psi^c$), and $\pi^\star$ is an optimal plan in (9), then

$$\varphi(x) + \psi(y) = c(x,y) \qquad \pi^\star\text{-a.e. on } \mathcal{M} \times \mathcal{M}.$$

In particular, $\mathrm{spt}(\pi^\star) \subset \{(x,y) : y \in \partial^c \varphi(x)\}$.

A central fact for the quadratic cost on a Riemannian manifold is that, under mild conditions, optimal couplings are induced by a transport map, which is known as McCann's theorem.

**Theorem B.3** (Existence and structure of optimal maps for $c = \frac{1}{2}d^2$ ((McCann, 2001))). *Let $\mu, \nu \in \mathcal{P}(\mathcal{M})$ and assume that $\mu$ is absolutely continuous with respect to $\mathrm{vol}_\mathcal{M}$. Then there exists a c-concave potential $\varphi$ such that the optimal plan for (9) is induced by a (essentially unique) map $T : \mathcal{M} \to \mathcal{M}$ satisfying*

$$T(x) \in \partial^c \varphi(x) \qquad \text{for } \mu\text{-a.e. } x \in \mathcal{M}.$$

*Moreover, $\varphi$ is differentiable $\mu$-a.e. and, at points of differentiability,*

$$T(x) = \exp_x \big( -\nabla \varphi(x) \big), \tag{15}$$

*where $\nabla$ denotes the Riemannian gradient and $\exp_x : T_x \mathcal{M} \to \mathcal{M}$ is the exponential map.*

*Remark* B.4 (Cut locus and smoothness). The function $(x,y) \mapsto \frac{1}{2}d^2(x,y)$ is smooth away from the cut locus, but generally fails to be smooth on it. As a result, $\varphi$ and the map formula (15) should be interpreted in an a.e. sense, and many quantitative statements are naturally restricted to regions avoiding neighborhoods of the cut locus.

# C. Review on Hölder Spaces

**Multi-index Notation.** Let $n \in \mathbb{N}$. For a multi-index $J = (j_1, \ldots, j_n) \in \mathbb{N}_0^n$ we set

$$|J| := j_1 + \cdots + j_n, \qquad D^J := \frac{\partial^{|J|}}{\partial x_1^{j_1} \cdots \partial x_n^{j_n}}.$$

We write $\|x\|$ for the Euclidean norm of $x \in \mathbb{R}^n$.

## C.1. Hölder Spaces on Euclidean Domains

Let $k \in \mathbb{N}_0$, $\alpha \in (0, 1]$, and $K \subset \mathbb{R}^n$ be a compact set. Suppose $f : K \to \mathbb{R}$ is a function such that all derivatives $D^J f$ with $|J| \leq k$ exist and extend continuously to the boundary of $K$. Define the seminorm

$$[f]_{C^{k,\alpha}(K)} := \sum_{|J|=k} \sup_{\substack{x,y \in K^\circ \\ x \neq y}} \frac{|D^J f(x) - D^J f(y)|}{\|x - y\|^\alpha},$$

and the Hölder norm

$$\|f\|_{C^{k,\alpha}(K)} := \sum_{j=1}^{k} \sup_{|J|=j} \|D^J f\|_\infty + [f]_{C^{k,\alpha}(K)}.$$

The Hölder space $C^{k,\alpha}(K)$ is defined as the set of all such functions with finite Hölder norm.

We will often take $K = [0,1]^n$ in our proofs. For $r = k + \alpha$ we denote

$$\mathcal{H}_{r,n} := \left\{ f \in C^{k,\alpha}([0,1]^n) : \|f\|_{C^{k,\alpha}([0,1]^n)} \leq 1 \right\}. \tag{16}$$

the unit ball in the Hölder space $C^{k,\alpha}([0,1]^n)$.

## C.2. Hölder Spaces on Smooth manifolds

Let $\mathcal{M}$ be a compact $p$-dimensional smooth manifold and fix $k \in \mathbb{N}_0$, $\alpha \in (0, 1]$. Choose a finite smooth atlas $\{(U_i, \varphi_i)\}_{i=1}^{N}$ such that each $\varphi_i : U_i \to V_i \subset \mathbb{R}^p$ is a $C^\infty$-diffeomorphism onto an open set $V_i$. Let $\{\chi_i\}_{i=1}^{N}$ be a smooth partition of unity subordinate to $\{U_i\}_{i=1}^{N}$:

$$\chi_i \in C^\infty(\mathcal{M}), \quad 0 \leq \chi_i \leq 1, \quad \operatorname{supp}(\chi_i) \subset U_i, \quad \sum_{i=1}^{N} \chi_i \equiv 1 \text{ on } \mathcal{M}.$$

For $f : \mathcal{M} \to \mathbb{R}$ define

$$f_i := (\chi_i f) \circ \varphi_i^{-1} \quad \text{on } V_i, \qquad K_i := \varphi_i(\operatorname{supp} \chi_i) \Subset V_i.$$

We say that $f \in C^{k,\alpha}(\mathcal{M})$ if $f_i \in C^{k,\alpha}(K_i)$ for all $i$ and

$$\|f\|_{C^{k,\alpha}(\mathcal{M})} := \max_{1 \leq i \leq N} \|f_i\|_{C^{k,\alpha}(K_i)} < \infty.$$

Different choices of finite atlases and subordinate partitions of unity yield equivalent norms. In particular, the Hölder space $C^{k,\alpha}(\mathcal{M})$ is well-defined up to equivalence of norms. A commonly used alternative characterization is that $f \in C^{k,\alpha}(\mathcal{M})$ if and only if for every chart $(U, \varphi)$ and every compact $K \Subset \varphi(U)$, the coordinate representative $f \circ \varphi^{-1}$ belongs to $C^{k,\alpha}(K)$.

## C.3. $C^{k,\alpha}$ Domains and Extension

We recall the notion of boundary regularity used in elliptic PDE theory and a basic extension lemma (Gilbarg & Trudinger, 2001, Section 6.2).

**Definition C.1** ($C^{k,\alpha}$ domains). Let $\Omega \subset \mathbb{R}^n$ be a bounded open set and let $k \geq 1$, $\alpha \in [0, 1]$. We say that $\Omega$ is a $C^{k,\alpha}$ *domain* if for every $x_0 \in \partial\Omega$ there exist a radius $r > 0$ and a $C^{k,\alpha}$ diffeomorphism

$$\psi : B(x_0, r) \to D \subset \mathbb{R}^n, \qquad \psi, \, \psi^{-1} \in C^{k,\alpha},$$

such that

$$\psi\big(B(x_0, r) \cap \Omega\big) \subset \mathbb{R}^n_+, \qquad \psi\big(B(x_0, r) \cap \partial\Omega\big) \subset \partial\mathbb{R}^n_+,$$

where $\mathbb{R}^n_+ := \{x \in \mathbb{R}^n : x_n > 0\}$ and $\partial\mathbb{R}^n_+ := \{x_n = 0\}$. Equivalently, $\psi$ *straightens the boundary* near $x_0$.

If $T \subset \partial\Omega$, we say that $T$ is a $C^{k,\alpha}$ *boundary portion* if the above property holds at every $x_0 \in T$, with the additional requirement that $B(x_0, r) \cap \partial\Omega \subset T$.

**Lemma C.2** (Extension from a $C^{k,\alpha}$ domain; Lemma 6.7 of (Gilbarg & Trudinger, 2001))**.** *Let $\Omega \subset \mathbb{R}^n$ be a bounded $C^{k,\alpha}$ domain with $k \geq 1$ and let $\Omega'$ be an open set with $\overline{\Omega} \subset \Omega'$. If $u \in C^{k,\alpha}(\overline{\Omega})$, then there exists $w \in C^{k,\alpha}(\Omega')$ such that $w = u$ on $\Omega$ and*

$$\|w\|_{C^{k,\alpha}(\Omega')} \leq C \|u\|_{C^{k,\alpha}(\overline{\Omega})},$$

*where $C$ depends only on $k$, $\alpha$, $\Omega$, and $\Omega'$.*

*Remark* C.3 (On norms on $\overline{\Omega}$). When we write $C^{k,\alpha}(\overline{\Omega})$, we mean that $u$ has derivatives up to order $k$ which extend continuously to $\overline{\Omega}$ and satisfy the Hölder condition of order $\alpha$ on $\overline{\Omega}$, with the norm defined as in the compact-set definition above (taking $K = \overline{\Omega}$).

# D. Related Work

**Neural optimal transport in Euclidean space.** A large body of research studies learning optimal transport (OT) between probability distributions in Euclidean spaces using neural parameterizations of either transport maps or Kantorovich potentials. A representative approach for the quadratic cost $c(x, y) = \frac{1}{2}\|x - y\|^2$ is to learn a convex Kantorovich potential and recover the Monge map via its gradient; Makkuva et al. (2020) pursue this strategy using input-convex neural networks (ICNNs) and a minimax formulation. Beyond Brenier/ICNN parameterizations, a standard route to scalability is the *semi-dual* of (entropically regularized) OT, where one dual variable is optimized while the other is recovered in closed form; this yields sample-based stochastic optimization schemes, see, e.g., Genevay et al. (2016); Cuturi & Peyré (2018); Vacher & Vialard (2023). More recent "continuous" neural OT solvers directly learn dual variables and/or transport plans, and provide out-of-sample evaluation of couplings and maps; see, e.g., Korotin et al. (2023). Neural approaches also learn Monge maps directly from samples using one-potential objectives closely related to semi-dual training; see, e.g., Rout et al. (2022); Fan et al. (2023).

**Optimal transport and generative modeling on manifolds.** From the theoretical perspective, OT on Riemannian manifolds with quadratic cost is well understood: under mild assumptions (e.g. $\mu \ll \mathrm{vol}_{\mathcal{M}}$), optimal plans concentrate on a (a.e. unique) map that can be expressed through the exponential map and a $c$-concave potential (see, e.g., McCann (2001); Villani (2016); Ambrosio et al. (2024)). In machine learning, manifold-valued generative modeling has been approached via Riemannian normalizing flows and continuous-time dynamics on manifolds (e.g. Mathieu & Nickel (2020)), as well as via constructions tailored to specific compact manifolds such as spheres and tori (e.g. Rezende et al. (2020)).

This is closely related in spirit to classical *distance-to-landmarks* embeddings of metric spaces, which represent points by their distances to a collection of reference points/sets; see Proposition 2.3 (after Gromov (1983)) for the distance-to-landmarks embedding we use in this paper. In our setting, such distance-based feature maps play a critical analytic role: they provide injective and regular Euclidean representations of $\mathcal{M}$ that enable importing quantitative approximation rates.

Stochastic approaches such as Schrödinger bridges provide an entropically regularized relaxation of optimal transport, connecting OT with diffusion processes (De Bortoli et al., 2022). These methods yield transport plans (or stochastic flows) rather than deterministic Monge maps, and thus do not directly guarantee map-based transport except in the small-noise limit. A complementary line of work introduces the notion of the Monge gap(Uscidda & Cuturi, 2023), using it as a regularization term to encourage learned transport maps to satisfy Monge-type optimality properties.

Directly related to this paper are OT-inspired manifold flows based on $c$-concave potentials. Cohen et al. (2021) introduced RCPMs, a family of expressive diffeomorphisms on compact Riemannian manifolds built from *discrete $c$-concave potentials*; the resulting maps are obtained by composing the exponential map with (sub)gradients of the learned potentials. Building on this viewpoint, Rezende & Racanière (2021) proposed *Implicit Riemannian Concave Potential Maps* (IRCPMs), which extend RCPMs by allowing more general $c$-concave potentials and by recovering the map implicitly through the solution of an inner minimization (and implicit differentiation), enabling efficient likelihood-based training and symmetry handling. In particular, Rezende & Racanière (2021) are (to our knowledge) among the first to make this OT-inspired inner minimization an explicit *implicit layer* in a manifold flow model and to differentiate through the argmin via implicit differentiation.

**Universal approximation on manifolds and curse-of-dimensionality considerations.** Our approximation arguments combine two themes: (i) reducing approximation on a manifold to approximation in Euclidean space via an injective/sufficiently regular feature map, and (ii) importing quantitative approximation rates from Euclidean approximation theory. On the universality side, Kratsios & Bilokopytov (2020) give general conditions under which composing a Euclidean universal approximator with suitable feature/readout maps yields universality for non-Euclidean learning problems, including manifold-valued settings. A quantitative and differentiable-geometric perspective is developed in Kratsios & Papon (2022), which studies approximation on manifolds under geometric constraints and identifies regimes where data-dependent conditions can soften or avoid classical curse-of-dimensionality behavior. On the Euclidean approximation side, Yarotsky & Zhevnerchuk (2020) establish fast ("uncursed") approximation regimes for certain architectures/activations, including nearly exponential rates for suitable periodic/structured activations. In comparison, approximation results for generic ReLU networks on manifolds typically yield (polynomial-in-$\varepsilon^{-1}$) rates governed by the intrinsic dimension, up to logarithmic factors; see, e.g., Schmidt-Hieber (2019); Chen et al. (2019). Our functional approximation theorem is complementary: it targets the particular geometric function classes arising from squared-distance OT (notably $c$-concave potentials) and combines a globally regular feature-map reduction with Euclidean *uncursed* approximation regimes, leading to different complexity scalings when such Euclidean rates are available.

**Positioning of our contribution.** The above works motivate our focus: we study OT potentials and Monge maps *on compact Riemannian manifolds* with quadratic cost, and we develop approximation guarantees tailored to the geometric OT structure. In contrast to Euclidean neural OT, our analysis must explicitly account for geometric singularities (e.g. cut loci) where the squared distance and the associated $c$-transforms lose smoothness. In contrast to manifold flow constructions that emphasize likelihood training and empirical expressivity, we provide a quantitative approximation theory for $c$-concave OT potentials (and the induced transport maps) using RNOT parameterizations, with rates that mirror uncursed Euclidean approximation once an appropriate feature map/embedding mechanism is in place.

# E. Implementation Details

## E.1. Gromov Embedding Implementation

*Landmark selection.* Our embedding layer represents each point $x \in \mathcal{M}$ by its distances to a set of landmarks $L = \{\ell_j\}_{j=1}^M \subset \mathcal{M}$,

$$\varphi(x) := (d(x, \ell_j))_{j=1}^M \in \mathbb{R}^M.$$

We consider two practical strategies to choose $L$:

(i) *Random landmarks (RND):* we sample $\ell_j$ i.i.d. from a reference measure on $\mathcal{M}$ (in our experiments, the uniform measure on $\mathcal{M}$).

(ii) *Farthest-point sampling (FPS):* to obtain a more space-filling landmark set, we run a greedy $k$-center procedure on a candidate pool $C = \{c_n\}_{n=1}^N \subset \mathcal{M}$ (either sampled from the same reference measure or taken from training data).

FPS initializes with a random seed $\ell_1 \in C$ and then iteratively selects

$$\ell_{t+1} := \arg\max_{c \in C} \min_{1 \le j \le t} d(c, \ell_j), \qquad t = 1, \dots, M - 1,$$

i.e., each new landmark is the candidate point farthest from the current landmark set in geodesic distance. To quantify the space-filling property, we define the *(empirical) covering radius* of a landmark set $L$ with respect to $C$ as

$$R(L; C) := \max_{c \in C} \min_{\ell \in L} d(c, \ell),$$

i.e., the smallest radius $r$ such that $C \subset \bigcup_{\ell \in L} B(\ell, r)$. The discrete $k$-center objective is to minimize $R(L; C)$ over all $|L| = M$ subsets of $C$. FPS is a standard greedy approximation to this objective (often called *farthest-first traversal*) and tends to produce well-spread landmarks with small covering radius (see e.g. Williamson & Shmoys (2012); Eppstein et al. (2020)). Moreover, it yields *nested* landmark sets: the first $M$ landmarks are a prefix of those obtained for any larger $M' > M$.

*Embedding diagnostics and choice of $M$.* Theoretical injectivity conditions for distance-coordinate embeddings depend on geometric constants that are typically unavailable in closed form. Consequently, we choose $M$ using empirical diagnostics on a held-out set $X_{\text{val}} \subset \mathcal{M}$. For each candidate $M$, we compute embeddings $z_n = \varphi(x_n)$ for $x_n \in X_{\text{val}}$ and monitor *non-collapse* statistics in the embedded space. Specifically, we estimate the minimum pairwise separation

$$s_M := \min_{a \ne b} \|z_a - z_b\|_2,$$

and the fraction of near-collisions

$$\rho_M(\varepsilon) := \frac{1}{|P|} \sum_{(a,b) \in P} \mathbf{1}_{\{\|z_a - z_b\|_2 < \varepsilon\}},$$

where $P$ is a set of sampled distinct pairs from $X_{\text{val}}$ (we subsample pairs for efficiency). Equivalently, $\rho_M(\varepsilon)$ is an empirical estimate of $\Pr(\|\varphi_M(x) - \varphi_M(y)\|_2 < \varepsilon)$ under the sampling scheme used to draw pairs $(x, y)$ from $X_{\text{val}}$.

**Embedding diagnostics and choice of $M$.** Theoretical injectivity conditions for distance-coordinate embeddings depend on geometric constants that are typically unavailable in closed form. Consequently, we choose $M$ using empirical diagnostics on a held-out validation set $X_{\text{val}} \subset \mathcal{M}$. For each candidate $M$, we compute embeddings $z_n = \varphi(x_n) \in \mathbb{R}^M$ for $x_n \in X_{\text{val}}$, where $\varphi(x) = (d(x, \ell_1), \dots, d(x, \ell_M))$ is the landmark distance embedding. We monitor non-collapse statistics in the embedded space: the minimum pairwise separation $s_M$ and the near-collision fraction $\rho_M(\varepsilon)$. We additionally report the empirical coverage radius

$$R_M^{\text{val}} := \max_{x \in X_{\text{val}}} \min_{1 \le j \le M} d(x, \ell_j),$$

which measures how well the landmark set covers the region of interest.

In practice, we repeat these estimates over multiple random subsets of $X_{\text{val}}$ to reduce variance, and select the smallest $M$ such that $s_M$ is stably above a numerical tolerance and $\rho_M(\varepsilon) \approx 0$, which provides a practical proxy for injectivity at the resolution of the data. Figure 4 compares RAND landmark sampling against FPS, demonstrating that FPS achieves lower coverage radius and comparable separation with fewer landmarks (see Figure 4).

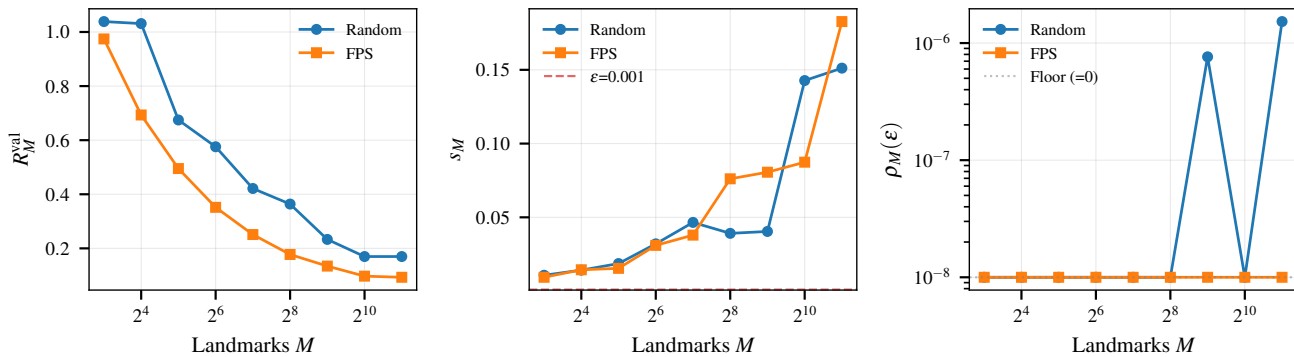

*Figure 4.* Embedding diagnostics on $\mathbb{S}^2$: coverage radius $R_M^{\text{val}}$ (left), minimum pairwise separation $s_M$ (center), and near-collision fraction $\rho_M(\varepsilon)$ (right) for random vs. farthest-point sampling (FPS). FPS achieves lower coverage radius while maintaining comparable separation, with $\rho_M \approx 0$ for all $M$ tested.

## E.2. Training RNOT Maps via the Kantorovich Semi-Dual

We now describe how RNOT maps are trained in practice. This section provides details on the feature representations used to parameterize prepotentials on the manifold, the optimization objective, and the numerical treatment of the $c$-transform. Our proposed procedure allows for end-to-end training of RNOT maps using only samples from the source and target distributions, without discretizing the manifold nor evaluating Jacobian determinants. The implementation detailed here aligns with the prior theory and is designed to be stable and scalable in high-dimensional settings.

We begin by describing the feature map used to represent functions on $\mathcal{M}$.

**Distance-to-Landmarks Feature Representation.** We use a distance-coordinate feature map inspired by Proposition 2.3. In principle, that result guarantees the existence of a geometric threshold $\delta_0 > 0$ such that, for any $\delta \in (0, \delta_0)$ and any maximal $\delta$-separated set $S = \{x_i\}_{i=1}^{I} \subset \mathcal{M}$, the map

$$\varphi_S(x) \;=\; \big(d(x, x_i)\big)_{i=1}^{I}$$

is injective, hence a topological embedding on the compact manifold $\mathcal{M}$. Since $\delta_0$ depends on the (typically unknown) geometry of $\mathcal{M}$ and is not available in closed form, we do not attempt to explicitly construct such an $S$.

Instead, we form a (larger) landmark set $L = \{\ell_j\}_{j=1}^{M}$ using either random sampling or farthest-point sampling (Appendix E.1) and define

$$\varphi(x) \;:=\; \big(d(x, \ell_j)\big)_{j=1}^{M} \in \mathbb{R}^M. \tag{17}$$

This feature map is continuous, and increasing $M$ cannot destroy injectivity: if $L$ contains an injective subset (as in Proposition 2.3), then $\varphi$ is injective as well. In practice, we treat $M$ as a tunable resolution parameter and increase it until the representation is empirically non-collapsing on held-out samples. Importantly, we emphasize that these landmarks are used solely to construct a continuous feature representation of functions on $\mathcal{M}$; they do not discretize the transport problem itself, and the resulting RNOT maps remain fully continuous with unrestricted output support. Concretely, we monitor the embedding diagnostics described in Appendix E.1 on a validation set, and choose $M$ so that these quantities remain above a fixed numerical tolerance.

Given this representation, we now give the objective used to train RNOT maps.

**Training Objective: Kantorovich Semi-Dual.** Let $\mu$ denote the source distribution on $\mathcal{M}$ and $\nu$ the target distribution. RNOT maps induce a transport map $T_\theta : \mathcal{M} \to \mathcal{M}$ and thus a model distribution $\nu_\theta := T_{\theta\#}\mu$. A standard alternative is to train by maximum likelihood (equivalently minimizing $\text{KL}(\nu_\theta \| \nu)$ up to a constant), but this requires evaluating $\log \nu_\theta$ via a change-of-variables formula and computing Jacobian log-determinants of $T_\theta$, which becomes costly and often unstable in high-dimensional settings. Unlike discretization-based OT methods, the use of landmarks here does not restrict the support of the learned transport map.

Instead, we train using the Kantorovich semi-dual objective, which depends only on samples from $\mu$ and $\nu$

$$\mathcal{J}(\psi) := \mathbb{E}_{x \sim \mu}\big[\psi^c(x)\big] \; + \; \mathbb{E}_{y \sim \nu}\big[\psi(y)\big],$$
$$W_2(\mu, \nu) := \sup_{\psi} \mathcal{J}(\psi).$$

We parameterize $\psi$ as $\psi_\theta = f_\theta \circ \varphi$, where $f_\theta$ is an MLP acting on the landmark features (17), and maximize $\mathcal{J}(\psi_\theta)$. Equivalently, we minimize the negative semi-dual loss

$$\mathcal{L}(\theta) := -\mathbb{E}_{x \sim \mu}[\psi_\theta^c(x)] - \mathbb{E}_{y \sim \nu}[\psi_\theta(y)]$$

Given mini-batches $\{x_i\}_{i=1}^B \sim \mu$ and $\{y_j\}_{j=1}^B \sim \nu$, we use the Monte Carlo estimator

$$\widehat{\mathcal{L}}(\theta) = -\frac{1}{B} \sum_{i=1}^B \psi_\theta^c(x_i) - \frac{1}{B} \sum_{j=1}^B \psi_\theta(y_j).$$

Evaluating the semi-dual objective requires solving an inner optimization problem, which we address next.

**Initializing the Inner Minimization.** Evaluating the $c$-transform requires solving an inner optimization problem on $\mathcal{M}$. To improve stability and convergence, we start this inner loop using a softmin (LogSumExp) approximation computed over a finite set of target samples $\{y_k\}_{k=1}^K \sim \nu$. Specifically, we define

$$y_0(x) = \Pi_{\mathcal{M}}\Big( \sum_{k=1}^K \text{softmax}_k\Big( \tfrac{\psi_\theta(y_k) - c(x, y_k)}{\gamma} \Big) y_k \Big).$$

where $\gamma > 0$ is a temperature parameter and $\Pi_{\mathcal{M}}$ denotes projection onto $\mathcal{M}$. Gradients are stopped through this initialization. Starting from $y_0(x)$, the inner problem can be solved by Riemannian gradient descent (with or without momentum).

**Differentiating through the $c$-Transform.** The main computational challenge is evaluating and differentiating $\psi_\theta^c(x)$, which involves an inner minimization. Define

$$F_\theta(x, y) := c(x, y) - \psi_\theta(y),$$
$$\psi_\theta^c(x) := \min_{y \in \mathcal{M}} F_\theta(x, y),$$
$$y_\theta^\star(x) \in \arg\min_{y \in \mathcal{M}} F_\theta(x, y).$$

Assume that for the current $(x, \theta)$ the minimizer $y_\theta^\star(x)$ is (locally) unique and that $F_\theta(x, \cdot)$ is differentiable at $y_\theta^\star(x)$, satisfying the first-order optimality condition

$$\nabla_y F_\theta\big(x, y_\theta^\star(x)\big) = 0, \tag{18}$$

where $\nabla_y$ denotes the Riemannian gradient in the second argument. Then, differentiating via the chain rule and invoking the first-order optimality condition (18) to eliminate the derivative through the minimizer, we obtain

$$\nabla_\theta \psi_\theta^c(x) = \nabla_\theta F_\theta\big(x, y_\theta^\star(x)\big) = \nabla_\theta \psi_\theta\big(y_\theta^\star(x)\big), \tag{19}$$

since $c$ does not depend on $\theta$. In practice, this corresponds to differentiating $\psi_\theta(y)$ while *stopping gradients* through the argmin $y_\theta^\star(x)$.

Our inner solver returns an approximate minimizer $\tilde{y}_\theta(x)$, so (19) holds only approximately. The induced bias is controlled by the stationarity residual

$$r_\theta(x, \tilde{y}) := \big\| \nabla_y F_\theta(x, \tilde{y}) \big\|_2.$$

For the squared-distance cost $c(x, y) = \tfrac{1}{2} d(x, y)^2$, we have

$$\nabla_y \Big( \tfrac{1}{2} d(x, y)^2 \Big) = -\log_y(x),$$

so the stationarity condition becomes $-\log_y(x) - \nabla \psi_\theta(y) = 0$. Accordingly, we monitor the diagnostic

$$g_\theta(x, \tilde{y}) := -\log_{\tilde{y}}(x) - \nabla \psi_\theta(\tilde{y}), \qquad \|g_\theta(x, \tilde{y})\|_2,$$

during training to ensure that the inner minimization is sufficiently accurate for stable outer-loop optimization.

## E.3. Algorithm

Algorithm 1 below can be implemented with either heavy ball (momentum) or Riemannian Adam by changing line 10. For simplicity of exposition we present this algorithm using standard gradient descent. Additionally we exclude from our presentation any initialisation approaches (e.g. based on LogSumExp).

---

**Algorithm 1** Semi-dual RNOT maps with landmark distance embeddings

---

**Require:** Manifold $(\mathcal{M}, g)$; source $\mu$, target $\nu$ on $\mathcal{M}$; landmarks $\{\ell_m\}_{m=1}^{L} \subset \mathcal{M}$; cost $c(x, y) = \frac{1}{2} d(x, y)^2$; batch size $B$; steps $T$; inner steps $K$; step sizes $\eta, \alpha$.

1: Define embedding $\varphi(x) \in \mathbb{R}^L$ by $\varphi_m(x) = d(x, \ell_m)$ for $m = 1, \ldots, L$.
2: Parameterize potential $\psi_\theta(x) = \text{MLP}_\theta(\varphi(x))$.
3: Initialize $\theta$.
4: **for** $t = 1, \ldots, T$ **do**
5:     Sample $\{x_i\}_{i=1}^{B} \sim \mu$ and $\{y_j\}_{j=1}^{B} \sim \nu$.
6:     **for** $i = 1, \ldots, B$ **do**
7:         $y \leftarrow x_i$.
8:         **for** $k = 1, \ldots, K$ **do**
9:             $g \leftarrow \nabla_y \big( c(x_i, y) - \psi_\theta(y) \big) = -\log_y(x_i) - \nabla \psi_\theta(y)$.
10:            $y \leftarrow \text{Exp}_y(-\alpha g)$.
11:         **end for**
12:         $\psi_\theta^c(x_i) \leftarrow c(x_i, y) - \psi_\theta(y)$    ($y$ as constant for $\nabla_\theta$ [envelope theorem]).
13:     **end for**
14:     $\mathcal{L}(\theta) \leftarrow -\frac{1}{B} \sum_{j=1}^{B} \psi_\theta(y_j) - \frac{1}{B} \sum_{i=1}^{B} \psi_\theta^c(x_i)$.
15:     $\theta \leftarrow \text{Update}(\theta, \nabla_\theta \mathcal{L}(\theta))$.
16: **end for**
17: **Return:** $\psi_\theta$.

---

# F. Experiment Set-up

## F.1. Real World Experiments

**Dataset and Geometric Construction.** Following Cohen et al. (2021), we apply our framework to the study of continental drift (Müller et al., 2018). We construct source and target empirical distributions on $\mathbb{S}^2$, representing land masses at 150 Ma (Jurassic) and present day (0 Ma). Paleogeographic coastline reconstructions are obtained from the GPlates Web Service (Müller et al., 2018) using the Zahirovic et al. (2022) plate motion model. We rejection-sample 50,000 points uniformly on $\mathbb{S}^2$ conditioned on lying within reconstructed continental boundaries. During training, we sample uniformly from this empirical point cloud; for density evaluation, using a Gaussian KDE with bandwidth $h = 0.1$.

**Training Setup.** We train our model for 500 iterations, with 1024 batch size and landmarks, and up to 1000 internal solves per iteration. The resulting optimal transport map $T : \mathbb{S}^2 \to \mathbb{S}^2$ captures the aggregate motion of continental drift over 150 million years *with no knowledge of plate tectonics*.

**Qualitative Results.** Figure 1 illustrates the transported map alongside the base and target densities. The transported configurations recover characteristic large-scale patterns of continental drift, including coherent plate motion and long-range mass transport, demonstrating that the model learns a meaningful geometric correspondence between paleogeographic configurations.

**Qualitative Validation.** We validate that the learned map preserves tectonic structure by measuring *plate purity*, defined as the fraction of points from each source plate that are transported to a single destination plate. Transported points are assigned plate labels via nearest-neighbor lookup in the present-day point cloud. The learned map achieves a mean plate purity of 90% mean (as a fraction of points from each source plate transported to a single destination) and a Monge gap

$$\mathbb{E}[c(x, T(x))] + \mathcal{L}(\theta) < 0.1\%,$$

indicating near-optimal transport (Uscidda & Cuturi, 2023).

## F.2. Synthetic Experiments

**Data and Geometric Setup.** We evaluate transport quality on two families of Riemannian manifolds: the $n$-dimensional unit sphere $\mathbb{S}^n = \{x \in \mathbb{R}^{n+1} : |x| = 1\}$ and the $n$-dimensional flat torus $\mathbb{T}^n = (\mathbb{S}^1)^n$ (product manifold), for dimensions $n \in 2, \ldots, 10$. In both settings, the source distribution is uniform over the manifold, and the target is a wrapped normal with scale $\sigma = 0.3$, centered at the south pole $(-1, 0, \ldots, 0)$ on the sphere and at the analogous point $(\pi, \ldots, \pi)$ on the torus.

**Baselines and Metrics.** We compare our method against RCPMs (Cohen et al., 2021), RCNFs (Mathieu & Nickel, 2020) and Moser Flows (Rozen et al., 2021). In the case of RCPMs, a sweep was conducted across regularization parameters

$$\gamma \in \{1.0, 0.1, 0.05, 0.01, 0.005, 0.001\},$$

where smaller $\gamma$ sharpens the cost function and approaches an exact transport map in the limit $\gamma \to 0$.

Performance is reported using the forward Kullback–Leibler (KL) divergence $D_{\mathrm{KL}}(T_{\#}\mu|\nu)$ and effective sample size (ESS); metrics are averaged over 5 batches of 1024 samples.

**Training Protocol.** We train RNOT maps (our method) using a 2-layer neural network with 128 nodes per layer; 128 landmarks; a batch size of 256; and 1,000 training steps, with a maximum of 2,500 inner solve iterations. RCPMs are trained using the hyperparameters of (Cohen et al., 2021), with 68 landmarks, a batch size of 256, and 5,000 iterations. RCNFs and Moser flow were trained with batch size 512 for 5,000 iterations (RCNF) and 10,000 iterations (Moser Flow) using the default architecture and hyperparameters. All experiments were conducted on AMD MI300X GPUs with 192 GB of memory. We observe that RCPMs become numerically unstable as dimension increases, particularly for small $\gamma$, limiting our comparison to $n \leq 10$ with 5,000 training iterations.

**Ablations and Sensitivity.** Ablation studies (Appendix G.1) reveal that FPS landmark selection and LogSumExp-based initialization are the most impactful design choices, indicating that landmark geometry and initialization strategies are important factors driving performance and promising directions for future work.

## F.3. Metrics

To quantitatively assess how well the learned transport map matches the target distribution, we report an estimate of the Kullback–Leibler (KL) divergence between the pushforward of the base measure and the target, together with an effective sample size (ESS) diagnostic based on importance weights.

Let $\mu$ denote the source distribution on the manifold $\mathcal{M}$, $\nu$ the target distribution, and let $T_\theta : \mathcal{M} \to \mathcal{M}$ be the learned transport map. The model distribution is the pushforward

$$\nu_\theta = (T_\theta)_\# \mu.$$

**KL Divergence.** We track the KL divergence

$$\mathrm{KL}(\nu_\theta \| \nu) := \int_{\mathcal{M}} \log\left(\frac{\mathrm{d}\nu_\theta}{\mathrm{d}\nu}(y)\right) \mathrm{d}\nu_\theta(y) = \mathbb{E}_{y \sim \nu_\theta}[\log \nu_\theta(y) - \log \nu(y)]$$

In practice we estimate this quantity by Monte Carlo using samples $x_i \sim \mu$ and their transported points $y_i = T_\theta(x_i)$.

Suppose $\mu$ and $\nu$ admit densities (still denoted $\mu(\cdot)$, $\nu(\cdot)$) with respect to $\mathrm{vol}_{\mathcal{M}}$. Let $T_\theta : \mathcal{M} \to \mathcal{M}$ be a $C^1$ diffeomorphism, and define the pushforward $\nu_\theta := (T_\theta)_\# \mu$. Then $\nu_\theta$ also admits a density with respect to $\mathrm{vol}_{\mathcal{M}}$, and the manifold change-of-variables formula gives, for $y = T_\theta(x)$,

$$\nu_\theta(T_\theta(x))\big|\det(\mathrm{d}T_\theta(x))\big| = \mu(x).$$

where $\big|\det(\mathrm{d}T_\theta(x))\big|$ denotes the absolute determinant of the linear map $\mathrm{d}T_\theta(x) : T_x\mathcal{M} \to T_{T_\theta(x)}\mathcal{M}$ expressed in orthonormal bases (Falorsi et al., 2019, Lemma D.1); see also Stern (2013, Section 2). Equivalently,

$$\log \nu_\theta(y) = \log \mu(x) - \log|\det(\mathrm{d}T_\theta(x))|, \qquad y = T_\theta(x).$$

For a smooth map $T_\theta : \mathcal{M} \to \mathcal{M}$, its differential at $x \in \mathcal{M}$ is the linear map

$$\mathrm{d}T_\theta(x) : T_x\mathcal{M} \to T_{T_\theta(x)}\mathcal{M}$$

defined by how $T_\theta$ pushed forward tangent vectors at $x$. Concretely, if $\gamma(t)$ is a smooth curve with $\gamma(0) = x$ and $\dot{\gamma}(0) = v \in T_x\mathcal{M}$ then

$$\mathrm{d}T_\theta(x)[v] = \frac{\mathrm{d}}{\mathrm{d}t}\bigg|_{t=0} T_\theta(\gamma(t)) \in T_{T_\theta(x)}\mathcal{M}.$$

Because $\mathrm{d}T_\theta(x)$ maps between different vector spaces $T_x\mathcal{M}$ and $T_y\mathcal{M}$ (with $y = T_\theta(x)$), its Jacobian determinant is defined using the Riemannian metric. Assume $\mathcal{M}$ is embedded in an ambient $\mathbb{R}^D$ and $T_\theta$ is represented by a smooth ambient map $\tilde{T}_\theta : \mathbb{R}^D \to \mathbb{R}^D$ with $\tilde{T}_\theta|_{\mathcal{M}} = T_\theta$. Let $E_x \in \mathbb{R}^{D \times p}$ and $E_y \in \mathbb{R}^{D \times p}$ have columns forming orthonormal bases of $T_x\mathcal{M}$ and $T_y\mathcal{M}$, respectively. The matrix of $\mathrm{d}T_\theta(x)$ in these bases is

$$J(x) = E_y^\top (\mathrm{d}\tilde{T}_\theta(x)) E_x \in \mathbb{R}^{p \times p}.$$

The intrinsic Jacobian determinant is then

$$\big|\det(\mathrm{d}T_\theta(x))\big| \equiv \big|\det(J(x))\big|, \qquad \log\big|\det(\mathrm{d}T_\theta(x))\big| = \log\big|\det(J(x))\big|.$$

This quantity does not depend on the particular choice of orthonormal bases: changing bases multiplies $J$ on the left/right by orthogonal matrices, which changes $\det(J)$ only by a sign, and hence leaves $|\det(J)|$ invariant.

Plugging this into the KL definition gives the estimator

$$\widehat{\mathrm{KL}}(\nu_\theta \| \nu) = \frac{1}{N} \sum_{i=1}^{N} (\log \mu(x_i) - \log|\det(\mathrm{d}T_\theta(x_i))| - \log \nu(y_i)).$$

**Effective Sample Size.** Alongside KL, we compute an importance-sampling diagnostic based on weights that compare the target density to the model density on transported samples. Define the (unnormalized) importance weights

$$w_i = \frac{\nu(y_i)}{\nu_\theta(y_i)} = \exp(\log \nu(y_i) - \log \nu_\theta(y_i)).$$

The empirical normalizing constant estimate is

$$\widehat{Z} = \frac{1}{N} \sum_{i=1}^{N} w_i,$$

which should be close to 1 when $\nu_\theta$ and $\nu$ have similar support and the density ratio is well-behaved. Using these weights, we report an effective sample size (ESS),

$$\text{ESS} = \frac{\left(\sum_{i=1}^{N} w_i\right)^2}{\sum_{i=1}^{N} w_i^2} = \frac{1}{\sum_{i=1}^{N} \tilde{w}_i^2},$$

where $\tilde{w}_i = w_i / \sum_j w_j$ are the normalized weights. When $T_\theta$ is the optimal transport map and $\nu_\theta = \nu$, all weights are equal and $\text{ESS} = N$. We report the normalized ratio $\text{ESS}/N \in [0, 1]$.

The ESS quantifies weight degeneracy: $\widehat{\text{ESS}} \approx N$ indicates near-uniform weights (good overlap), whereas small ESS indicates that only a few samples carry most of the mass (poor overlap).

**Computing the Jacobian via the Implicit Function Theorem.** While we provide code to backpropagate through the iterative inner solver, unrolling many iterations can be numerically brittle (and memory-intensive) when performed at scale. Instead, we compute the transport Jacobian $\mathrm{d}T_\theta(x)$ by *implicit differentiation* using the Implicit Function Theorem (IFT) (e.g. Lee (2012, Chapter ye 4); see also implicit-layer discussions in Blondel et al. (2022)).

The transport map $T_\theta(x) = y^\star(x)$ is defined implicitly as the solution of the first-order optimality condition

$$F(x, y) := -\log_y(x) - \nabla \psi_\theta(y) = 0,$$

where $\log_y(x) \in T_y \mathcal{M}$ is the Riemannian logarithm (at pairs $(y, x)$ away from the cut locus) and $\nabla \psi_\theta(y) \in T_y \mathcal{M}$ is the Riemannian gradient of the dual potential. In particular, $F(x, y) \in T_y \mathcal{M}$ is a tangent vector at the output point $y$.

Assuming the partial differential in the second argument,

$$D_y F(x, y^\star(x)) : \ T_{y^\star(x)} \mathcal{M} \to T_{y^\star(x)} \mathcal{M},$$

is invertible, the IFT implies that $y^\star(x)$ is locally differentiable and its differential satisfies

$$\mathrm{d}T_\theta(x) = -\big(D_y F(x, y^\star(x))\big)^{-1} \circ D_x F(x, y^\star(x)),$$

where $D_x F(x, y^\star(x)) : T_x \mathcal{M} \to T_{y^\star(x)} \mathcal{M}$.

To obtain an intrinsic $p \times p$ Jacobian matrix, we work in orthonormal tangent bases. Let $E_x \in \mathbb{R}^{D \times p}$ and $E_y \in \mathbb{R}^{D \times p}$ have columns forming orthonormal bases of $T_x \mathcal{M}$ and $T_{y^\star(x)} \mathcal{M}$, respectively (in an ambient representation $\mathcal{M} \subset \mathbb{R}^D$). Using automatic differentiation on the ambient implementation of $F$, we form the ambient Jacobian operators

$$\big(D_y F\big)_{\text{amb}} \in \mathbb{R}^{D \times D}, \qquad \big(D_x F\big)_{\text{amb}} \in \mathbb{R}^{D \times D},$$

and project them to tangent coordinates:

$$[D_y F] = E_y^\top \big(D_y F\big)_{\text{amb}} E_y \in \mathbb{R}^{p \times p}, \qquad [D_x F] = E_y^\top \big(D_x F\big)_{\text{amb}} E_x \in \mathbb{R}^{p \times p}. \tag{20}$$

The intrinsic Jacobian matrix $J(x)$ is then obtained by solving the $p \times p$ linear system

$$J(x) = -[D_y F]^{-1} [D_x F], \tag{21}$$

and we compute the change-of-variables term via $\log|\det(J(x))|$.

The per-sample cost is dominated by $O(p^3)$ linear algebra (solve and determinant), plus the cost of obtaining the projected Jacobians in (20) (which can be implemented efficiently via JVP/VJP products with the $p$ basis vectors, rather than forming full $D \times D$ matrices).

Estimating KL divergence on manifolds can be numerically delicate because it combines log-densities and Jacobian determinants, so small errors may compound. In addition, the Monte Carlo estimator of KL can be negative for finite $N$ due to sampling variability, and numerical error can exacerbate this effect; we observed occasional negative *estimates* in both Cohen et al. (2021) and our own implementation.

The IFT-based Jacobian is also sensitive to inner-solver accuracy: if the transported point $y^\star(x)$ does not satisfy the stationarity condition $F(x, y^\star(x)) \approx 0$ to sufficient precision, or if $[D_y F]$ is ill-conditioned, then (21) becomes unreliable. Empirically, we find that residual norms below $10^{-2}$ yield stable KL estimates, though this threshold is problem-dependent.

Finally, we emphasize that the semi-dual objective optimizes transport quality directly, not KL divergence. Consequently, a well-trained model may yield excellent transport maps (e.g. in cost or visual diagnostics) while the KL *estimates* remain noisy. We therefore interpret KL primarily as a diagnostic summary rather than a training target.

**Rationale for Alternative Methods Presented.** We compare RNOT to representative normalizing-flow baselines on manifolds: RCNFs (Mathieu & Nickel, 2020), Moser Flow (Rozen et al., 2021), and Riemannian Convex Potential Maps (RCPMs) (Cohen et al., 2021). These methods represent the major areas of learning learning transport maps on Riemannian manifolds: ODE-based flows (RCNF), divergence-based density matching (Moser), and discrete optimal transport via convex potentials (RCPM).

As stated previously, our apporach optimised the semi-dual loss, but for comparison, we use the KL divergence and effective sample size (ESS), which present an alternative measure of transport quality. *Importantly*, many recent Riemannian diffusion models are designed primarily as generative models and therefore rely on different evaluation protocols; in particular, score-based diffusion models (Huang et al., 2022; De Bortoli et al., 2022) typically report negative log-likelihood on held-out data rather than the KL divergence of an explicit learned transport map. Our selected baselines, RCNF, Moser Flow, and RCPM, all admit tractable density evaluation via the change-of-variables formula, enabling principled comparison under the same metrics. Methods where KL divergence was not implemented by the authors, and not trivial to calculate (e.g. for RCNFs or Moser FLows) would have required ad hoc additions which could unfairly represent their approaches, or introduce unforeseen errors.

# G. Further Results

## G.1. Ablation and Sensitivity Analysis

We conduct an ablation study to understand the contribution of key design choices in our method. We consider four manifold settings, $\mathbb{S}^2$, $\mathbb{T}^2$, $\mathbb{S}^{10}$, and $\mathbb{T}^{10}$, and evaluate performance using KL divergence (lower is better).

**Summary of Findings.** Table 3 summarizes the relative performance of each ablation compared to the baseline. We highlight three key findings:

1. **FPS landmark selection consistently improves performance.** Using farthest point sampling (FPS) instead of random sampling for landmark selection yields the best or near-best KL divergence across all manifolds, with improvements of 27–52% on spheres and 34% on $\mathbb{T}^2$.

2. **LogSumExp initialization is important for torus manifolds.** Disabling LogSumExp initialization can cause failure on tori ($2.2\times$ worse on $\mathbb{T}^2$, $3.5\times$ worse on $\mathbb{T}^{10}$) while having minimal effect on spheres. This suggests the initialization is essential for handling the periodic structure of tori.

3. **Inner learning rate significantly impacts convergence.** A lower inner learning rate ($10^{-3}$ instead of $5 \times 10^{-2}$) severely degrades performance on tori, achieving $6.2\times$ worse KL on $\mathbb{T}^2$ and $12.3\times$ worse on $\mathbb{T}^{10}$.

**Detailed Analysis. Solver design.** The argmin solver's configuration has the largest impact on performance. The LogSumExp-based initialization provides a warm start that is important for torus manifolds, where the periodic boundary conditions make optimization more challenging. The Adam optimizer provides consistent benefits over vanilla SGD, particularly on higher-dimensional tori where the loss landscape is more complex.

**Network architecture.** FPS landmark selection provides the most consistent improvement, suggesting that well-distributed landmarks better capture the geometry of the manifold. Network width and depth have modest effects: smaller networks ([32, 32]) can match or exceed baseline performance on $\mathbb{S}^{10}$, indicating the baseline may be slightly overparameterized for simpler manifolds.

**Training hyperparameters.** A lower outer learning rate ($10^{-4}$ instead of $10^{-3}$) consistently hurts performance, likely due to insufficient training within the fixed step budget or due to a lack of ability to escape from poor solutions. Larger batch sizes (512) improve results on spheres but have less effect on tori. These results suggest larger batch sizes are not essential with increased dimensions.

**Recommendations.** Based on these ablations, we recommend:

- Use FPS landmark selection for improved geometry coverage

- Keep LogSumExp initialization enabled, especially for tori

- Use Adam optimizer for the inner loop

- Use batch size 256–512 for spheres

## G.2. Additional Experiments: Semi-Dual RCPM and More Complex Geometries

**Training RCPM with the semi-dual loss.** To separate the effect of the training objective from the effect of the transport parameterization, we additionally train RCPMs using the same Kantorovich semi-dual loss used by RNOT. This experiment is motivated by the concern that the degradation observed in Fig. 3 could be caused by the objective rather than by the discrete parameterization itself.

There is, however, an important tension in applying the semi-dual objective to RCPM. Training with the semi-dual loss is meaningful when the parameterization recovers a valid OT potential, whereas the RCPM parameterization of Cohen et al. (2021) recovers the hard minimum, and hence a true discrete $c$-concave potential, only in the limit $\gamma \to 0$. We therefore evaluate whether the same curse-of-dimensionality behavior persists when RCPM is trained with the semi-dual loss for small values of $\gamma$. Table 4 reports KL divergence for comparability with the main experiments and includes RNOT as a reference.

*Table 3.* Ablation study results showing KL divergence ($\downarrow$) across manifolds. Bold indicates best result; underline indicates $> 50\%$ degradation from baseline. All results averaged over 5 evaluation batches of 1024 samples.

| Configuration | $\mathbb{S}^2$ | $\mathbb{T}^2$ | $\mathbb{S}^{10}$ | $\mathbb{T}^{10}$ |
|---|---|---|---|---|
| Baseline | 0.040 | 0.125 | 0.048 | 1.07 |
| *Solver Ablations* | | | | |
| No LogSumExp init | 0.043 | 0.270 | 0.048 | 3.79 |
| Inner steps: 500 | 0.041 | 0.129 | 0.048 | 1.11 |
| Inner steps: 4000 | 0.040 | 0.122 | 0.048 | 1.06 |
| Inner LR: $10^{-3}$ | 0.055 | 0.775 | 0.048 | 13.13 |
| $\gamma_{\text{LSE}}$: 1.0 | 0.040 | 0.124 | 0.048 | 1.24 |
| $\gamma_{\text{LSE}}$: 0.01 | 0.040 | 0.127 | 0.048 | 1.07 |
| $\gamma_{\text{LSE}}$: 0.001 | 0.040 | 0.126 | 0.048 | 1.06 |
| No Adam (SGD) | 0.046 | 0.219 | 0.048 | 1.44 |
| *Architecture Ablations* | | | | |
| Landmarks: FPS | **0.019** | **0.083** | **0.035** | **0.03** |
| Landmarks: 32 | 0.036 | 0.099 | 0.054 | 3.71 |
| Landmarks: 64 | 0.041 | 0.203 | 0.048 | 1.47 |
| Landmarks: 256 | 0.036 | 0.036 | 0.053 | — |
| Hidden: [32, 32] | 0.046 | 0.202 | 0.040 | 1.16 |
| Hidden: [64, 64] | 0.034 | 0.170 | 0.044 | 1.23 |
| Hidden: [256, 256] | 0.037 | 0.104 | 0.045 | 1.04 |
| Hidden: [128, 128, 128] | 0.040 | 0.128 | 0.047 | 1.19 |
| *Training Ablations* | | | | |
| Outer LR: $10^{-4}$ | 0.067 | 0.184 | 0.060 | 1.59 |
| Batch size: 512 | 0.036 | 0.118 | **0.034** | 1.15 |
| Batch size: 128 | 0.072 | 0.183 | 0.040 | 1.16 |

The results show that RCPM performance degrades dramatically with dimension regardless of $\gamma$, while RNOT remains substantially more stable. This supports the theoretical prediction of Theorem 3.1 and Corollary 3.2: the curse of dimensionality is a structural property of the discrete parameterization class, not an artifact of the particular training objective.

**Experiments on** $\text{SO}(3)$ **and compactly supported** $\text{SE}(3)$. We also evaluate RNOT on more complex geometries beyond spheres and tori. On $\text{SO}(3)$, we consider the analogous transport problem from a uniform source to a wrapped normal target, consistent with our experiments on $\mathbb{S}^n$ and $\mathbb{T}^n$. On $\text{SE}(3)$, which is non-compact due to its translational component, we consider compactly supported measures: the source is a product of a uniform distribution on $\text{SO}(3)$ and a uniform distribution on a compact subset of $\mathbb{R}^3$, while the target is a product of a wrapped normal on $\text{SO}(3)$ and a truncated normal on the Euclidean component. This provides a practical intermediate regime between the compact-manifold theory developed in the paper and fully non-compact applications.

On $\text{SO}(3)$, RNOT and heavily regularized RCPM obtain comparable KL values, though RCPM achieves a higher ESS at the largest tested $\gamma$. However, as $\gamma$ decreases and the RCPM parameterization approaches the hard-minimum OT potential, its ESS collapses to nearly zero. This suggests that the apparent benefit of heavily regularized RCPM comes from smoothing rather than from approximating the true OT map. On compactly supported $\text{SE}(3)$, RNOT substantially outperforms RCPM, and RCPM becomes numerically unstable for smaller $\gamma$.

## G.3. Further Results for High-Dimensional Manifolds

*Table 4.* RCPM trained with the Kantorovich semi-dual loss. We report KL divergence across dimensions/manifolds and include RNOT for reference. RCPM continues to degrade with dimension for all tested values of $\gamma$, supporting the conclusion that the degradation is caused by the discrete parameterization rather than by the training objective.

| Manifold | Ours (RNOT) | RCPM ($\gamma = 0.01$) | RCPM ($\gamma = 0.005$) | RCPM ($\gamma = 0.001$) |
|---|---|---|---|---|
| $\mathbb{S}^{10}$ | $\mathbf{0.04 \pm 0.00}$ | $5.29 \pm 0.06$ | $5.34 \pm 0.07$ | $5.40 \pm 0.06$ |
| $\mathbb{S}^{20}$ | $\mathbf{0.02 \pm 0.00}$ | $8.81 \pm 0.03$ | $8.81 \pm 0.03$ | $8.82 \pm 0.03$ |
| $\mathbb{S}^{30}$ | $\mathbf{0.03 \pm 0.00}$ | $13.46 \pm 0.01$ | $13.47 \pm 0.02$ | $13.44 \pm 0.02$ |
| $\mathbb{T}^{10}$ | $\mathbf{0.92 \pm 0.02}$ | $46.15 \pm 0.41$ | $45.92 \pm 0.40$ | $45.59 \pm 0.38$ |
| $\mathbb{T}^{20}$ | $\mathbf{5.25 \pm 0.08}$ | $83.77 \pm 0.39$ | $83.73 \pm 0.40$ | $83.70 \pm 0.42$ |
| $\mathbb{T}^{30}$ | $\mathbf{3.21 \pm 0.10}$ | $126.66 \pm 0.44$ | $126.74 \pm 0.43$ | $126.75 \pm 0.44$ |

*Table 5.* Additional experiments on more complex geometries. We report KL divergence and ESS ratio. On $\mathrm{SO}(3)$, RNOT is competitive with heavily regularized RCPM, while smaller $\gamma$ values lead to near-zero ESS for RCPM. On compactly supported $\mathrm{SE}(3)$, RNOT substantially outperforms RCPM, and RCPM becomes numerically unstable for smaller $\gamma$.

| Manifold | Method | KL $\downarrow$ | ESS ratio $\uparrow$ |
|---|---|---|---|
|  | Ours (RNOT) | $2.96 \pm 0.01$ | $0.225$ |
|  | RCPM ($\gamma = 1.0$) | $2.86 \pm 0.03$ | $0.391$ |
|  | RCPM ($\gamma = 0.1$) | $3.29 \pm 0.10$ | $0.002$ |
| $\mathrm{SO}(3)$ | RCPM ($\gamma = 0.05$) | $3.54 \pm 0.06$ | $0.006$ |
|  | RCPM ($\gamma = 0.01$) | $2.83 \pm 0.10$ | $0.003$ |
|  | RCPM ($\gamma = 0.005$) | $3.72 \pm 0.10$ | $0.004$ |
|  | RCPM ($\gamma = 0.001$) | $5.90 \pm 0.06$ | $0.004$ |
|  | Ours (RNOT) | $2.50 \pm 0.01$ | $0.683$ |
| $\mathrm{SE}(3)$ | RCPM ($\gamma = 1.0$) | $14.38 \pm 0.09$ | $0.007$ |
|  | RCPM ($\gamma < 1$) | Numerically unstable | $-$ |

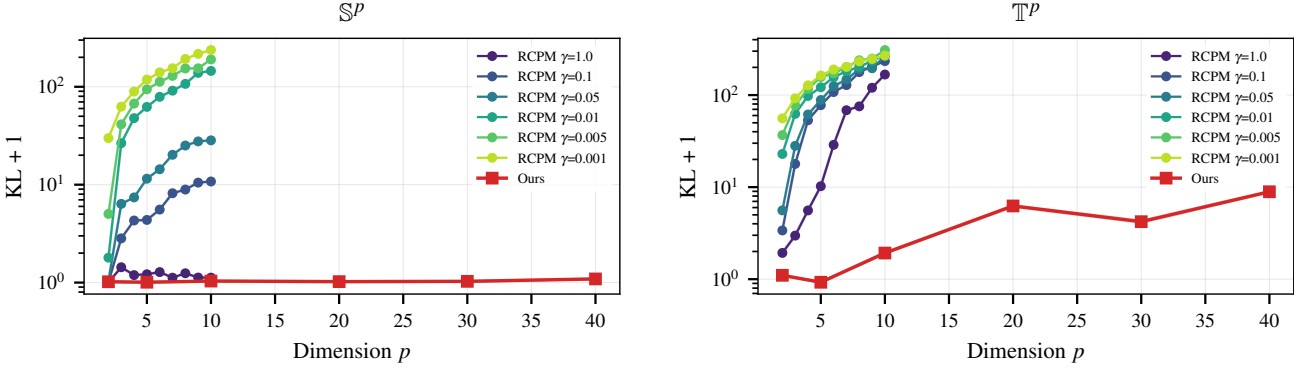

*Figure 5.* KL divergence scaling with dimension on high-dimensional manifolds. We compare our approach against RCPMs with varying regularization parameters $\gamma \in \{0.001, 0.005, 0.01, 0.05, 0.1, 1.0\}$. RCPM results are shown for $p \in \{2, \ldots, 10\}$ only due to computational intractability at higher dimensions, while our method extends to $p = 40$. Our method achieves consistently lower KL divergence across all tested dimensions and scales favorably to high-dimensional settings, empirically supporting our theoretical results.

# H. Proofs

## H.1. Proof of Theorem 3.1

It is convenient to recall the following definition of the quantization error from Iacobelli (2016).

**Definition H.1** (Quantization error; Def. 1.1 in Iacobelli (2016)). Let $(\mathcal{M}, d)$ be a complete Riemannian manifold, $\rho \in \mathcal{P}(\mathcal{M})$, $r \geq 1$, and $N \in \mathbb{N}$. The $N$-th quantization error of order $r$ is

$$V_{N,r}(\rho) := \inf_{\alpha \subset \mathcal{M}: \, |\alpha| \leq N} \int_{\mathcal{M}} \min_{a \in \alpha} d(a, y)^r \, \mathrm{d}\rho(y).$$

A key result we will use in proving Theorem 3.1 is Theorem 1.4 of Iacobelli (2016) which we now state here for convenience. To state in detail our main result we need to introduce some notation from Iacobelli (2016): given a point $x_0 \in \mathcal{M}$, we can consider polar coordinates $(\rho, \theta)$ on $T_{x_0}\mathcal{M} \simeq \mathbb{R}^p$ induced by the constant metric $g_{x_0}$, where $\theta$ denotes a vector on the unit sphere $\mathbb{S}^{p-1}$. Then, we can define the following quantity that measures the size of the differential of the exponential map when restricted to a sphere $\mathbb{S}^{p-1}_\rho \subset T_{x_0}\mathcal{M}$ of radius $\rho$:

$$A_{x_0}(\rho) := \sup_{v \in \mathbb{S}^{p-1}_\rho, \, w \in T_v \mathbb{S}^{p-1}_\rho, \, |w|_{x_0} = \rho} \| d_v \exp_{x_0}[w] \|_{\exp_{x_0}(v)}.$$

We are now ready to present Theorem 1.4 of Iacobelli (2016).

**Theorem H.2** (Theorem 1.4 in (Iacobelli, 2016)). *Let $(\mathcal{M}, g)$ be a $p$-dimensional complete Riemannian manifold without boundary, and let $\mu = h\mathrm{dvol} + \mu^s$ be a probability measure on $\mathcal{M}$. Assume there exist a point $x_0 \in \mathcal{M}$ and $\delta > 0$ such that*

$$\int_{\mathcal{M}} d(x, x_0)^{r+\delta} \mathrm{d}\mu(x) + \int_{\mathcal{M}} A_{x_0}(d(x, x_0))^r \mathrm{d}\mu(x) < \infty. \tag{22}$$

*Then*

$$\lim_{N \to \infty} N^{r/p} V_{N,r}(\mu) = Q \left( \int_{\mathcal{M}} h^{p/(p+r)} \mathrm{d}x \right)^{(p+r)/p}.$$

*for a positive constant $Q$.*

Our next result shows that Theorem H.2 is immediately applicable our setting (i.e. when $(\mathcal{M}, g)$ is a compact Riemannian manifold) as the sufficient conditions are always verified.

**Lemma H.3** (Iacobelli's integrability condition is automatic on compact manifolds). *Let $(\mathcal{M}, g)$ be a compact Riemannian manifold (hence complete and without boundary), let $\mu \in \mathcal{P}(\mathcal{M})$, and fix $x_0 \in \mathcal{M}$, $r \geq 1$, and $\delta > 0$. Then the integrability condition (22) holds, i.e.*

$$\int_{\mathcal{M}} d(x, x_0)^{r+\delta} \, \mathrm{d}\mu(x) \; + \; \int_{\mathcal{M}} A_{x_0}(d(x, x_0))^r \, \mathrm{d}\mu(x) < \infty.$$

The proof of Lemma H.3 is postponed to Appendix H.7.

**Lemma H.4** (From $V_{m,2}$ to $W_2$). *Let $\rho \in \mathcal{P}_2(\mathcal{M})$ and let $\eta$ be supported on at most $m$ points. Then*

$$V_{m,2}(\rho) \; \leq \; W_2(\rho, \eta)^2$$

The proof of Lemma H.4 is postponed to Appendix H.8.

**Lemma H.5** (Wasserstein stability under a common source measure). *Let $\mu \in \mathcal{P}(\mathcal{M})$ and let $t, s : \mathcal{M} \to \mathcal{M}$ be measurable. Set $\nu_t := t_{\#}\mu$ and $\nu_s := s_{\#}\mu$. Then $\nu_t, \nu_s \in \mathcal{P}_2(\mathcal{M})$ and*

$$W_2(\nu_t, \nu_s) \; \leq \; \mathrm{RMSE}_\mu(t, s). \tag{23}$$

The proof of Lemma H.5 is postponed to Appendix H.9.

*Proof of Theorem 3.1.* Let $p := \dim(\mathcal{M})$ and define $\nu := (T_\star)_{\#}\mu$.

**Step 1: Lower bound on $V_{m,2}(\nu)$.** Since $\nu \ll \mathrm{vol}_{\mathcal{M}}$, write $\nu = h \, \mathrm{dvol}_{\mathcal{M}}$ with $h \geq 0$ and $\int_{\mathcal{M}} h \, \mathrm{dvol}_{\mathcal{M}} = 1$. Apply Theorem H.2 with $r = 2$ and $\rho = \nu$. Because $\mathcal{M}$ is compact, Lemma H.3 implies that the integrability condition (22) holds, so the theorem yields the asymptotic formula

$$\lim_{m \to \infty} m^{2/p} V_{m,2}(\nu) = Q \left( \int_{\mathcal{M}} h^{p/(p+2)} \, \mathrm{dvol}_{\mathcal{M}} \right)^{(p+2)/p} =: C_{\text{quant}}.$$

Moreover $C_{\text{quant}} > 0$: indeed, since $\nu$ is a probability measure, $h > 0$ on a set of positive Riemannian volume, hence $\int_{\mathcal{M}} h^{p/(p+2)} \, \mathrm{dvol}_{\mathcal{M}} > 0$.

By the definition of the limit, there exists $m_0 \in \mathbb{N}$ such that for all $m \geq m_0$,

$$m^{2/p} V_{m,2}(\nu) \geq \frac{C_{\text{quant}}}{2}, \qquad \text{equivalently} \qquad V_{m,2}(\nu) \geq \frac{C_{\text{quant}}}{2} m^{-2/p}.$$

For the finitely many indices $1 \leq m < m_0$, define

$$c_\star := \min_{1 \leq m < m_0} m^{2/p} V_{m,2}(\nu).$$

Each term in this minimum is strictly positive: since $\nu \ll \mathrm{vol}_{\mathcal{M}}$, it cannot be supported on finitely many points, hence $V_{m,2}(\nu) > 0$ for every fixed $m$. Therefore $c_\star > 0$. Finally set

$$C_0 := \min\left\{ \frac{C_{\text{quant}}}{2}, c_\star \right\} > 0.$$

Then for every $m \in \mathbb{N}$,

$$V_{m,2}(\nu) \geq C_0 \, m^{-2/p}, \tag{24}$$

which is the desired uniform lower bound.

**Step 2: Discrete-output maps imply a Wasserstein lower bound.** Fix $m \in \mathbb{N}$ and let $T \in \mathsf{D}_m(\mu)$. Set $\nu_T := T_{\#}\mu$, which is supported on at most $m$ points by definition of $\mathsf{D}_m(\mu)$. Applying Lemma H.4 gives

$$V_{m,2}(\nu) \leq W_2(\nu, \nu_T)^2.$$

Together with (24) this yields

$$W_2(\nu, \nu_T) \geq \sqrt{C_0} \, m^{-1/p}. \tag{25}$$

**Step 3: From Wasserstein to map RMSE.** Applying Lemma H.5 with $t := T$ and $s := T_\star$, we obtain

$$W_2(\nu_T, \nu) \leq \mathrm{RMSE}_\mu(T, T_\star).$$

Combining with (25) gives, for every $T \in \mathsf{D}_m(\mu)$,

$$\mathrm{RMSE}_\mu(T, T_\star) \geq \sqrt{C_0} \, m^{-1/p}.$$

Taking the infimum over $T \in \mathsf{D}_m(\mu)$ proves (6) with $C := \sqrt{C_0}$. The condition $\mathrm{RMSE}_\mu(T, T_\star) \leq \delta$ implies $m \geq (C/\delta)^p$ by rearranging. $\qquad \square$

### H.2. Proof of Corollary 3.2

*Proof of Corollary 3.2.* Fix $m \in \mathbb{N}$ and let $\phi \in \mathcal{C}_m(\mathcal{M})$. By definition of $\mathcal{C}_m(\mathcal{M})$, there exist sites $\{y_i\}_{i \in [m]} \subset \mathcal{M}$ and scalars $\{\alpha_i\}_{i \in [m]} \subset \mathbb{R}$ such that, for all $x \in \mathcal{M}$,

$$\phi(x) = \min_{i \in [m]} \left( \tfrac{1}{2} d(x, y_i)^2 + \alpha_i \right).$$

Set

$$f_i(x) := \tfrac{1}{2} d(x, y_i)^2 + \alpha_i, \qquad i \in [m].$$

**Step 1: Defining the cells and the tie set.** For each $i \in [m]$, define the (strict) cell

$$A_i := \{x \in \mathcal{M} : \ f_i(x) < f_j(x) \text{ for all } j \neq i\},$$

and define the tie set

$$B := \Big\{x \in \mathcal{M} : \ \exists\, i \neq j \text{ with } f_i(x) = f_j(x) = \phi(x)\Big\}.$$

Lemma 1 in (McCann, 2001) shows that $x \mapsto \frac{1}{2}d(x, y_i)^2$ is Lipschitz on $\mathcal{M}$ for each fixed $y_i$, hence each $f_i$ is Lipschitz (and therefore continuous). For $i \neq j$ define $g_{ij} := f_i - f_j$; then $g_{ij}$ is continuous, and

$$A_i = \bigcap_{j \neq i}\{x \in \mathcal{M} : \ g_{ij}(x) < 0\} = \bigcap_{j \neq i} g_{ij}^{-1}((-\infty, 0))$$

is open as a finite intersection of preimages of an open set under continuous maps. Every point $x \in \mathcal{M}$ either has a unique minimizer or a tie, hence

$$\mathcal{M} = \Big(\bigcup_{i=1}^{m} A_i\Big) \cup B, \qquad A_i \cap A_j = \varnothing \ (i \neq j).$$

**Step 2: Differentiability of $\phi$ almost everywhere.** Since $\phi$ is the minimum of finitely many Lipschitz functions, it is Lipschitz on $\mathcal{M}$. By Rademacher's theorem on Riemannian manifolds (Burago et al., 2001, Theorem 5.5.7), there exists a set $\mathcal{U} \subset \mathcal{M}$ with $\mathrm{vol}_{\mathcal{M}}(\mathcal{U}) = 0$ such that $\phi$ is differentiable on $\mathcal{M} \setminus \mathcal{U}$.

**Step 3: Identification of $\nabla \phi$ on each $A_i$.** Fix $i \in [m]$ and $x \in A_i \setminus \mathcal{U}$. By definition of $A_i$ we have $f_i(x) < f_j(x)$ for all $j \neq i$. By continuity, there exists $r > 0$ such that the strict inequalities persist on $B(x, r)$:

$$f_i(z) < f_j(z) \quad \forall j \neq i, \ \forall z \in B(x, r).$$

Thus $\phi(z) = f_i(z)$ for all $z \in B(x, r)$, so $\phi$ coincides locally with the smooth function $f_i$ and

$$\nabla \phi(x) = \nabla f_i(x) = \nabla\Big(\tfrac{1}{2}d(x, y_i)^2\Big).$$

Let $\mathrm{Cut}(y_i)$ denote the cut locus of $y_i$. By (8), if $x \in \mathcal{M} \setminus (\mathrm{Cut}(y_i) \cup \{y_i\})$, then $y_i \notin \mathrm{Cut}(x)$ and

$$\nabla\Big(\tfrac{1}{2}d(x, y_i)^2\Big) = -\log_x(y_i),$$

hence for every $x \in A_i \setminus \mathcal{U}$ with $x \notin \mathrm{Cut}(y_i) \cup \{y_i\}$ we obtain

$$\nabla \phi(x) = -\log_x(y_i).$$

Therefore, for such $x$,

$$T_\phi(x) := \exp_x\big(-\nabla \phi(x)\big) = \exp_x\big(\log_x(y_i)\big) = y_i.$$

Let

$$\mathrm{Cut} := \Big(\bigcup_{i=1}^{m} \mathrm{Cut}(y_i)\Big) \cup \{y_1, \dots, y_m\}.$$

The cut locus of a point has zero Riemannian volume, hence $\mathrm{vol}_{\mathcal{M}}(\mathrm{Cut}) = 0$ as a finite union.

**Step 4: The tie set $B$ has zero Riemannian volume.** For each $i \neq j$ define the hypersurface candidate

$$H_{ij} := \{x \in \mathcal{M} : \ f_i(x) = f_j(x)\}.$$

Then $B \subseteq \bigcup_{1 \leq i < j \leq m} H_{ij}$, so it suffices to show $\mathrm{vol}_{\mathcal{M}}(H_{ij}) = 0$. Fix $i \neq j$ and define

$$h_{ij}(x) := f_i(x) - f_j(x) = \tfrac{1}{2}\big(d(x, y_i)^2 - d(x, y_j)^2\big) + (\alpha_i - \alpha_j).$$

Let

$$S_{ij} := \mathrm{Cut}(y_i) \cup \mathrm{Cut}(y_j) \cup \{y_i, y_j\}.$$

On $\mathcal{M} \setminus S_{ij}$, both squared-distance functions are smooth, so $h_{ij}$ is smooth there and

$$\nabla h_{ij}(x) = -\log_x(y_i) + \log_x(y_j) \qquad (x \in \mathcal{M} \setminus S_{ij}).$$

We claim $\nabla h_{ij}(x) \neq 0$ for all $x \in H_{ij} \setminus S_{ij}$. Indeed, if $\nabla h_{ij}(x) = 0$ at some $x \notin S_{ij}$ then $\log_x(y_i) = \log_x(y_j)$, and applying $\exp_x$ gives $y_i = y_j$, contradicting $i \neq j$. Hence $0$ is a regular value of $h_{ij} : \mathcal{M} \setminus S_{ij} \to \mathbb{R}$, so $H_{ij} \setminus S_{ij} = h_{ij}^{-1}(0)$ is an embedded $(p-1)$-dimensional submanifold of $\mathcal{M} \setminus S_{ij}$, and in particular has zero $p$-dimensional Riemannian volume. Since $\mathrm{vol}_{\mathcal{M}}(S_{ij}) = 0$, we conclude $\mathrm{vol}_{\mathcal{M}}(H_{ij}) = 0$. By finiteness of the union,

$$\mathrm{vol}_{\mathcal{M}}(B) = 0.$$

**Step 5: Discrete-output property $\mu$-a.e.**   Define the exceptional set

$$N := \mathcal{U} \cup \mathrm{Cut} \cup B.$$

We have $\mathrm{vol}_{\mathcal{M}}(N) = 0$. Under the assumptions of Theorem 3.1 we have $\mu \ll \mathrm{vol}_{\mathcal{M}}$, hence $\mu(N) = 0$. On $\mathcal{M} \setminus N$, every $x$ lies in exactly one cell $A_i$ and satisfies $T_\phi(x) = y_i$; thus

$$T_\phi(x) \in \{y_1, \ldots, y_m\} \qquad \text{for } \mu\text{-a.e. } x \in \mathcal{M}.$$

Equivalently, $T_\phi$ takes at most $m$ distinct values on a set of full $\mu$-measure (and $T_{\phi\#}\mu$ is supported on at most $m$ points). Hence $T_\phi \in \mathsf{D}_m(\mu)$.

**Step 6: Curse of dimensionality lower bound.**   Since $\{T_\phi : \phi \in \mathcal{C}_m(\mathcal{M})\} \subseteq \mathsf{D}_m(\mu)$, we have

$$\inf_{\phi \in \mathcal{C}_m(\mathcal{M})} \mathrm{RMSE}_\mu(T_\phi, T_\star) \;\geq\; \inf_{T \in \mathsf{D}_m(\mu)} \mathrm{RMSE}_\mu(T, T_\star).$$

Applying Theorem 3.1 yields a constant $C > 0$ such that for all $m \in \mathbb{N}$,

$$\inf_{\phi \in \mathcal{C}_m(\mathcal{M})} \mathrm{RMSE}_\mu(T_\phi, T_\star) \;\geq\; C\, m^{-1/p},$$

which is (7). The sample-complexity statement follows immediately: if $\mathrm{RMSE}_\mu(T_\phi, T_\star) \leq \delta$, then necessarily $m \geq (C/\delta)^p$. $\qquad \square$

## H.3. Proof of Theorem 4.1

Before proving Theorem 4.1 we show the following convergence result.

**Lemma H.6** (Convergence of gradients along $c$-transform approximants). *Let $\mu \in \mathcal{P}(\mathcal{M})$ satisfy $\mu \ll \mathrm{vol}_{\mathcal{M}}$. Let $\varphi : \mathcal{M} \to \mathbb{R}^n$ be a feature map satisfying Assumption 2.2 and assume that $\mathcal{F}$ is dense in $C(\mathbb{R}^n, \mathbb{R})$ in the ucc topology. Then for every $\phi \in \Psi_c(\mathcal{M})$ there exists a sequence $\psi_k \in \varphi^*\mathcal{F}$ and a measurable set $N_\star \subset \mathcal{M}$ with $\mu(N_\star) = 1$ such that, with $\phi_k := \psi_k^c$,*

$$\|\phi_k - \phi\|_\infty \to 0,$$

*and for every $x \in N_\star$ the functions $\phi$ and $\phi_k$ are differentiable at $x$ and*

$$\nabla \phi_k(x) \to \nabla \phi(x).$$

The proof of Lemma H.6 is postponed to the Appendix H.10. We are now ready to prove Theorem 4.1.

*Proof of Theorem 4.1.* By Theorem B.3, since $\mu \ll \mathrm{vol}_{\mathcal{M}}$ and $c(x, y) = \frac{1}{2}d(x, y)^2$, there exists a $c$-concave potential $\phi \in \Psi_c(\mathcal{M})$ such that the optimal transport map $T_\star$ from $\mu$ to $\nu$ is given $\mu$-a.e. by

$$T_\star(x) = \exp_x\big(-\nabla \phi(x)\big).$$

**Step 1: Construct approximating potentials with a.e. gradient convergence.** Since $\mathcal{F}$ is dense in $C(\mathbb{R}^n, \mathbb{R})$ in the ucc topology and $\varphi$ satisfies Assumption 2.2, the pullback class $\varphi^* \mathcal{F}$ is dense in $C(\mathcal{M}, \mathbb{R})$ in the ucc topology. Hence the assumptions of Lemma H.6 are satisfied. Applying Lemma H.6 to the above $\phi$ yields a sequence $\psi_k \in \varphi^* \mathcal{F}$ and the associated $c$-transforms

$$\phi_k := \psi_k^c \in \mathfrak{C}(\varphi^* \mathcal{F})$$

such that

$$\|\phi_k - \phi\|_\infty \to 0 \qquad \text{and} \qquad \nabla \phi_k(x) \to \nabla \phi(x) \text{ for } \mu\text{-a.e. } x \in \mathcal{M}.$$

Let $N_\star \subset \mathcal{M}$ be a Borel set with $\mu(N_\star) = 1$ on which $\phi$ and all $\phi_k$ are differentiable and $\nabla \phi_k(x) \to \nabla \phi(x)$ for every $x \in N_\star$.

**Step 2: Define everywhere-defined maps $T_k$ and $T_\star$.** Fix an arbitrary point $y_0 \in \mathcal{M}$ and define maps $T_k, T_\star : \mathcal{M} \to \mathcal{M}$ by

$$T_k(x) := \begin{cases} \exp_x(-\nabla \phi_k(x)), & x \in N_\star, \\ y_0, & x \notin N_\star, \end{cases} \qquad T_\star(x) := \begin{cases} \exp_x(-\nabla \phi(x)), & x \in N_\star, \\ y_0, & x \notin N_\star. \end{cases}$$

This modification on $N_\star^c$ is immaterial since $\mu(N_\star^c) = 0$, but it ensures that $T_k$ and $T_\star$ are defined everywhere (hence can be viewed as random variables on $(\mathcal{M}, \mu)$).

**Step 3: Pointwise convergence on $N_\star$.** Fix $x \in N_\star$. Since $\mathcal{M}$ is compact, it is complete, and by the Hopf–Rinow Theorem (see Theorem A.2) the exponential map $\exp_x : T_x\mathcal{M} \to \mathcal{M}$ is defined on all of $T_x\mathcal{M}$ and is smooth, hence continuous, as a function of the tangent vector. Therefore, from $\nabla \phi_k(x) \to \nabla \phi(x)$ in $T_x\mathcal{M}$, we obtain

$$T_k(x) = \exp_x(-\nabla \phi_k(x)) \longrightarrow \exp_x(-\nabla \phi(x)) = T_\star(x).$$

Thus $T_k(x) \to T_\star(x)$ for every $x \in N_\star$, and since $\mu(N_\star) = 1$ we have

$$T_k(x) \to T_\star(x) \qquad \text{for } \mu\text{-a.e. } x \in \mathcal{M}.$$

**Step 4: Almost sure convergence implies convergence in probability.** Recall the standard result in probability: if $X_k \to X$ almost surely under a probability measure $\mu$, then $X_k \to X$ in probability. Apply this with the random variables $X_k(x) := T_k(x)$ and $X(x) := T_\star(x)$ under the law $x \sim \mu$. Since $T_k \to T_\star$ $\mu$-a.e., it follows that

$$T_k \to T_\star \qquad \text{in probability under } \mu.$$

This proves that the sequence $(T_k)$ universally approximates the optimal transport map $T_\star$ in probability under $\mu$, as claimed. $\square$

## H.4. Proof of Theorem 5.1

We study neural-network approximation of functions on compact sets. The approximation results we rely on from Yarotsky & Zhevnerchuk (2020) are stated for the network model introduced by Yarotsky (2017), where a network is described as a feedforward directed acyclic graph (DAG). This graph-based viewpoint is slightly more general than the traditional fully connected layered template: it naturally allows sparse connectivity and skip connections, and it measures complexity directly in terms of the number of scalar trainable parameters carried by the graph. To minimize notational friction and to quote Yarotsky & Zhevnerchuk (2020) verbatim, we adopt this convention in the next definitions.

**Definition H.7** (Feedforward neural network architecture). Fix an input dimension $n \in \mathbb{N}$ and an activation $\sigma : \mathbb{R} \to \mathbb{R}$. A *feedforward network architecture* is a directed acyclic graph whose vertices (units) are partitioned into *layers* $V_0, \ldots, V_{L-1}$ with the following structure:

- $V_0 = \{1, \ldots, n\}$ is the *input layer*; for $x \in \mathbb{R}^n$ we set $z_i = x_i$ for each $i \in V_0$.

- $V_{L-1} = \{\text{out}\}$ is the *output layer* consisting of a single unit.

- For each $\ell \in \{1, \ldots, L-1\}$ and each unit $v \in V_\ell$, the set of incoming neighbors satisfies

$$\text{In}(v) \subseteq \bigcup_{j=0}^{\ell-1} V_j,$$

so that $\text{In}(v)$ may include units from *any preceding layer* (skip connections are allowed), and the architecture contains directed edges $(u \to v)$ for all $u \in \text{In}(v)$.

Given weights and biases $(w_{vu})_{(u \to v) \in E}$ and $(b_v)_{v \in \cup_{\ell=1}^{L-1} V_\ell}$, the associated realization $\hat{f} : \mathbb{R}^n \to \mathbb{R}$ is defined by the recursion

$$z_v = \sigma\left(\sum_{u \in \text{In}(v)} w_{vu}\, z_u + b_v\right), \qquad v \in \bigcup_{\ell=1}^{L-2} V_\ell,$$

and the output rule (no activation)

$$\hat{f}(x) = z_{\text{out}} = \sum_{u \in \text{In(out)}} w_{\text{out},u}\, z_u + b_{\text{out}}.$$

The previous definition specifies the *architecture* (the layered DAG) and the corresponding computation rule once parameters are assigned. We now fix the complexity measures used in Yarotsky (2017); Yarotsky & Zhevnerchuk (2020), namely the depth (number of layers) and the size (number of scalar parameters) of the underlying graph.

**Definition H.8** (Depth and number of weights). Let an architecture be given as in Definition H.7. Its *depth* is the number of layers $L$ (in particular, a network with one hidden layer has depth 3). Let $E$ denote the set of directed edges (connections), and define the number of computation units

$$U := \sum_{\ell=1}^{L-1} |V_\ell|$$

(the hidden units plus the output unit). The *number of weights* (network size) is

$$W := |E| + U,$$

i.e. one scalar parameter per connection and one scalar bias per computation unit. We write $\mathcal{NN}_{n,L,W}^\sigma$ for the set of all functions $\hat{f} : \mathbb{R}^n \to \mathbb{R}$ representable by architectures of depth at most $L$ and with at most $W$ weights (in the above sense).

The key point of Definition H.8 is that complexity is measured intrinsically by the graph: each directed edge contributes one scalar weight and each computation unit contributes one scalar bias, so that $W = |E| + U$. This contrasts with the traditional dense matrix parameterizations, where one counts all entries of each layer matrix, including coefficients corresponding to edges that are structurally absent in a sparse graph. Finally, note that the depth in this convention counts the input and output layers as well; hence "$L = 3$" corresponds to one hidden layer. In the sequel, whenever we invoke Yarotsky & Zhevnerchuk (2020), the symbols $L$ and $W$ refer to this convention.

Our analysis is restricted to *piecewise linear* activation functions. That is, we consider $\sigma \in C(\mathbb{R}, \mathbb{R})$ for which there exists $B \in \mathbb{N}$ and pairwise distinct breakpoints $x_1, \ldots, x_B \in \mathbb{R}$ such that every point $x \in \mathbb{R} - \{x_b\}_{b=1}^B$ lies in some open interval on which $\sigma$ is affine, while no such affine neighborhood exists at any breakpoint $x_b$ (for $b = 1, \ldots, B$). In particular, if $\sigma$ is piecewise linear and non-affine, then necessarily $B \geq 1$. This class includes the ReLU activation function of Fukushima (1969), the leaky-ReLU activation function of Maas et al. (2013), the pReLU activation function of He et al. (2015), and commonly used piecewise linear approximations of the Heaviside function (implemented for example in Abadi et al. (2015)).

A key ingredient in proving Theorem 5.1 is the following direct specialization of Yarotsky & Zhevnerchuk (2020, Thm. 3.3) (the "deep discontinuous phase") to the unit ball in the Hölder space $\mathcal{H}_{r,n}$ (see (16)). We state it in an $\varepsilon$-form convenient for learning theory.

**Theorem H.9** (Instantiation of Theorem 3.3 in Yarotsky & Zhevnerchuk (2020) for piecewise-linear activations). *Fix an input dimension $n \in \mathbb{N}$ and a smoothness level $r > 0$. Let $\beta$ satisfy*

$$\frac{r}{n} < \beta \leq \frac{2r}{n}.$$

*Let $\sigma : \mathbb{R} \to \mathbb{R}$ be any non-affine piecewise-linear activation function. Then there exist constants $\widetilde{C}_{r,n,\beta,\sigma} > 0$ and $C_{r,n,\sigma} > 0$ such that for every $\varepsilon \in (0,1)$ and every $f \in \mathcal{H}_{r,n}$, there exist integers $W_\varepsilon \geq 1$ and $L_\varepsilon \geq 2$ and a neural network $\hat{f}_\varepsilon \in \mathcal{NN}^\sigma_{n,L_\varepsilon,W_\varepsilon}$ such that*

$$\|f - \hat{f}_\varepsilon\|_\infty \leq \varepsilon,$$

*and*

$$W_\varepsilon \leq \left( \frac{\widetilde{C}_{r,n,\beta,\sigma}}{\varepsilon} \right)^{1/\beta}, \qquad L_\varepsilon \leq C_{r,n,\sigma}\, W_\varepsilon^{\beta n/r - 1}.$$

The proof of Theorem H.9 is postponed to Appendix H.12.

We now leverage Theorem H.9 to prove Theorem 5.1. For readability, we decompose the proof into several lemmas.

*Proof strategy.* The proof proceeds by reducing approximation on the $p$-dimensional manifold $\mathcal{M}$ to approximation on a $2p$-dimensional Euclidean cube. The reduction has three conceptual parts: (i) embed $\mathcal{M}$ into $[0,1]^{2p}$ by a fixed smooth feature map $\varphi_\star$; (ii) transfer $f$ to the embedded copy $K' = \varphi_\star(\mathcal{M})$ and then extend it to a globally defined Hölder function on the cube; (iii) apply the Euclidean approximation theorem on $[0,1]^{2p}$ and pull the resulting network back to $\mathcal{M}$ by composing with $\varphi_\star$. We now formalize each step in separate lemmas.

**Lemma H.10** (Embedding and normalization into the open cube). *Let $\mathcal{M}$ be a compact smooth manifold of dimension $p$. Then there exist*

- *a smooth embedding $\Phi : \mathcal{M} \to \mathbb{R}^{2p}$,*

- *an invertible affine map $A : \mathbb{R}^{2p} \to \mathbb{R}^{2p}$,*

*such that the feature map $\varphi_\star := A \circ \Phi$ is a smooth embedding with image $K' := \varphi_\star(\mathcal{M}) \subset (0,1)^{2p}$, and $K'$ is compact.*

*Proof.* By the strong Whitney's embedding theorem (see (Lee, 2012, Theorem 6.20)) there exists a smooth embedding $\Phi : \mathcal{M} \to \mathbb{R}^{2p}$. Since $\Phi$ is smooth it is continuous, and since $\mathcal{M}$ is compact, the image

$$K := \Phi(\mathcal{M}) \subset \mathbb{R}^{2p}$$

is compact since it is continuous image of a compact set. In particular $K$ is bounded (e.g. by Heine–Borel Theorem in $\mathbb{R}^{2p}$). Fix $c \in \mathbb{R}^{2p}$ and define

$$R := \sup_{z \in K} \|z - c\|_2 < \infty.$$

Equivalently, since $z \mapsto \|z - c\|_2$ is continuous and $K$ is compact, the supremum is actually a maximum by the Extreme Value Theorem, so $R = \max_{z \in K} \|z - c\|_2$.

If $R = 0$, then $\|z - c\|_2 = 0$ for every $z \in K$, hence $K = \{c\}$. Since $\Phi$ is an embedding, in particular it is injective, so $\Phi(\mathcal{M}) = \{c\}$ implies $\mathcal{M}$ is a singleton, and the statement is trivial. Assume $R > 0$.

Define the affine map $B : \mathbb{R}^{2p} \to \mathbb{R}^{2p}$ by

$$B(z) := \frac{z - c}{2R} + \frac{1}{2}\mathbf{1}.$$

Here the linear part is $\frac{1}{2R}I$, hence $B$ is invertible with inverse $B^{-1}(u) = 2R\left(u - \frac{1}{2}\mathbf{1}\right) + c$. Then for every $z \in K$ we have $\|z - c\|_\infty \leq \|z - c\|_2 \leq R$. Thus $|(z - c)_j| \leq R$ for each coordinate $j$, so

$$-\frac{1}{2} \leq \frac{(z - c)_j}{2R} \leq \frac{1}{2} \quad \Longrightarrow \quad 0 \leq \frac{(z - c)_j}{2R} + \frac{1}{2} \leq 1,$$

i.e. $B(z) \in [0,1]^{2p}$. Hence

$$B(K) \subset [0,1]^{2p}.$$

To obtain strict inclusion into the open cube, fix any $\eta \in (0, 1/2)$ and define the affine contraction $S_\eta : \mathbb{R}^{2p} \to \mathbb{R}^{2p}$ by

$$S_\eta(u) := (1 - 2\eta)u + \eta\mathbf{1}.$$

Since $1 - 2\eta > 0$, its linear part $(1 - 2\eta)I$ is invertible, hence $S_\eta$ is invertible with inverse

$$S_\eta^{-1}(v) = \frac{v - \eta\mathbf{1}}{1 - 2\eta}.$$

Moreover, for every $u \in [0,1]^{2p}$ and each coordinate $j = 1, \ldots, 2p$,

$$0 \le u_j \le 1 \quad \implies \quad \eta \le (1 - 2\eta)u_j + \eta \le 1 - \eta,$$

so $S_\eta(u) \in [\eta, 1 - \eta]^{2p} \subset (0,1)^{2p}$. In particular,

$$S_\eta([0,1]^{2p}) \subset [\eta, 1 - \eta]^{2p} \subset (0,1)^{2p}.$$

Now set

$$A := S_\eta \circ B, \qquad \varphi_\star := A \circ \Phi, \qquad K' := \varphi_\star(\mathcal{M}) = A(K).$$

Then $A$ is an invertible affine map (composition of invertible affine maps). Since $A$ is a diffeomorphism of $\mathbb{R}^{2p}$ and $\Phi$ is an embedding, $\varphi_\star = A \circ \Phi$ is again a smooth embedding (composition of an embedding with a diffeomorphism preserves injectivity, immersion, and the homeomorphism onto the image). Finally,

$$K' = A(K) = S_\eta(B(K)) \subset S_\eta([0,1]^{2p}) \subset [\eta, 1 - \eta]^{2p} \subset (0,1)^{2p}.$$

Moreover $K'$ is compact as the continuous image of the compact set $\mathcal{M}$ (equivalently, $K' = A(K)$ is compact as the continuous image of the compact set $K$). $\qquad\square$

*Step 1: Fix a Euclidean model of $\mathcal{M}$.* Lemma H.10 constructs a smooth embedding $\varphi_\star : \mathcal{M} \to [0,1]^{2p}$, so that all subsequent approximations can be carried out on the fixed cube $[0,1]^{2p}$. We write $K' := \varphi_\star(\mathcal{M})$ for the embedded copy. The next lemma transfers the target function $f$ to $K'$ and compares Hölder norms.

**Lemma H.11** (Pullback and Hölder norm control). *Let $\varphi_\star : \mathcal{M} \to K'$ be as in Lemma H.10. Given $f \in C^{k,1}(\mathcal{M})$, define*

$$h : K' \to \mathbb{R}, \qquad h(u) := f(\varphi_\star^{-1}(u)).$$

*Then $h \in C^{k,1}(K')$. Moreover, there exists a constant $C_{\mathrm{pb}} = C_{\mathrm{pb}}(\mathcal{M}, \varphi_\star, k)$ such that*

$$\|h\|_{C^{k,1}(K')} \le C_{\mathrm{pb}}\|f\|_{C^{k,1}(\mathcal{M})}.$$

*In particular, if $\|f\|_{C^{k,1}(\mathcal{M})} \le 1$ then $\|h\|_{C^{k,1}(K')} \le C_{\mathrm{pb}}$.*

*Proof.* Since $\varphi_\star$ is a smooth embedding, it induces a unique smooth structure on the image $K' = \varphi_\star(\mathcal{M})$ which turns $K'$ into a smooth embedded submanifold of $\mathbb{R}^{2p}$. The inverse map $\varphi_\star^{-1} : K' \to \mathcal{M}$ is smooth. For any $f : \mathcal{M} \to \mathbb{R}$, the function $h = f \circ \varphi_\star^{-1}$ is a well-defined map on $K'$.

To compare norms, fix a finite $C^\infty$ atlas $\{(U_i, \varphi_i)\}_{i=1}^N$ on $\mathcal{M}$ (finite since $\mathcal{M}$ is compact), and define the induced atlas on $K'$ by $V_i := \Psi(U_i)$ and $\tilde{\varphi}_i := \varphi_i \circ \Psi^{-1} : V_i \to \Omega_i := \varphi_i(U_i)$. Then $\tilde{\varphi}_i^{-1} = (\varphi_i \circ \Psi^{-1})^{-1} = \Psi \circ \varphi_i^{-1}$, and

$$h \circ \tilde{\varphi}_i^{-1} = f \circ \Psi^{-1} \circ (\varphi_i \circ \Psi^{-1})^{-1} = f \circ \Psi^{-1} \circ (\Psi \circ \varphi_i^{-1}) = f \circ \varphi_i^{-1}.$$

In particular, if we set $f_i := f \circ \varphi_i^{-1}$ and $h_i := h \circ \tilde{\varphi}_i^{-1}$, then $h_i = f_i$ pointwise on $\Omega_i$. Since $f \in C^{k,1}(\mathcal{M})$, by definition $f_i \in C^{k,1}(\Omega_i)$, i.e. for every multi-index $\alpha$ with $|\alpha| \le k$ the Euclidean derivative $D^\alpha f_i$ exists and is continuous on $\Omega_i$, and for $|\alpha| = k$ the Lipschitz seminorm

$$[D^\alpha f_i]_{\mathrm{Lip}(\Omega_i)} := \sup_{x \neq y \in \Omega_i} \frac{|D^\alpha f_i(x) - D^\alpha f_i(y)|}{\|x - y\|}$$

is finite. Because $h_i = f_i$, the derivatives agree wherever they exist, and hence for all $|\alpha| \le k$ we have $D^\alpha h_i = D^\alpha f_i$, and in particular for $|\alpha| = k$,

$$[D^\alpha h_i]_{\mathrm{Lip}(\Omega_i)} = [D^\alpha f_i]_{\mathrm{Lip}(\Omega_i)} < \infty.$$

Thus $h_i \in C^{k,1}(\Omega_i)$ for every $i$, which is precisely $h \in C^{k,1}(K')$.

Finally, since $K'$ is compact, any two atlas-based definitions of $\|\cdot\|_{C^{k,1}(K')}$ are equivalent, yielding a constant $C_{\mathrm{pb}} = C_{\mathrm{pb}}(\mathcal{M}, \Psi, k)$ such that

$$\|h\|_{C^{k,1}(K')} \leq C_{\mathrm{pb}} \max_{1 \leq i \leq N} \|h \circ \tilde{\varphi}_i^{-1}\|_{C^{k,1}(\Omega_i)} = C_{\mathrm{pb}} \|f\|_{C^{k,1}(\mathcal{M})}.$$

$\square$

*Step 2: Work on the embedded copy $K'$.* Given $f \in C^{k,1}(\mathcal{M})$, we set $h := f \circ \Psi^{-1}$ on $K'$. Lemma H.11 shows that $h \in C^{k,1}(K')$ and provides a uniform bound on $\|h\|_{C^{k,1}(K')}$ in terms of $\|f\|_{C^{k,1}(\mathcal{M})}$. To apply a Euclidean approximation theorem, however, we must further extend $h$ to a function defined on an open neighborhood of $K'$, and ultimately on the full cube. This is achieved using a tubular neighborhood around $K'$.

**Lemma H.12** (A bounded smooth domain inside the cube). *Let $K' \Subset (0,1)^{2p}$ be a compact embedded $C^\infty$ submanifold of $\mathbb{R}^{2p}$. Then there exists a radius $\rho > 0$ and an open set $\Omega \subset \mathbb{R}^{2p}$ such that*

$$K' \Subset \Omega \Subset (0,1)^{2p},$$

*$\Omega$ is bounded, and $\partial\Omega$ is a $C^\infty$ hypersurface (in particular, $\Omega$ is a $C^{k,1}$ domain for every $k \geq 1$).*

*Proof.* For each $x \in K'$, define the normal space to $K'$ at $x$ to be the $p$-dimensional subspace $N_x K' \subseteq T_x \mathbb{R}^{2p}$ consisting of all vectors that are orthogonal to $T_x K'$ with respect to the Euclidean dot product. Furthermore, we define the normal bundle of $K'$ defined as

$$NK' = \{(x,v) \in \mathbb{R}^{2p} \times \mathbb{R}^{2p} : x \in K', v \in N_x K'\}.$$

By the tubular neighborhood theorem (Lee, 2012, Theorem 6.24), there exists an open neighborhood $\mathcal{U}$ of the zero section $K_0 = \{(x,0) : x \in K'\} \subseteq NK'$, an open neighborhood $U$ of $K'$ in $\mathbb{R}^{2p}$ and a smooth diffeomorphism

$$F : \mathcal{U} \to U \subset \mathbb{R}^{2p}$$

such that $F(x,0) = x$ for all $x \in K'$.

**Step 1: Find a uniform radius $\rho_0 > 0$ so that the $\rho_0$-disk bundle lies in $\mathcal{U}$.** Since $\mathcal{U}$ is open and contains the zero section, for every $x \in K'$ the point $(x,0) \in NK'$ lies in $\mathcal{U}$. Because $\mathcal{U}$ is open, there exist an open neighborhood $W_x \subset NK'$ of $(x,0)$ such that $W_x \subset \mathcal{U}$. By local triviality of the normal bundle, we can choose $W_x$ of the form

$$W_x \supset \{(y,v) : y \in O_x, \|v\| < \rho_x\}$$

for some open neighborhood $O_x \subset K'$ of $x$ and some $\rho_x > 0$. Hence

$$\{(y,v) : y \in O_x,\ \|v\| < \rho_x\} \subset \mathcal{U}.$$

The family $\{O_x\}_{x \in K'}$ is an open cover of $K'$. Since $K'$ is compact, there exist finitely many points $x_1, \ldots, x_m \in K'$ such that

$$K' \subset \bigcup_{j=1}^{m} O_{x_j}.$$

Define $\rho_0 := \min_{1 \leq j \leq m} \rho_{x_j}$. Because the minimum of finitely many positive numbers is positive, we have $\rho_0 > 0$. Now take any $(x,v) \in NK'$ with $\|v\| < \rho_0$. Since $\{O_{x_j}\}$ coves $K'$, then there exists some $j$ with $x \in O_{x_j}$. Then $\|v\| < \rho \leq \rho_{x_j}$, so

$$(x,v) \in \{(y,w) : y \in O_{x_j}, \|w\| < \rho_{x_j}\} \subset \mathcal{U}.$$

Therefore the $\rho_0$-disk bundle

$$D_{\rho_0} := \{(x,v) \in NK' : \|v\| < \rho_0\}$$

satisfies

$$D_{\rho_0} \subset \mathcal{U}.$$

**Step 2: Use the assumption $K' \Subset (0,1)^{2p}$ to keep the tube inside the cube.** The notation $K' \Subset (0,1)^{2p}$ means that $K'$ is compact and contained in the open set $(0,1)^{2p}$. Equivalently $K' \cap \partial[0,1]^d = \varnothing$. Consider the distance from $K'$ to the closet set $\partial[0,1]^{2p}$:

$$\delta := \text{dist}(K', \partial[0,1]^{2p}) := \inf\{\|x - y\| : x \in K', y \in \partial[0,1]^{2p}\}$$

We claim that $\delta > 0$. Indeed, the function $x \mapsto \text{dist}(x, \partial[0,1]^{2p})$ is continuous on $\mathbb{R}^{2p}$, hence its restriction to the compact set $K'$ attains a minimum by the extreme value theorem. Since $K' \subset (0,1)^{2p}$ and $\partial[0,1]^{2p}$ is the boundary of the cube, every $x \in K'$ satisfies $\text{dist}(x, \partial[0,1]^{2p}) > 0$. Therefore the minimum is strictly positive, and thus $\delta > 0$.

Now choose $\rho := \min\{\rho_0/2, \delta/2\}$ and define

$$D_\rho := \{(x,v) \in NK' : \|v\| < \rho\}, \qquad \Omega := F(D_\rho) \subset \mathbb{R}^d.$$

**Step 3: $\Omega$ is open and contains $K'$.** Since $D_\rho$ is open in $NK'$ (it is defined by the strict inequality $\|v\| < \rho$), and $F$ is a diffeomorphism, it follows that

$$\Omega = F(D_\rho)$$

is open in $\mathbb{R}^{2p}$. Moreover, $K' \subset \Omega$: if $x \in K'$, then $(x,0) \in D_\rho$, and therefore

$$x = F(x,0) \in F(D_\rho) = \Omega.$$

Thus $K' \subset \Omega$. Then $\Omega$ is open (image of an open set under a diffeomorphism), and $K' \subset \Omega$ since $(x,0) \in D_\rho$ and $F(x,0) = x$.

**Step 4: $\Omega \Subset (0,1)^{2p}$, hence $\Omega$ is bounded.** We show that $\Omega \subset (0,1)^{2p}$. Let $z \in \Omega$. Then $z = F(x,v)$ for some $(x,v) \in D_\rho$, so $\|v\| < \rho \leq \delta/2$. For tubular neighborhoods in $\mathbb{R}^{2p}$, one has the geometric interpretation that $F(x,v)$ is obtained by moving from $x \in K'$ in a normal direction by length $\|v\|$; in particular the Euclidean distance from $z$ to $x$ is $\|v\|$, hence

$$\|z - x\|_2 \leq \|v\| < \rho \leq \delta/2.$$

Now suppose for contradiction that $z \notin (0,1)^{2p}$. Since $(0,1)^{2p}$ is the interior of $[0,1]^{2p}$ this implies that $z \in \mathbb{R}^{2p} \setminus (0,1)^{2p}$. and therefore the segment from $x \in (0,1)^{2p}$ to $z \notin (0,1)^{2p}$ must cross the boundary $\partial[0,1]^{2p}$. In particular,

$$\text{dist}(x, \partial[0,1]^{2p}) \leq \|z - x\|_2.$$

But $\text{dist}(x, \partial[0,1]^{2p}) \geq \delta$ for every $x \in K'$ by definition of $\delta$, so we would get

$$\delta \leq \|z - x\|_2 < \delta/2,$$

a contradiction. Hence $z \in (0,1)^{2p}$. This shows that $\Omega \subset (0,1)^{2p}$. To see that $\Omega \Subset (0,1)^{2p}$, it remains to show that $\overline{\Omega} \subset (0,1)^{2p}$. Since $\|v\| \leq \rho$ implies $(x,v) \in \overline{D_\rho}$, the closure satisfies

$$\overline{\Omega} = \overline{F(D_\rho)} = F(\overline{D_\rho}),$$

because $F$ is a homeomorphism, hence it maps closures to closures. Now the same distance argument $\|v\| \leq \rho \leq \delta/2$ shows that every point of $F(\overline{D_\rho})$ still lies at distance at most $\rho$ from some $x \in K'$, and therefore cannot reach $\partial[0,1]^{2p}$. Thus $\overline{\Omega} \subset (0,1)^{2p}$, i.e. $\Omega \Subset (0,1)^{2p}$. Since $(0,1)^{2p}$ is bounded and $\Omega \subset (0,1)^{2p}$, it follows immediately that $\Omega$ is bounded.

**Step 5: $\partial\Omega$ is a $C^{k,1}$ domain.** Finally, consider the sphere bundle

$$S_\rho := \{(x,v) \in NK' : \|v\| = \rho\}.$$

We claim that $S_\rho$ is a $C^\infty$ hypersurface in $NK'$. To see this, recall first that $NK'$ is a smooth vector bundle (as $K' \subset \mathbb{R}^d$ is an embedded $C^\infty$ submanifold), so each local trivialization

$$\tau_O : NK'|_O \to O \times \mathbb{R}^p$$

is a $C^\infty$ diffeomorphism. Here $O \times \mathbb{R}^p$ is equipped with its standard product smooth structure, i.e. the one generated by the product charts $(\varphi \times \text{id})(U \times \mathbb{R}^p)$ whenever $(U, \varphi)$ is a chart on $O$. Thus any smoothness statement about subsets of $NK'|_O$ may be checked in the product coordinates $O \times \mathbb{R}^p$ (Lee, 2012, Chapter 10).

Fix such a trivialization over an open set $O \subset K'$. In these coordinates define

$$G : O \times \mathbb{R}^p \to \mathbb{R}, \qquad G(y, w) = \|w\|^2.$$

The map $G$ is $C^\infty$ (indeed polynomial in the fiber coordinates), and moreover

$$S_\rho \cap (O \times \mathbb{R}^p) = \{(y, w) : \|w\| = \rho\} = \{(y, w) : \|w\|^2 = \rho^2\} = G^{-1}(\rho^2).$$

Now take any $(y, w) \in G^{-1}(\rho^2)$. Then $|w| = \rho > 0$, hence $w \neq 0$. The differential of $G$ at $(y, w)$ is the linear map

$$dG_{(y,w)} : T_{(y,w)}(O \times \mathbb{R}^p) \to \mathbb{R}.$$

Using the canonical identification $T_{(y,w)}(O \times \mathbb{R}^p) \simeq T_y O \times T_w \mathbb{R}^p$ and $T_w \mathbb{R}^p \simeq \mathbb{R}^p$, we may regard a tangent vector at $(y, w)$ as a pair $(\dot{y}, \dot{w}) \in T_y O \times \mathbb{R}^p$. Since $G(y, w) = |w|^2$ does not depend on $y$, its differential has no $\dot{y}$-contribution, and differentiating in the fiber direction gives

$$dG_{(y,w)}(\dot{y}, \dot{w}) = \left.\frac{d}{dt}\right|_{t=0} G(y, w + t\dot{w}) = \left.\frac{d}{dt}\right|_{t=0} |w + t\dot{w}|^2 = 2\langle w, \dot{w}\rangle.$$

Choosing the admissible tangent direction $(\dot{y}, \dot{w}) = (0, w) \in T_y O \times \mathbb{R}^p$ yields

$$dG_{(y,w)}(0, w) = 2\langle w, w\rangle = 2\|w\|^2 = 2\rho^2 > 0,$$

so $dG_{(y,w)}$ is not the zero map. Since the target is $\mathbb{R}$, this is equivalent to surjectivity of $dG_{(y,w)}$. Therefore $\rho^2$ is a regular value of $G$, and by the regular level set theorem (preimage theorem) $G^{-1}(\rho^2)$ is a $C^\infty$ submanifold of codimension 1 in $O \times \mathbb{R}^p$, i.e. a $C^\infty$ hypersurface (Lee, 2012, Corollary 5.14).

Since the above description holds in every bundle chart, we now show that this implies that $S_\rho$ is a global $C^\infty$ hypersurface in $NK'$. Let $\{(O_i, \tau_i)\}_{i \in I}$ be a smooth bundle atlas for $NK'$, where

$$\tau_i : NK'|_{O_i} \longrightarrow O_i \times \mathbb{R}^p$$

is a $C^\infty$ diffeomorphism onto its image and the transition maps

$$\tau_{ij} := \tau_j \circ \tau_i^{-1} : (O_i \cap O_j) \times \mathbb{R}^p \longrightarrow (O_i \cap O_j) \times \mathbb{R}^p$$

are $C^\infty$ and fiberwise linear. In the chart $\tau_i$, the sphere bundle is represented as

$$\tau_i(S_\rho \cap NK'|_{O_i}) = \{(y, w) \in O_i \times \mathbb{R}^p : \|w\| = \rho\}.$$

By the regular level set theorem applied to $G(y, w) = \|w\|^2$, this set is a $C^\infty$ hypersurface in $O_i \times \mathbb{R}^p$. Now consider an overlap $O_i \cap O_j \neq \varnothing$. Since $\tau_{ij}$ is a diffeomorphism, it sends $C^\infty$ hypersurfaces to $C^\infty$ hypersurfaces. Moreover, because $\tau_{ij} = \tau_j \circ \tau_i^{-1}$, we have the identity of sets on the overlap:

$$\tau_{ij}\Big(\tau_i(S_\rho \cap NK'|_{O_i \cap O_j})\Big) = \tau_j(S_\rho \cap NK'|_{O_i \cap O_j}).$$

Thus the local hypersurface charts obtained in $\tau_i$ and $\tau_j$ agree on overlaps via the smooth transition map $\tau_{ij}$. Consequently, the family $\{S_\rho \cap NK'|_{O_i}\}_{i \in I}$ defines a globally well-defined embedded $C^\infty$ submanifold of codimension 1 in $NK'$, i.e. $S_\rho$ is a global $C^\infty$ hypersurface in $NK'$. (Lee, 2012, see e.g. smooth vector bundles, bundle atlases, and smooth transition maps)

We claim that $\partial D_\rho = S_\rho$. Indeed, the fiberwise norm $(x, v) \mapsto \|v\|$ is continuous, so

$$\overline{D_\rho} = \{(x, v) : \|v\| \le \rho\}.$$

If $\|v\| < \rho$ then $(x, v) \in D_\rho$ (hence not on the boundary), while if $\|v\| > \rho$ then $(x, v) \notin \overline{D_\rho}$ (hence not on the boundary). Thus any boundary point must satisfy $\|v\| = \rho$, so $\partial D_\rho \subset S_\rho$. Conversely, if $\|v\| = \rho$, then every neighborhood of $(x, v)$ in $NK'$ contains points with $\|v\| < \rho$ and points with $\|v\| > \rho$, so $(x, v) \in \partial D_\rho$. Hence $\partial D_\rho = S_\rho$.

Since $F : \mathcal{U} \to U$ is a diffeomorphism and we chose $\rho < \rho_0$ so that $\overline{D_\rho} \subset D_{\rho_0} \subset \mathcal{U}$, it follows that $F$ restricts to a homeomorphism on $\mathcal{U}$ (in particular on a neighborhood of $\overline{D_\rho}$). A homeomorphism maps boundaries to boundaries: for any $A \subset \mathcal{U}$,

$$F(\partial A) = \partial(F(A)),$$

since it preserves closures and interiors. Applying this with $A = D_\rho$ and $\Omega := F(D_\rho)$ yields

$$\partial\Omega = \partial(F(D_\rho)) = F(\partial D_\rho) = F(S_\rho).$$

Finally, since $S_\rho$ is a $C^\infty$ hypersurface in $NK'$ and $F$ is a diffeomorphism on $\mathcal{U}$, the restriction $F|_{S_\rho} : S_\rho \to F(S_\rho) = \partial\Omega$ is a $C^\infty$ diffeomorphism onto its image. Hence $\partial\Omega$ is a $C^\infty$ hypersurface in $\mathbb{R}^{2p}$.

To connect this with Definition C.1, fix $z_\star \in \partial\Omega$. Because $\partial\Omega$ is a $C^\infty$ hypersurface in $\mathbb{R}^{2p}$, it is in particular a $(2p-1)$-dimensional embedded $C^\infty$ submanifold. Hence the tangent space $T_{z_\star}(\partial\Omega) \subset \mathbb{R}^{2p}$ is a $(2p-1)$-dimensional linear subspace. Composing with a rigid motion of $\mathbb{R}^{2p}$ (translation by $-z_\star$ followed by an orthogonal rotation), which is a $C^\infty$ diffeomorphism with $C^\infty$ inverse, we may assume without loss of generality that $z_\star = 0$ and the the tangent hyperplane at $0 \in \partial\Omega$ is the standard "horizontal" hyperplane where the last coordinate is zero, that is

$$T_0(\partial\Omega) = \{x_{2p} = 0\}.$$

Since $\partial\Omega$ is an embedded $C^\infty$ hypersurface, it is locally a regular level set: by (Lee, 2012, Prop. 5.16) there exist a neighborhood $U$ of $0$ and a $C^\infty$ submersion $\Phi : U \to \mathbb{R}$ such that

$$\partial\Omega \cap U = \Phi^{-1}(0) = \{x \in U : \; \Phi(x) = 0\}.$$

In particular, since $\Phi$ is a submersion, $d\Phi_0 \neq 0$, which in Euclidean coordinates is equivalent to $\nabla\Phi(0) \neq 0$.

Moreover, since $\partial\Omega \cap U = \{\Phi = 0\}$ with $\nabla\Phi(0) \neq 0$, the tangent space of the level set at $0$ satisfies

$$T_0(\partial\Omega) = \ker(d\Phi_0) = \{v \in \mathbb{R}^{2p} : \; d\Phi_0(v) = 0\} = \{v \in \mathbb{R}^{2p} : \; \nabla\Phi(0) \cdot v = 0\},$$

i.e. $T_0(\partial\Omega)$ is the hyperplane orthogonal to $\nabla\Phi(0)$. The assumption $T_0(\partial\Omega) = \{v \in \mathbb{R}^{2p} : v_{2p} = 0\}$ therefore implies that

$$\nabla\Phi(0) \in \big(T_0(\partial\Omega)\big)^\perp = \mathrm{span}\{e_{2p}\},$$

where $e_{2p} = (0, \ldots, 0, 1)$ is the $x_{2p}$-axis direction. Hence there exists $\lambda \neq 0$ such that

$$\nabla\Phi(0) = \lambda e_{2p},$$

and in particular the last component of the gradient is nonzero:

$$\partial_{x_{2p}}\Phi(0) = (\nabla\Phi(0))_{2p} = \lambda \neq 0.$$

By continuity of $\partial_{x_{2p}}\Phi$, we may (after possibly shrinking $U$) assume that $\partial_{x_{2p}}\Phi(x) \neq 0$ for all $x \in U$, which is the condition needed to apply the implicit function theorem and solve $\Phi(x', x_{2p}) = 0$ for $x_{2p}$ as a function of $x'$.

By the implicit function theorem applied to the equation $\Phi(x', x_{2p}) = 0$, there exist $r > 0$ and a $C^\infty$ function $g : B^{2p-1}(0, r) \to \mathbb{R}$ such that
$$\partial\Omega \cap B(0, r) = \{(x', x_{2p}) \in B(0, r) : \; x_{2p} = g(x')\}.$$

Next, set $H(x', x_{2p}) := x_{2p} - g(x')$. Since $g$ is continuous, $H$ is continuous, and therefore the sets

$$U_+ := \{(x', x_{2p}) \in B(0, r) : \; H(x', x_{2p}) > 0\} = \{x_{2p} > g(x')\} \cap B(0, r),$$
$$U_- := \{(x', x_{2p}) \in B(0, r) : \; H(x', x_{2p}) < 0\} = \{x_{2p} < g(x')\} \cap B(0, r)$$

are open in $B(0, r)$. Moreover, $U_+ \cap U_- = \varnothing$, and every point of $B(0, r) \setminus \partial\Omega$ satisfies $x_{2p} \neq g(x')$, hence belongs to exactly one of $U_+$ or $U_-$. Thus

$$B(0, r) \setminus \partial\Omega = U_+ \,\dot\cup\, U_-.$$

Since $\Omega$ is a domain (open and connected), the intersection $\Omega \cap B(0, r)$ is open, and by shrinking $r$ if necessary we may assume $\Omega \cap B(0, r)$ is connected.[1] As $\partial\Omega \cap B(0, r)$ separates the ball into exactly the two components $U_+$ and $U_-$, the connected set $\Omega \cap B(0, r)$ must be contained in exactly one of them. Replacing $g$ by $-g$ (equivalently swapping the labels of the two sides) if necessary, we may assume that

$$\Omega \cap B(0, r) \subset U_+, \qquad \text{i.e.} \qquad \Omega \cap B(0, r) \subset \{(x', x_{2p}) \in B(0, r) : x_{2p} > g(x')\}.$$

Define the boundary-straightening map

$$\psi : B(0, r) \to \psi(B(0, r)), \qquad \psi(x', x_{2p}) := (x', x_{2p} - g(x')).$$

Since $g \in C^\infty$, the map $\psi$ is $C^\infty$. Its inverse is given explicitly by

$$\psi^{-1}(y', y_{2p}) = (y', y_{2p} + g(y')),$$

which is also $C^\infty$. Equivalently, the Jacobian matrix of $\psi$ is

$$D\psi(x) = \begin{pmatrix} I_{2p-1} & 0 \\ -\nabla g(x')^\top & 1 \end{pmatrix}, \qquad \text{so} \qquad \det D\psi(x) = 1,$$

hence $\psi$ is a $C^\infty$ diffeomorphism from $B(0, r)$ onto its image $\psi(B(0, r))$, with inverse $\psi^{-1}(y', y_{2p}) = (y', y_{2p} + g(y'))$. In particular, $\psi, \psi^{-1} \in C^\infty \subset C^{k,1}$ for every $k \geq 1$.

By the one-sided inclusion above, if $(x', x_{2p}) \in \Omega \cap B(0, r)$ then $x_{2p} > g(x')$, so the last coordinate of $\psi(x', x_{2p})$ is $x_{2p} - g(x') > 0$. Thus

$$\psi\big(B(0, r) \cap \Omega\big) \subset \mathbb{R}_+^{2p}.$$

On the other hand, if $(x', x_{2p}) \in \partial\Omega \cap B(0, r)$, then $x_{2p} = g(x')$, so the last coordinate of $\psi(x', x_{2p})$ is $0$, and therefore

$$\psi\big(B(0, r) \cap \partial\Omega\big) \subset \partial\mathbb{R}_+^{2p}.$$

This verifies the boundary-straightening property in Definition C.1. Hence $\Omega$ is a $C^{k,1}$ domain for every $k \geq 1$.

$\square$

*Step 3: Produce a regular domain around $K'$.* Lemma H.12 constructs a bounded $C^{k,1}$ domain $\Omega$ such that $K' \Subset \Omega \Subset (0, 1)^d$. This provides a setting where classical extension results for Hölder functions on domains with regular boundary apply. In the next lemma we combine the tubular retraction $\pi$ (to define a function on $\overline{\Omega}$ agreeing with $h$ on $K'$) with the extension Lemma C.2 to obtain a function on the entire cube.

**Lemma H.13** (Extension from $K'$ to the cube with controlled $C^{k,1}$ norm). *Let $K' \Subset (0, 1)^{2p}$ be a compact embedded $C^\infty$ submanifold and let $h \in C^{k,1}(K')$. Then there exists $H \in C^{k,1}([0, 1]^{2p})$ such that $H = h$ on $K'$ and*

$$\|H\|_{C^{k,1}([0,1]^{2p})} \leq C_{\text{ext}}\|h\|_{C^{k,1}(K')},$$

*for some constant $C_{\text{ext}} = C_{\text{ext}}(K', k)$.*

*Proof.* **Step 1: A tubular neighborhood and a smooth retraction onto $K'$.** Let $F : \mathcal{U} \to U \subset \mathbb{R}^{2p}$ be a tubular neighborhood diffeomorphism given by the tubular neighborhood theorem, defined on an open neighborhood $\mathcal{U}$ of the zero section in the normal bundle $NK'$, with $F(x, 0) = x$ for all $x \in K'$ (see e.g. (Lee, 2012, Thm. 6.24)). As in Lemma H.12, choose $\rho_0 > 0$ such that

$$D_{\rho_0} := \{(x, v) \in NK' : \|v\| < \rho_0\} \subset \mathcal{U},$$

and then choose $0 < \rho < \rho_0$ so that $\overline{D_\rho} \subset D_{\rho_0} \subset \mathcal{U}$ (e.g. $\rho := \rho_0/2$). Define the tubular domain

$$\Omega := F(D_\rho) \Subset (0, 1)^{2p}.$$

---

[1] For instance, because open connected sets are locally path-connected, one can choose $r > 0$ so small that any two points of $\Omega \cap B(0, r)$ can be joined by a path contained in $\Omega \cap B(0, r)$.

By construction $F$ and $F^{-1}$ are $C^\infty$ on open neighborhoods of $\overline{D}_\rho$ and $\overline{\Omega}$, respectively.

Let $\mathrm{pr} : NK' \to K'$ denote the bundle projection $(x, v) \mapsto x$, and define

$$\pi := \mathrm{pr} \circ F^{-1} \quad \text{on an open neighborhood of } \overline{\Omega}.$$

Then $\pi$ is $C^\infty$, maps $\Omega$ into $K'$, and satisfies $\pi|_{K'} = \mathrm{id}_{K'}$, because for $x \in K'$ we have $F^{-1}(x) = (x, 0)$ and thus $\pi(x) = \mathrm{pr}(x, 0) = x$.

**Step 2: Extend $h$ from $K'$ to $\overline{\Omega}$ by retraction.** Define

$$u : \overline{\Omega} \to \mathbb{R}, \qquad u(z) := h(\pi(z)).$$

Then $u = h$ on $K'$, since $\pi|_{K'} = \mathrm{id}_{K'}$.

We claim that $u \in C^{k,1}(\overline{\Omega})$ and that

$$\|u\|_{C^{k,1}(\overline{\Omega})} \le C_0 \|h\|_{C^{k,1}(K')}$$

for some constant $C_0 = C_0(K', F, k)$. To see this, pull back $u$ to $D_\rho$ via $F$: for $(x, v) \in D_\rho$,

$$(u \circ F)(x, v) = h\big(\pi(F(x, v))\big) = h\big(\mathrm{pr}(x, v)\big) = h(x).$$

Thus $u \circ F$ depends only on the base variable $x$ and is constant along each fiber direction $v$.

Fix a finite $C^\infty$ atlas $\{(O_i, \varphi_i)\}_{i=1}^N$ on $K'$ and corresponding local trivializations $NK'|_{O_i} \simeq O_i \times \mathbb{R}^p$. In the induced coordinates $(y, w) \in \varphi_i(O_i) \times B^p(0, \rho)$, the function $u \circ F$ is represented by

$$(u \circ F) \circ (\varphi_i^{-1} \times \mathrm{id})(y, w) = h \circ \varphi_i^{-1}(y),$$

which is independent of $w$. Consequently:

- all partial derivatives in the $w$-variables vanish;

- partial derivatives in the $y$-variables up to order $k$ coincide with those of $h \circ \varphi_i^{-1}$;

- the Lipschitz seminorm of the $k$-th derivatives (in $(y, w)$) is controlled by the Lipschitz seminorm in $y$ since there is no $w$-dependence.

Therefore, for each chart $i$,

$$\|(u \circ F) \circ (\varphi_i^{-1} \times \mathrm{id})\|_{C^{k,1}(\varphi_i(O_i) \times B^p(0,\rho))} \le \|h \circ \varphi_i^{-1}\|_{C^{k,1}(\varphi_i(O_i))}.$$

Since $u = (u \circ F) \circ F^{-1}$ and $F^{-1}$ is $C^\infty$ on a neighborhood of $\overline{\Omega}$, the chain rule up to order $k$ and the Lipschitz control of $k$-th derivatives yield the claimed estimate with a constant $C_0$ depending on uniform bounds of derivatives of $F^{-1}$ on $\overline{\Omega}$ up to order $k + 1$. This proves $u \in C^{k,1}(\overline{\Omega})$ and the stated bound.

**Step 3: Apply the $C^{k,1}$ extension theorem and restrict to the cube.** By Lemma H.12, $\Omega$ is a bounded $C^\infty$ domain, hence a bounded $C^{k,1}$ domain. Choose $\eta > 0$ and set

$$\Omega' := (-\eta, 1 + \eta)^{2p},$$

so that $\overline{\Omega} \subset \Omega'$ and $[0, 1]^{2p} \subset \Omega'$. Applying Lemma C.2 (with $\alpha = 1$ and $n = 2p$) to $u \in C^{k,1}(\overline{\Omega})$, we obtain $w \in C^{k,1}(\Omega')$ such that $w = u$ on $\Omega$ and

$$\|w\|_{C^{k,1}(\Omega')} \le C_1 \|u\|_{C^{k,1}(\overline{\Omega})}, \qquad C_1 = C_1(k, 1, \Omega, \Omega').$$

Finally define $H := w|_{[0,1]^{2p}}$. Then $H \in C^{k,1}([0, 1]^{2p})$, and since $K' \subset \Omega$ and $w = u$ on $\Omega$ with $u = h$ on $K'$, we have $H = h$ on $K'$. Moreover, restriction cannot increase the $C^{k,1}$ norm, hence

$$\|H\|_{C^{k,1}([0,1]^{2p})} \le \|w\|_{C^{k,1}(\Omega')} \le C_1 \|u\|_{C^{k,1}(\overline{\Omega})} \le C_1 C_0 \|h\|_{C^{k,1}(K')}.$$

Setting $C_{\mathrm{ext}} := C_0 C_1$ completes the proof. $\qquad\square$

*Step 4: Reduce to Euclidean approximation on* $[0,1]^{2p}$. Lemma H.13 produces a function $H \in C^{k,1}([0,1]^{2p})$ such that $H = h$ on $K'$, together with the norm control $\|H\|_{C^{k,1}([0,1]^{2p})} \lesssim \|h\|_{C^{k,1}(K')}$. At this point the problem is purely Euclidean: approximate $H$ uniformly on the cube by a ReLU network. This is exactly the content of Theorem H.9, after a normalization to the Hölder unit ball.

**Lemma H.14** (Yarotsky approximation on the cube after normalization). *Let* $H \in C^{k,1}([0,1]^{2p})$ *and set* $B := \|H\|_{C^{k,1}([0,1]^{2p})}$. *If* $B > 0$, *define* $\widetilde{H} := H/B$. *Then* $\|\widetilde{H}\|_{C^{k,1}([0,1]^{2p})} \leq 1$, *and for every* $0 < \varepsilon < 1$, *Theorem H.9 yields a ReLU network* $\widetilde{g}_\varepsilon$ *such that*

$$\|\widetilde{H} - \widetilde{g}_\varepsilon\|_\infty \leq \frac{\varepsilon}{B}.$$

*Consequently,* $g_\varepsilon := B\widetilde{g}_\varepsilon$ *satisfies*

$$\|H - g_\varepsilon\|_\infty \leq \varepsilon,$$

*with the corresponding size/depth bounds from Theorem H.9 evaluated at accuracy* $\varepsilon/B$.

*Proof.* Immediate from the scaling $\widetilde{H} = H/B$ and Theorem H.9. □

*Step 5: Pull the network back to* $\mathcal{M}$. Lemma H.14 yields a network $g_\varepsilon$ approximating $H$ uniformly on $[0,1]^{2p}$. Since $H$ agrees with $h$ on $K'$, and $h$ is the pushforward of $f$ through $\varphi_\star$, we recover an approximant on $\mathcal{M}$ simply by composition: $\widehat{g}_\varepsilon := g_\varepsilon \circ \varphi_\star$. The final lemma records that the uniform approximation error on the cube immediately implies uniform approximation on $\mathcal{M}$.

**Lemma H.15** (Pullback of the Euclidean approximant to $\mathcal{M}$). *Let* $\varphi_\star : \mathcal{M} \to K' \subset [0,1]^{2p}$ *be as above and let* $H \in C^{k,1}([0,1]^{2p})$ *satisfy* $H = h$ *on* $K'$, *where* $h = f \circ \varphi_\star^{-1}$. *If* $g_\varepsilon$ *satisfies* $\|H - g_\varepsilon\|_\infty \leq \varepsilon$, *then the composed function*

$$\widehat{g}_\varepsilon := g_\varepsilon \circ \varphi_\star$$

*satisfies*

$$\sup_{x \in \mathcal{M}} |f(x) - \widehat{g}_\varepsilon(x)| \leq \varepsilon.$$

*Proof.* For $x \in \mathcal{M}$, $\varphi_\star(x) \in K'$ and $H(\varphi_\star(x)) = h(\varphi_\star(x)) = f(x)$. Thus

$$|f(x) - \widehat{g}_\varepsilon(x)| = |H(\varphi_\star(x)) - g_\varepsilon(\varphi_\star(x))| \leq \sup_{u \in [0,1]^d} |H(u) - g_\varepsilon(u)| \leq \varepsilon.$$

Taking the supremum over $x \in \mathcal{M}$ yields the claim. □

*Completion of the proof.* With these ingredients in place, the proof of Theorem 5.1 is obtained by concatenating the lemmas: embedding and normalization (Lemma H.10), norm-controlled pullback (Lemma H.11), norm-controlled extension to the cube (Lemmas H.12–H.13), Euclidean approximation on the cube (Lemma H.14), and pullback of the approximant (Lemma H.15).

*Proof of Theorem 5.1.* Let $\psi \in C^{kp,1}(\mathcal{M})$ and fix $\varepsilon \in (0,1)$. Define

$$M := \|\psi\|_{C^{kp,1}(\mathcal{M})}.$$

If $M = 0$, then $\|\psi\|_\infty \leq \|\psi\|_{C^{kp,1}(\mathcal{M})} = 0$, hence $\psi \equiv 0$ and the claim is trivial. From now on assume $M > 0$.

If $\varepsilon \geq M$, then $\|\psi\|_\infty \leq M \leq \varepsilon$, so the constant network $\hat{g}_\varepsilon \equiv 0$ yields $\|\psi - \hat{g}_\varepsilon \circ \varphi_\star\|_\infty \leq \varepsilon$ for any feature map $\varphi_\star$. Thus we may assume

$$0 < \varepsilon < M, \qquad \text{so that} \qquad 0 < \frac{\varepsilon}{M} < 1.$$

Define the normalized function

$$\tilde{\psi} := \frac{1}{M}\psi, \qquad \text{so that} \qquad \|\tilde{\psi}\|_{C^{kp,1}(\mathcal{M})} = 1.$$

By Lemma H.10, choose a smooth embedding $\varphi^\star : \mathcal{M} \to K' \subset [0,1]^{2p}$. Define

$$\widetilde{h} := \tilde{\psi} \circ \varphi_\star^{-1} \quad \text{on } K'.$$

By Lemma H.11, we have $\widetilde{h} \in C^{kp,1}(K')$ and

$$\|\widetilde{h}\|_{C^{kp,1}(K')} \leq C_{\mathrm{pb}}\|\tilde{\psi}\|_{C^{kp,1}(\mathcal{M})} = C_{\mathrm{pb}}.$$

By Lemma H.13, there exists $\widetilde{H} \in C^{kp,1}([0,1]^{2p})$ such that $\widetilde{H} = \widetilde{h}$ on $K'$ and

$$\|\widetilde{H}\|_{C^{kp,1}([0,1]^{2p})} \leq C_{\mathrm{ext}}\|\widetilde{h}\|_{C^{kp,1}(K')} \leq C_{\mathrm{ext}}C_{\mathrm{pb}}.$$

Set

$$\widetilde{B} := \|\widetilde{H}\|_{C^{kp,1}([0,1]^{2p})}.$$

Since $M > 0$, $\tilde{\psi} \not\equiv 0$, hence $\widetilde{h} \not\equiv 0$ on $K'$. Because $\widetilde{H} = \widetilde{h}$ on $K'$, $\widetilde{H} \not\equiv 0$ on $[0,1]^{2p}$, hence $\|\widetilde{H}\|_\infty > 0$. Since $\|\widetilde{H}\|_{C^{kp,1}([0,1]^{2p})} \geq \|\widetilde{H}\|_\infty$, it follows that

$$\widetilde{B} > 0.$$

Normalize once more:

$$\overline{H} := \frac{1}{\widetilde{B}}\widetilde{H}, \quad \text{so that} \quad \|\overline{H}\|_{C^{kp,1}([0,1]^{2p})} = 1,$$

Define the accuracy parameter

$$\delta := \frac{\varepsilon/M}{\widetilde{B}}.$$

If $\delta \geq 1$, then the zero network already approximates $\overline{H}$ within error 1, hence within error $\delta$, so the approximation step is trivial. Thus we may assume $\delta \in (0,1)$.

Now we apply Theorem H.9 with the following choices:

$$n := 2p, \qquad r := kp+1, \qquad \beta := \frac{3r}{2n} = \frac{3(kp+1)}{4p}.$$

This choice satisfies

$$\frac{r}{n} < \frac{kp+1}{2p} < \frac{3(kp+1)}{4p} = \beta \leq \frac{kp+1}{p} = \frac{2r}{n},$$

so Theorem H.9 applies. Therefore, there exist $W_\varepsilon$, $L_\varepsilon$ and a network

$$\bar{g}_\varepsilon \in \mathcal{NN}^\sigma_{2p,W_\varepsilon,L_\varepsilon}$$

such that

$$\|\overline{H} - \bar{g}_\varepsilon\|_\infty \leq \delta,$$

and

$$W_\varepsilon \leq \left(\frac{\tilde{C}_{r,2p,\beta,\sigma}}{\delta}\right)^{1/\beta}, \qquad L_\varepsilon \leq C_{r,2p,\sigma}W^{\beta(2p)/r-1}.$$

Rescale back by defining

$$\tilde{g}_\varepsilon := \widetilde{B}\bar{g}_\varepsilon.$$

Then

$$\|\widetilde{H} - \tilde{g}_\varepsilon\|_\infty = \widetilde{B}\|\overline{H} - \bar{g}_\varepsilon\|_\infty \leq \widetilde{B}\delta = \frac{\varepsilon}{M}.$$

Finally define

$$g_\varepsilon := M\tilde{g}_\varepsilon.$$

Scaling the output by $M$ (and previously by $\widetilde{B}$) is implemented by scaling the final affine layer, hence it does *not* change width $W_\varepsilon$ or depth $L_\varepsilon$. Moreover,

$$\|M\widetilde{H} - g_\varepsilon\|_\infty = M\|\widetilde{H} - \tilde{g}_\varepsilon\|_\infty \leq M \cdot \frac{\varepsilon}{M} = \varepsilon.$$

Set $H := M\widetilde{H}$. On $K'$ we have $H = M\widetilde{h} = h$, where

$$h := \psi \circ \varphi_\star^{-1},$$

because $\widetilde{h} = (1/M)h$.

Define the pullback approximant on $\mathcal{M}$ by

$$\hat{\psi}_\varepsilon := \hat{g}_\varepsilon \circ \varphi_\star.$$

For any $x \in \mathcal{M}$, let $u := \varphi_\star(x) \in K'$. Then $\psi(x) = h(u) = H(u)$ and $\hat{\psi}_\varepsilon(x) = \hat{g}_\varepsilon(u)$, hence

$$|\psi(x) - \hat{\psi}_\varepsilon(x)| = |H(u) - g_\varepsilon(u)| \le \|H - g_\varepsilon\|_\infty \le \varepsilon.$$

Taking the supremum over $x \in \mathcal{M}$ yields

$$\|\psi - \hat{\psi}_\varepsilon\|_\infty \le \varepsilon.$$

It remains to extract the stated rates. Since $\delta = (\varepsilon/M)/\widetilde{B}$,

$$W_\varepsilon \le \left(\frac{\widetilde{C}_{r,2p,\beta,\sigma}}{\delta}\right)^{1/\beta} = \left(\frac{\widetilde{C}_{r,2p,\beta,\sigma}\widetilde{B}M}{\delta}\right)^{1/\beta}.$$

Using $\widetilde{B} \le C_{\mathrm{ext}}C_{\mathrm{pb}}$, we get

$$W_\varepsilon \le C\left(\frac{M}{\varepsilon}\right)^{1/\beta}, \qquad C := \left(\widetilde{C}_{r,2p,\beta,\sigma}C_{\mathrm{ext}}C_{\mathrm{pb}}\right)^{1/\beta}.$$

With $\beta = \frac{3(kp+1)}{4p}$, we have

$$\frac{1}{\beta} = \frac{4p}{3(kp+1)},$$

so

$$W_\varepsilon \le C\left(\frac{M}{\varepsilon}\right)^{\frac{4p}{3(kp+1)}}.$$

Next, compute the exponent in the depth bound:

$$\beta\frac{2p}{kp+1} - 1 = \frac{3(kp+1)}{4p}\frac{2p}{kp+1} - 1 = \frac{3}{2} - 1 = \frac{1}{2}.$$

Hence

$$L_\varepsilon \le C_{r,2p,\sigma}W_\varepsilon^{1/2} \le C'\left(\frac{M}{\varepsilon}\right)^{\frac{2p}{3(kp+1)}}$$

for a constant $C'$ depending only on $(\mathcal{M}, \varphi_\star, k, \sigma)$.

Finally, since $\psi$ is fixed, $M = \|\psi\|_{C^{kp,1}(\mathcal{M})}$ is a fixed constant; absorbing $M$ into the implicit constant gives exactly the stated polynomial rates:

$$W_\varepsilon = \mathcal{O}\left(\varepsilon^{-\frac{4p}{3(kp+1)}}\right), \qquad L_\varepsilon = \mathcal{O}\left(\varepsilon^{-\frac{2p}{3(kp+1)}}\right).$$

This concludes the proof. $\qquad\square$

### H.5. Proof of Corollary 5.2

*Proof.* Fix $0 < \varepsilon < 1$. Define the *prepotential*

$$\psi_\star := \phi_\star^c.$$

By assumption, $\psi_\star \in C^{kp,1}(\mathcal{M})$. Let

$$\phi := \psi_\star^c.$$

Since $\phi_\star$ is $c$-concave, we have $\phi = \phi_\star$.

**Step 1: Uniform approximation of the prepotential $\psi_\star$.** Apply Theorem 5.1 to the function $\psi_\star \in C^{kp,1}(\mathcal{M})$. Then there exist a feature map $\varphi_\star : \mathcal{M} \to \mathbb{R}^{2p}$ satisfying Assumption 2.2, integers $W_\varepsilon, L_\varepsilon \in \mathbb{N}$, and a network $\hat{g}_\varepsilon \in \mathcal{NN}^\sigma_{2p, W_\varepsilon, L_\varepsilon}$ such that the pullback network

$$\hat{\psi}_\varepsilon := \hat{g}_\varepsilon \circ \varphi_\star$$

satisfies

$$\|\psi_\star - \hat{\psi}_\varepsilon\|_\infty < \varepsilon, \tag{26}$$

and $W_\varepsilon, L_\varepsilon$ obey the bounds stated in Theorem 5.1.

**Step 2: Transfer the error through the $c$-transform.** Define the $c$-transform $\hat{\phi}_\varepsilon := \hat{\psi}_\varepsilon^c$. We claim that

$$\|\phi - \hat{\phi}_\varepsilon\|_\infty \leq \|\psi_\star - \hat{\psi}_\varepsilon\|_\infty. \tag{27}$$

To prove this, set $\delta := \|\psi_\star - \hat{\psi}_\varepsilon\|_\infty$. Then for every $y \in \mathcal{M}$,

$$\psi_\star(y) - \delta \leq \hat{\psi}_\varepsilon(y) \leq \psi_\star(y) + \delta.$$

Fix $x \in \mathcal{M}$. For every $y \in \mathcal{M}$ we therefore have

$$c(x,y) - \psi_\star(y) - \delta \leq c(x,y) - \hat{\psi}_\varepsilon(y) \leq c(x,y) - \psi_\star(y) + \delta.$$

Taking the infimum over $y \in \mathcal{M}$ yields

$$\inf_y \big( c(x,y) - \psi_\star(y) \big) - \delta \leq \inf_y \big( c(x,y) - \hat{\psi}_\varepsilon(y) \big) \leq \inf_y \big( c(x,y) - \psi_\star(y) \big) + \delta.$$

By definition of the $c$-transform (with the convention $\eta^c(x) = \inf_y(c(x,y) - \eta(y))$), this is exactly

$$\phi(x) - \delta \leq \hat{\phi}_\varepsilon(x) \leq \phi(x) + \delta.$$

Hence $|\phi(x) - \hat{\phi}_\varepsilon(x)| \leq \delta$ for all $x \in \mathcal{M}$, and taking the supremum over $x$ proves (27).

Combining (27) with (26) gives

$$\|\phi - \hat{\phi}_\varepsilon\|_\infty \leq \|\psi_\star - \hat{\psi}_\varepsilon\|_\infty < \varepsilon.$$

Recalling $\phi = \phi_\star$, we obtain

$$\|\phi_\star - \hat{\phi}_\varepsilon\|_\infty < \varepsilon.$$

**Step 3: Complexity bounds.** The only neural approximation step is Step 1, where $\hat{\psi}_\varepsilon = \hat{g}_\varepsilon \circ \varphi_\star$ is constructed using Theorem 5.1. Step 2 applies the deterministic operator $\eta \mapsto \eta^c$ and does not alter the architecture of $\hat{g}_\varepsilon$. Therefore $\hat{g}_\varepsilon$ satisfies exactly the same width/depth bounds as in Theorem 5.1. This completes the proof. $\square$

### H.6. Proof of Theorem 5.3

Throughout this appendix, to streamline the notation and improve readability, we write $\phi$ in place of $\phi_\star$ and $\psi$ in place of $\psi_\star$. Before our proof, we review a few technical results from Cordero-Erausquin et al. (2001). We have the following characterizations about $c$-concave functions on $\mathcal{M}$.

**Lemma H.16.** *(Cordero-Erausquin et al., 2001, Lemma 3.3) Let $\phi$ be a $c$-concave function on $\mathcal{M}$ and define $T(x) = \exp_x(-\nabla\phi(x))$.*

- *The function $\phi$ is Lipschitz and hence differentiable almost everywhere on $\mathcal{M}$;*

- *Fix any $x \in \mathcal{M}$ where $\phi$ is differentiable. Then $y = T(x)$ if and only if $y$ minimizes*

$$c(x,y) - \phi(x) - \phi^c(y) \tag{28}$$

*over $y' \in \mathcal{M}$. In the latter case one has $\nabla\phi(x) = \nabla d_y^2(x)$.*

Fix $U \subseteq \mathcal{M}$ open. A function $f : U \to \mathbb{R}$ is *semi-concave* on $U$ if for any $x_0 \in U$, there exists a convex embedded ball $B_r(x_0)$ and a smooth function $V : B_r(x_0) \to \mathbb{R}$ such that $f + V$ is geodesically concave throughout $B_r(x_0)$. For a semi-concave function, its Hessian is defined in the following way.

**Definition H.17.** (Cordero-Erausquin et al., 2001, Definition 3.9) Let $f : U \to \mathbb{R}$ be a semi-concave function. Then $f$ has a Hessian at $x \in U$ if $f$ is differentiable at $x$ and there exists a self-adjoint operator $\mathrm{Hess}_x f : T_x\mathcal{M} \to T_x\mathcal{M}$ such that

$$\sup_{v \in \partial f(\exp_x(u))} \|\Pi_{x,u} v - \nabla f(x) - \mathrm{Hess} f u\|_x = o(\|u\|) \tag{29}$$

as $u \to 0$ in $T_x\mathcal{M}$. Here $\Pi_{x,u} : T_{\exp_x(u)} \to T_x\mathcal{M}$ denotes the parallel translation to $x$ along $\gamma(t) = \exp_x(tu)$.

If $f : U \to \mathbb{R}$ is semi-concave, the Aleksandrov–Bangert theorem claims that $\mathrm{Hess}_x f$ exists almost everywhere on $U$ with respect to the volume measure (Bangert, 1979). It turns out for compact $\mathcal{M}$, any $c$-concave function on $\mathcal{M}$ is semi-concave, hence admits a Hessian almost everywhere on $\mathcal{M}$ (Cordero-Erausquin et al., 2001, Proposition 3.14).

Let $\phi$ be a $c$-concave Kantorovich potential for the quadratic cost $c(x, y) = \frac{1}{2}d(x,y)^2$ and set $\psi := \phi^c$. At points where $\phi$ (resp. $\psi$) is differentiable, define

$$\overrightarrow{T}(x) := \exp_x(-\nabla\phi(x)), \qquad \overleftarrow{T}(y) := \exp_y(-\nabla\psi(y)).$$

and introduce the sets

$$E_\phi := \{x \in \mathcal{M} : \mathrm{Hess}_x\phi \text{ exists in the sense of (29)}\},$$
$$E_\psi := \{y \in \mathcal{M} : \mathrm{Hess}_y\psi \text{ exists in the sense of (29)}\}.$$

For $x \in E_\phi$ set $y = \overrightarrow{T}(x)$, and for $y \in E_\psi$ set $x = \overleftarrow{T}(y)$. Define the strict-positivity sets

$$\widetilde{E}_\phi := \left\{x \in E_\phi : \mathrm{Hess}_x\left(\tfrac{1}{2}d(\,\cdot\,, \overrightarrow{T}(x))^2 - \phi\right) > 0\right\},$$
$$\widetilde{E}_\psi := \left\{y \in E_\psi : \mathrm{Hess}_y\left(\tfrac{1}{2}d(\overleftarrow{T}(y), \,\cdot\,)^2 - \psi\right) > 0\right\}, \tag{30}$$

and the set of *nice points*

$$\Omega := \{x \in \widetilde{E}_\phi : \overrightarrow{T}(x) \in \widetilde{E}_\psi\}.$$

By (Cordero-Erausquin et al., 2001, Claim 4.4), $\mu(\Omega) = 1$.

We now proceed to our proof of Theorem 5.3.

*Proof of Theorem 5.3.* Fix $x \in \Omega$ and set

$$y^\star := \overrightarrow{T}(x).$$

Let $\varepsilon \in (0, 1)$, and let $\psi_\varepsilon \in C(\mathcal{M})$ satisfy

$$\|\psi_\varepsilon - \psi\|_\infty \leq \varepsilon.$$

Define the unperturbed and perturbed objectives

$$F_x(y) := \tfrac{1}{2}d(x,y)^2 - \psi(y), \qquad F_{\varepsilon,x}(y) := \tfrac{1}{2}d(x,y)^2 - \psi_\varepsilon(y), \qquad y \in \mathcal{M},$$

and choose any minimizer

$$y_\varepsilon^\star \in \arg\min_{y \in \mathcal{M}} F_{\varepsilon,x}(y).$$

By definition

$$y_\varepsilon^\star = T_\varepsilon(x)$$

for some admissible selection $T_\varepsilon$.

**Step 1: $y^\star$ is the unique minimizer of $F_x$.** Since $x \in \Omega \subset \widetilde{E}_\phi \subset E_\phi$, the potential $\phi$ admits a Hessian at $x$ and in particular is differentiable at $x$. By Lemma H.16, at differentiability points of $\phi$, $y^\star = \overrightarrow{T}(x)$ if and only if $y^\star$ minimizes (28). Since $\phi(x)$ is constant in $y$, minimizing H.16 is equivalent to minimizing $y \mapsto c(x, y) - \psi(y) = F_x(y)$. Moreover, Lemma H.16 implies any minimizer must coincide with $\overrightarrow{T}(x)$, hence $y^\star$ is the unique minimizer.

**Step 2: Strict positivity of** $\mathrm{Hess}F_x(y^\star)$ **and definition of** $C(x)$**.** Because $x \in \Omega$, we have $y^\star = \overrightarrow{T}(x) \in \widetilde{E}_\psi$. In particular, $y^\star \in E_\psi$, so $\psi$ is differentiable at $y^\star$ and $\overrightarrow{T}(y^\star)$ is well-defined. Since $y^\star \in \partial^c\phi(x)$ (by Step 1), the $c$-subdifferential symmetry $y \in \partial^c\phi(x) \iff x \in \partial^c\psi(y)$ from (Cordero-Erausquin et al., 2001, Sec. 3.1) yields $x \in \partial^c\psi(y^\star)$; as $\psi$ is differentiable at $y^\star$, Lemma H.16 implies that $\partial^c\psi(y^\star) = \{\overleftarrow{T}(y^\star)\}$, hence

$$\overleftarrow{T}(y^\star) = x.$$

By the definition of $\widetilde{E}_\psi$ in (30), this implies that the Aleksandrov–Bangert Hessian of

$$y \longmapsto \tfrac{1}{2}d\big(\overleftarrow{T}(y^\star), y\big)^2 - \psi(y) = \tfrac{1}{2}d(x, y)^2 - \psi(y) = F_x(y)$$

exists at $y^\star$ and is positive definite. Therefore there exists $C(x) > 0$ such that

$$\langle \mathrm{Hess}F_x(y^\star)u, u \rangle \ge 2C(x)\|u\|^2, \qquad \forall u \in T_{y^\star}\mathcal{M}.$$

**Step 3: Local quadratic growth for** $F_x$ **near** $y^\star$**.** Since $x \in \Omega$ and $y^\star = \overrightarrow{T}(x) \in \widetilde{E}_\psi$, the function $F_x(y) = \tfrac{1}{2}d(x, y)^2 - \psi(y)$ admits a (Aleksandrov–Bangert) Hessian at $y^\star$ in the sense of (29). In particular, $F_x$ is differentiable at $y^\star$ and (29) implies the second-order expansion: as $u \to 0$ in $T_{y^\star}\mathcal{M}$,

$$F_x(\exp_{y^\star}(u)) = F_x(y^\star) + \langle \nabla F_x(y^\star), u \rangle + \frac{1}{2}\langle \mathrm{Hess}F_x(y^\star)u, u \rangle + o(\|u\|^2). \tag{31}$$

Because $y^\star$ is a minimizer of $F_x$ (Step 1) and $F_x$ is differentiable at $y^\star$, we have $\nabla F_x(y^\star) = 0$, hence (31) becomes

$$F_x(\exp_{y^\star}(u)) - F_x(y^\star) = \frac{1}{2}\langle \mathrm{Hess}F_x(y^\star)u, u \rangle + o(\|u\|^2). \tag{32}$$

Moreover, strict positivity of the Hessian (Step 2) provides $C(x) > 0$ such that

$$\frac{1}{2}\langle \mathrm{Hess}F_x(y^\star)u, u \rangle \ge C(x)\|u\|^2 \qquad \forall u \in T_{y^\star}\mathcal{M}. \tag{33}$$

Combining (32)–(33) yields

$$F_x(\exp_{y^\star}(u)) - F_x(y^\star) \ge C(x)\|u\|^2 + o(\|u\|^2). \tag{34}$$

By definition of the Landau symbol $o(\|u\|^2)$, we have $o(\|u\|^2)/\|u\|^2 \to 0$ as $u \to 0$. Therefore, taking $\eta := C(x)/2 > 0$, there exists $\rho_x > 0$ such that whenever $0 < \|u\| < \rho_x$,

$$\left|\frac{o(\|u\|^2)}{\|u\|^2}\right| \le \frac{C(x)}{2} \qquad \Longrightarrow \qquad o(\|u\|^2) \ge -\frac{C(x)}{2}\|u\|^2. \tag{35}$$

Insert (35) into (34) to obtain, for all $\|u\| < \rho_x$,

$$F_x(\exp_{y^\star}(u)) - F_x(y^\star) \ge C(x)\|u\|^2 - \frac{C(x)}{2}\|u\|^2 = \frac{C(x)}{2}\|u\|^2.$$

Choose $r_x > 0$ such that

$$0 < r_x < \min\{\rho_x, \mathrm{inj}(y^\star)\},$$

where $\mathrm{inj}(y^\star)$ is the injectivity radius at $y^\star$. Then, for every $y \in B_{r_x}(y^\star)$ there exists a unique $u \in T_{y^\star}\mathcal{M}$ with $\|u\| < r_x$ such that $y = \exp_{y^\star}(u)$. Moreover, since $\|u\| < \mathrm{inj}(y^\star)$, the geodesic $\gamma(t) := \exp_{y^\star}(tu)$, $t \in [0, 1]$, is the unique minimizing geodesic from $y^\star$ to $y$; it has constant speed $\|\dot\gamma(t)\| = \|u\|$, hence length $L(\gamma) = \int_0^1 \|\dot\gamma(t)\|\, dt = \|u\|$. Therefore

$$d(y, y^\star) = L(\gamma) = \|u\|. \tag{36}$$

Using $y = \exp_{y^\star}(u)$ and (36) we arrive at the local quadratic growth: for all $y \in B_{r_x}(y^\star)$,

$$F_x(y) \ge F_x(y^\star) + \frac{C(x)}{2}\, d(y, y^\star)^2. \tag{37}$$

**Step 4: Uniform perturbation bound.** From $\|\psi_\varepsilon - \psi\|_\infty \le \varepsilon$ we immediately get, for all $y \in \mathcal{M}$,

$$F_x(y) - \varepsilon \ \le \ F_{\varepsilon,x}(y) \ \le \ F_x(y) + \varepsilon. \tag{38}$$

**Step 5: Objective gap at the perturbed minimizer.** By minimality of $y_\varepsilon^\star$ for $F_{\varepsilon,x}$, we have $F_{\varepsilon,x}(y_\varepsilon^\star) \le F_{\varepsilon,x}(y^\star)$. Using (38) at $y_\varepsilon^\star$ and $y^\star$ gives

$$F_x(y_\varepsilon^\star) - F_x(y^\star) \le 2\varepsilon. \tag{39}$$

**Step 6: Localization of $y_\varepsilon^\star$ near $y^\star$.** By Step 1, $y^\star$ is the unique minimizer of the continuous function $F_x$ on the compact manifold $\mathcal{M}$. Therefore, for the radius $r_x$ fixed in Step 3, the minimum value of $F_x - F_x(y^\star)$ on the compact set $\mathcal{M} \setminus B_{r_x}(y^\star)$ is attained and is strictly positive:

$$\delta_x := \min_{y \in \mathcal{M} \setminus B_{r_x}(y^\star)} \big( F_x(y) - F_x(y^\star) \big) \ > \ 0.$$

If $y_\varepsilon^\star \notin B_{r_x}(y^\star)$ then $F_x(y_\varepsilon^\star) - F_x(y^\star) \ge \delta_x$, contradicting (39) whenever $2\varepsilon < \delta_x$. Hence, setting $\varepsilon_x := \delta_x/4$, we obtain we conclude that if $\varepsilon \le \varepsilon_x$, then

$$y_\varepsilon^\star \in B_{r_x}(y^\star).$$

**Step 7: Conclude the distance bound.** Assume $\varepsilon \in (0, \min\{1, \varepsilon_x\})$, then $y_\varepsilon^\star \in B_{r_x}(y^\star)$. Applying the local quadratic growth (37) at $y = y_\varepsilon^\star$ and combining with (39) yields

$$\frac{C(x)}{2} \, d(y_\varepsilon^\star, y^\star)^2 \le F_x(y_\varepsilon^\star) - F_x(y^\star) \le 2\varepsilon,$$

and therefore

$$d(y_\varepsilon^\star, y^\star) \le 2 \sqrt{\frac{\varepsilon}{C(x)}}.$$

Notice that $y^\star = \overrightarrow{T}(x)$ and $y_\varepsilon^\star = T_\varepsilon(x)$, which proves our claim after identifying $\overrightarrow{T}$ with $T_\star$. $\qquad \square$

### H.7. Proof of Lemma H.3

*Proof.* Set $D := \operatorname{diam}(\mathcal{M}) < \infty$. Then $d(x, x_0) \le D$ for all $x \in \mathcal{M}$, hence

$$\int_{\mathcal{M}} d(x, x_0)^{r+\delta} \, \mathrm{d}\mu(x) \le \int_{\mathcal{M}} D^{r+\delta} \, \mathrm{d}\mu(x) = D^{r+\delta} < \infty,$$

where we used that $\mu$ is a probability measure.

It remains to control the second term in (22). Recall that for $\rho \ge 0$,

$$A_{x_0}(\rho) := \sup_{v \in \mathbb{S}_\rho^{p-1}} \ \sup_{\substack{w \in T_v \mathbb{S}_\rho^{p-1} \\ |w|_{x_0} = \rho}} \big\| d_v \exp_{x_0}[w] \big\|_{\exp_{x_0}(v)}.$$

**Step 1: Continuity of the integrand** $(v, w) \mapsto \|d_v \exp_{x_0}[w]\|_{\exp_{x_0}(v)}$. Since $\mathcal{M}$ is compact, it is complete as a metric space. By Hopf–Rinow, $\mathcal{M}$ is then geodesically complete; in particular, for every $v \in T_{x_0}\mathcal{M}$ the geodesic $\gamma_v$ with $\gamma_v(0) = x_0$ and $\dot{\gamma}_v(0) = v$ is defined at least up to time $t = 1$, and

$$\exp_{x_0}(v) = \gamma_v(1)$$

is well-defined for all $v \in T_{x_0}\mathcal{M}$ (Lee, 2018, Cor. 6.22). Moreover, $\exp_{x_0}$ is a smooth map (Lee, 2018, Prop. 5.19). Viewing $T_{x_0}\mathcal{M} \simeq \mathbb{R}^p$ as a smooth manifold, smoothness implies that the differential

$$d_v \exp_{x_0} : \ T_v(T_{x_0}\mathcal{M}) \simeq T_{x_0}\mathcal{M} \ \longrightarrow \ T_{\exp_{x_0}(v)}\mathcal{M}$$

depends smoothly on $v$, and therefore the associated evaluation map

$$T_{x_0}\mathcal{M} \times T_{x_0}\mathcal{M} \to T\mathcal{M}, \qquad (v, w) \longmapsto d_v \exp_{x_0}[w],$$

is smooth (hence continuous) (Lee, 2012, Ex. 3.4).

Next, because a Riemannian metric is a smooth field of inner products (Lee, 2018, Sec. 2), the induced norm on the tangent bundle depends continuously on the base point: in local coordinates $(U, \varphi)$, writing $\xi = \sum_i \xi^i \partial_i|_x \in T_p\mathcal{M}$ one has

$$\|\xi\|_x^2 = g_{ij}(x) \xi^i \xi^j,$$

and the coefficients $x \mapsto g_{ij}(x)$ are smooth, hence continuous. It follows that the map

$$(x, \xi) \longmapsto \|\xi\|_x \qquad (x \in \mathcal{M}, \ \xi \in T_x\mathcal{M})$$

is continuous on $T\mathcal{M}$. Combining this with the continuity of $(v, w) \mapsto d(\exp_{x_0})_v[w]$ shows that

$$F : T_{x_0}\mathcal{M} \times T_{x_0}\mathcal{M} \to \mathbb{R}, \qquad F(v, w) := \left\|d_v \exp_{x_0}[w]\right\|_{\exp_{x_0}(v)},$$

is continuous.

**Step 2: A uniform bound for $A_{x_0}(\rho)$ on $\rho \in [0, D]$.** Work in the Euclidean space $(T_{x_0}\mathcal{M}, g_{x_0})$. The sphere $\mathbb{S}_\rho^{p-1} = \{u : \|u\|_{x_0} = \rho\}$ is the level set of the smooth function $u \mapsto \|u\|_{x_0}^2$, hence it is a smooth hypersurface for $\rho > 0$. Its tangent space at $v \in \mathbb{S}_\rho^{p-1}$ is

$$T_v\mathbb{S}_\rho^{p-1} = \{w \in T_{x_0}\mathcal{M} : \langle v, w \rangle_{x_0} = 0\},$$

because differentiating $t \mapsto \|v + tw\|_{x_0}^2$ at $t = 0$ gives $\frac{d}{dt}|_{t=0}\|v + tw\|_{x_0}^2 = 2\langle v, w \rangle_{x_0}$ (Lee, 2012, Chap. 5).

Fix $\rho \in [0, D]$ and consider any admissible pair $(v, w)$ in the definition of $A_{x_0}(\rho)$. By definition, $v \in \mathbb{S}_\rho^{p-1}$ and $\|w\|_{x_0} = \rho$, hence

$$\|v\|_{x_0} = \rho \leq D, \qquad \|w\|_{x_0} = \rho \leq D.$$

Moreover $w \in T_v\mathbb{S}_\rho^{p-1}$, which is equivalent to $\langle v, w \rangle_{x_0} = 0$. Therefore every admissible $(v, w)$ belongs to

$$E := \Big\{(v, w) \in T_{x_0}\mathcal{M} \times T_{x_0}\mathcal{M} : \|v\|_{x_0} \leq D, \ \|w\|_{x_0} \leq D, \ \langle v, w \rangle_{x_0} = 0\Big\}.$$

We claim that $E$ is compact. Indeed, the sets $\{(v, w) : \|v\|_{x_0} \leq D\}$ and $\{(v, w) : \|w\|_{x_0} \leq D\}$ are closed balls in the finite-dimensional space $T_{x_0}\mathcal{M} \times T_{x_0}\mathcal{M} \simeq \mathbb{R}^{2p}$, hence compact. The constraint $\langle v, w \rangle_{x_0} = 0$ defines a closed subset because $(v, w) \mapsto \langle v, w \rangle_{x_0}$ is continuous and $\{0\}$ is closed. Thus $E$ is a closed subset of a compact set, hence compact.

Since $F$ is continuous and $E$ is compact, $F$ attains a maximum on $E$ by the Extreme Value Theorem. Define

$$M := \max_{(v,w) \in E} F(v, w) < \infty.$$

Because the admissible set for $A_{x_0}(\rho)$ is contained in $E$, we obtain for every $\rho \in [0, D]$,

$$A_{x_0}(\rho) = \sup_{\text{admissible } (v,w)} F(v, w) \leq \sup_{(v,w) \in E} F(v, w) = M.$$

**Step 3: Integrability of $A_{x_0}(d(x, x_0))^r$.** For any $x \in \mathcal{M}$ we have $d(x, x_0) \leq D$, hence $d(x, x_0) \in [0, D]$ and therefore

$$A_{x_0}(d(x, x_0)) \leq M \qquad \Longrightarrow \qquad A_{x_0}(d(x, x_0))^r \leq M^r.$$

Integrating against the probability measure $\mu$ yields

$$\int_\mathcal{M} A_{x_0}(d(x, x_0))^r \, \mathrm{d}\mu(x) \leq \int_\mathcal{M} M^r \, \mathrm{d}\mu(x) = M^r < \infty.$$

Combining this with the bound on the first term proves (22). □

## H.8. Proof of Lemma H.4

*Proof.* Let $\alpha := \mathrm{supp}(\eta)$. By assumption, $\alpha$ is finite and $|\alpha| \leq m$. By definition of the 2-Wasserstein distance,

$$W_2(\rho, \eta)^2 = \inf_{\gamma \in \Pi(\rho, \eta)} \int_{\mathcal{M} \times \mathcal{M}} d(y, z)^2 \, \mathrm{d}\gamma(y, z),$$

where $\Pi(\rho, \eta)$ denotes the set of couplings of $\rho$ and $\eta$.

**Step 1: Disintegrate an arbitrary coupling.** Fix any $\gamma \in \Pi(\rho, \eta)$. Since $\mathcal{M}$ is a complete separable metric space, we can apply the disintegration theorem (see, e.g., Figalli (2010, Theorem 1.4)) to $\gamma$ with respect to the projection $\mathrm{pr}_1 : \mathcal{M} \times \mathcal{M} \to \mathcal{M}$. Because $(\mathrm{pr}_1)_{\#}\gamma = \rho$, there exists a $\rho$-measurable family of probability measures $\{\gamma_y\}_{y \in \mathcal{M}} \subset \mathcal{P}(\mathcal{M})$ such that

$$\gamma(\mathrm{d}y, \mathrm{d}z) = \rho(\mathrm{d}y)\, \gamma_y(\mathrm{d}z), \tag{40}$$

i.e. for every bounded Borel function $\varphi : \mathcal{M} \times \mathcal{M} \to \mathbb{R}$,

$$\int_{\mathcal{M} \times \mathcal{M}} \varphi(y, z) \, \mathrm{d}\gamma(y, z) = \int_{\mathcal{M}} \left( \int_{\mathcal{M}} \varphi(y, z) \, \mathrm{d}\gamma_y(z) \right) \mathrm{d}\rho(y). \tag{41}$$

Since $\mathcal{M}$ is compact, the function $(y, z) \mapsto d(y, z)^2$ is bounded on $\mathcal{M} \times \mathcal{M}$. Therefore we may take $\varphi(y, z) = d(y, z)^2$ in (41) to obtain

$$\int_{\mathcal{M} \times \mathcal{M}} d(y, z)^2 \, \mathrm{d}\gamma(y, z) = \int_{\mathcal{M}} \left( \int_{\mathcal{M}} d(y, z)^2 \, \mathrm{d}\gamma_y(z) \right) \mathrm{d}\rho(y). \tag{42}$$

**Step 2: Use that $\gamma_y$ is supported on $\alpha$.** Because $\gamma \in \Pi(\rho, \eta)$ has second marginal $\eta$ and $\eta$ is supported on $\alpha$, we must have Because $\gamma \in \Pi(\rho, \eta)$, its second marginal is $\eta$, i.e.

$$(\mathrm{pr}_2)_{\#}\gamma = \eta,$$

where $\mathrm{pr}_2 : \mathcal{M} \times \mathcal{M} \to \mathcal{M}$ denotes the second projection $\mathrm{pr}_2(y, z) = z$. Hence, by the definition of pushforward measures, for any Borel set $B \subset \mathcal{M}$,

$$\eta(B) = ((\mathrm{pr}_2)_{\#}\gamma)(B) = \gamma(\mathrm{pr}_2^{-1}(B)).$$

Taking $B = \mathcal{M} \setminus \alpha$ and using that $\eta$ is supported on $\alpha$ (so $\eta(\mathcal{M} \setminus \alpha) = 0$), we obtain

$$\gamma\big(\mathrm{pr}_2^{-1}(\mathcal{M} \setminus \alpha)\big) = 0.$$

Finally, $\mathrm{pr}_2^{-1}(\mathcal{M} \setminus \alpha) = \mathcal{M} \times (\mathcal{M} \setminus \alpha)$, so

$$\gamma(\mathcal{M} \times (\mathcal{M} \setminus \alpha)) = \eta(\mathcal{M} \setminus \alpha) = 0.$$

Disintegrating this identity using (40) gives

$$0 = \gamma(\mathcal{M} \times (\mathcal{M} \setminus \alpha)) = \int_{\mathcal{M}} \gamma_y(\mathcal{M} \setminus \alpha) \, \mathrm{d}\rho(y),$$

hence $\gamma_y(\mathcal{M} \setminus \alpha) = 0$ for $\rho$-a.e. $y$. Equivalently,

$$\mathrm{supp}(\gamma_y) \subseteq \alpha \qquad \text{for } \rho\text{-a.e. } y.$$

**Step 3: Lower bound the conditional cost by the nearest-site distance.** Fix $y$ such that $\mathrm{supp}(\gamma_y) \subseteq \alpha$. Then, since $\gamma_y$ is a probability measure supported on $\alpha$,

$$\int_{\mathcal{M}} d(y, z)^2 \, \mathrm{d}\gamma_y(z) = \int_{\alpha} d(y, z)^2 \, \mathrm{d}\gamma_y(z) \geq \min_{a \in \alpha} d(y, a)^2 \int_{\alpha} \mathrm{d}\gamma_y(z) = \min_{a \in \alpha} d(y, a)^2.$$

Insert this into (42) to obtain

$$\int_{\mathcal{M} \times \mathcal{M}} d(y, z)^2 \, \mathrm{d}\gamma(y, z) \geq \int_{\mathcal{M}} \min_{a \in \alpha} d(y, a)^2 \, \mathrm{d}\rho(y). \tag{43}$$

**Step 4: Take the infimum over couplings and compare to $V_{m,2}(\rho)$.** Since (43) holds for every $\gamma \in \Pi(\rho, \eta)$, taking the infimum over $\gamma$ gives

$$W_2(\rho, \eta)^2 = \inf_{\gamma \in \Pi(\rho, \eta)} \int d(y, z)^2 \, \mathrm{d}\gamma \geq \int_{\mathcal{M}} \min_{a \in \alpha} d(y, a)^2 \, \mathrm{d}\rho(y).$$

Finally, by Definition H.1 of $V_{m,2}$ as the infimum over all sets of size at most $m$, and since $|\alpha| \leq m$, we have

$$\int_{\mathcal{M}} \min_{a \in \alpha} d(y, a)^2 \, \mathrm{d}\rho(y) \ \geq \ V_{m,2}(\rho).$$

Combining the last two displays proves the claim. $\qquad\square$

## H.9. Proof of Lemma H.5

*Proof.* If $\mathcal{M}$ is compact then every probability measure has finite second moment, hence $\nu_t, \nu_s \in \mathcal{P}_2(\mathcal{M})$ and $W_2(\nu_t, \nu_s)$ is well-defined.

Define the measurable map $(t, s) : \mathcal{M} \to \mathcal{M} \times \mathcal{M}$ by $x \mapsto (t(x), s(x))$ and set

$$\pi := (t, s)_{\#}\mu.$$

Let $\mathrm{pr}_1, \mathrm{pr}_2 : \mathcal{M} \times \mathcal{M} \to \mathcal{M}$ be the coordinate projections defined in Appendix B. Since $\mathrm{pr}_1 \circ (t, s) = t$ and $\mathrm{pr}_2 \circ (t, s) = s$, the composition rule for pushforwards (Teschl, 2025, §2.3, Example 2.12) yields

$$(\mathrm{pr}_1)_{\#}\pi = (\mathrm{pr}_1)_{\#}(t, s)_{\#}\mu = (\mathrm{pr}_1 \circ (t, s))_{\#}\mu = t_{\#}\mu = \nu_t, \qquad (\mathrm{pr}_2)_{\#}\pi = \nu_s.$$

Hence $\pi \in \Pi(\nu_t, \nu_s)$. By the definition of $W_2$,

$$W_2(\nu_t, \nu_s)^2 = \inf_{\gamma \in \Pi(\nu_t, \nu_s)} \int_{\mathcal{M} \times \mathcal{M}} d(y, z)^2 \, \mathrm{d}\gamma(y, z) \leq \int_{\mathcal{M} \times \mathcal{M}} d(y, z)^2 \, \mathrm{d}\pi(y, z).$$

Since $\mathcal{M}$ is compact, $g(y, z) = d(y, z)^2$ is bounded, hence integrable with respect to any probability measure on $\mathcal{M} \times \mathcal{M}$. Thus we can apply the change-of-variables formula for pushforwards (Teschl, 2025, Theorem 2.16),

$$\int_{\mathcal{M} \times \mathcal{M}} d(y, z)^2 \, \mathrm{d}\pi(y, z) = \int_{\mathcal{M}} d\big(t(x), s(x)\big)^2 \, \mathrm{d}\mu(x).$$

Taking square roots gives (23). $\qquad\square$

## H.10. Proof of Lemma H.6

**Lemma H.18** (Stability of minimizer)**.** *Let $K$ be a compact metric space and let $f_k, f : K \to \mathbb{R}$ be continuous with $f_k \to f$ uniformly. Suppose $f$ has a unique minimizer $z^\star \in K$. If for each $k$, $z_k \in \arg\min_K f_k$, then $z_k \to z^\star$.*

The proof of Lemma H.18 is postponed to Appendix H.11. We are now ready to prove Lemma H.6.

*Proof of Lemma H.6.* Let $\phi \in \Psi_c(\mathcal{M})$ and set $\psi := \phi^c \in C(\mathcal{M}, \mathbb{R})$, so that $\phi = \psi^c$.

**Step 1: Build $\psi_k \in \varphi^* \mathcal{F}$ approximating $\psi = \phi^c$.** Since $\mathcal{F}$ is dense in $C(\mathbb{R}^n, \mathbb{R})$ in the ucc topology and $\varphi$ satisfies Assumption 2.2, it follows (e.g. (Kratsios & Bilokopytov, 2020, Theorem 3.3)) that the pullback class $\varphi^* \mathcal{F}$ is dense in $C(\mathcal{M}, \mathbb{R})$ for uniform convergence. Hence there exists a sequence $\psi_k \in \varphi^* \mathcal{F}$ such that

$$\|\psi_k - \psi\|_\infty \xrightarrow[k \to \infty]{} 0.$$

Define $\phi_k := \psi_k^c \in \mathfrak{C}(\varphi^* \mathcal{F})$.

**Step 2: Uniform convergence $\phi_k \to \phi$.** For any $x \in \mathcal{M}$ and any $y \in \mathcal{M}$, we have $\psi(y) - \|\psi_k - \psi\|_\infty \leq \psi_k(y) \leq \psi(y) + \|\psi_k - \psi\|_\infty$, hence

$$c(x, y) - \psi(y) - \|\psi_k - \psi\|_\infty \leq c(x, y) - \psi_k(y) \leq c(x, y) - \psi(y) + \|\psi_k - \psi\|_\infty.$$

Taking $\inf_{y \in \mathcal{M}}$ yields

$$\phi(x) - \|\psi_k - \psi\|_\infty \leq \phi_k(x) \leq \phi(x) + \|\psi_k - \psi\|_\infty,$$

and therefore

$$\|\phi_k - \phi\|_\infty \leq \|\psi_k - \psi\|_\infty \xrightarrow[k \to \infty]{} 0.$$

**Step 3: Differentiability and gradient representation on a full $\mu$-measure set.** Each $\phi_k$ and $\phi$ is $c$-concave, hence Lipschitz on $\mathcal{M}$ (McCann, 2001, Lemma 2). By Rademacher's theorem on Riemannian manifolds (McCann, 2001, Lemma 4), $\phi$ and each $\phi_k$ are differentiable $\text{vol}_\mathcal{M}$-a.e., hence also $\mu$-a.e. since $\mu \ll \text{vol}_\mathcal{M}$. Denote by $\text{Diff}(\phi) \subset \mathcal{M}$ and $\text{Diff}(\phi_k) \subset \mathcal{M}$ the set of points where $\phi$ and $\phi_k$ is differentiable. Thus, $\mu(\text{Diff}(\phi)) = 1$ and $\mu(\text{Diff}(\phi_k)) = 1$ for every $k$.

Moreover, for the quadratic cost $c(x, y) = \frac{1}{2}d(x, y)^2$, McCann (2001, Lemma 7) shows that at every point $x \in \text{Diff}(\phi)$, the $c$-subdifferential $\partial^c \phi(x)$ is a singleton. Equivalently, the minimizer

$$y^\star(x) \in \arg\min_{y \in \mathcal{M}} \big(c(x, y) - \psi(y)\big)$$

is unique, and the gradient satisfies the first-order condition

$$\nabla \phi(x) = -\nabla_x c\big(x, y^\star(x)\big) = -\log_x\big(y^\star(x)\big).$$

The same statement holds for each $\phi_k$: for any $x \in \text{Diff}(\phi_k)$, the minimizer $y_k(x) \in \arg\min_{y \in \mathcal{M}} \big(c(x, y) - \psi_k(y)\big)$ is unique and

$$\nabla \phi_k(x) = -\nabla_x c\big(x, y_k(x)\big) = -\log_x\big(y_k(x)\big).$$

We define

$$N_\star := \text{Diff}(\phi) \cap \bigcap_{k=1}^\infty \text{Diff}(\phi_k).$$

Then $N_\star$ is measurable and satisfies $\mu(N_\star) = 1$, since it is a countable intersection of measurable sets of full $\mu$-measure. In particular, for every $x \in N_\star$ all gradients $\nabla \phi(x)$ and $\nabla \phi_k(x)$ are well-defined, and the unique minimizers $y^\star(x)$ and $y_k(x)$ satisfy

$$\nabla \phi(x) = -\log_x\big(y^\star(x)\big), \qquad \nabla \phi_k(x) = -\log_x\big(y_k(x)\big).$$

Fix $x \in N_\star$ for the remainder of the proof.

**Step 5: Stability of minimizers $y_k(x) \to y^\star(x)$.** Define continuous functions on the compact set $\mathcal{M}$ by

$$h_x(y) := c(x, y) - \psi(y), \qquad h_{k,x}(y) := c(x, y) - \psi_k(y).$$

Then $y^\star(x) = \arg\min h_x$ and $y_k(x) = \arg\min h_{k,x}$ are unique by construction of $N_\star$. Moreover,

$$\sup_{y \in \mathcal{M}} |h_{k,x}(y) - h_x(y)| = \sup_{y \in \mathcal{M}} |\psi_k(y) - \psi(y)| = \|\psi_k - \psi\|_\infty \to 0.$$

Since $h_{k,x} \to h_x$ uniformly on the compact space $\mathcal{M}$ and $h_x$ has a unique minimizer, Lemma H.18 implies

$$y_k(x) \to y^\star(x).$$

**Step 6: Convergence of gradients.** By Step 3 we have the identity

$$\nabla\phi(x) = -\nabla_x c\big(x, y^\star(x)\big) = -\log_x\big(y^\star(x)\big), \qquad c(x, y) = \tfrac{1}{2}d(x, y)^2.$$

In particular, the Riemannian logarithm map is well-defined at $y^\star(x)$.

Therefore, $y^\star(x) \notin \mathrm{Cut}(x)$. Since $\mathcal{M} \setminus \mathrm{Cut}(x)$ is open and $\log_x$ is smooth on it, there exists an open neighborhood $V_x \subset \mathcal{M} \setminus \mathrm{Cut}(x)$ of $y^\star(x)$ on which $\log_x$ is smooth. Equivalently, there exists $r_x > 0$ such that

$$B_{r_x}\big(y^\star(x)\big) \subset V_x \subset \mathcal{M} \setminus \mathrm{Cut}(x).$$

From Step 5 we already know that $y_k(x) \to y^\star(x)$ in $\mathcal{M}$. Hence, by the definition of convergence in a metric space, for the radius $r_x$ above there exists $k_0(x)$ such that for all $k \geq k_0(x)$,

$$d\big(y_k(x), y^\star(x)\big) < r_x, \qquad \text{so that} \qquad y_k(x) \in B_{r_x}\big(y^\star(x)\big) \subset V_x.$$

In particular, for all $k \geq k_0(x)$ the points $y_k(x)$ lie in a region where $\log_x$ is smooth (hence continuous). Therefore,

$$\log_x\big(y_k(x)\big) \longrightarrow \log_x\big(y^\star(x)\big),$$

and using the gradient representations,

$$\nabla\phi_k(x) = -\log_x\big(y_k(x)\big) \longrightarrow -\log_x\big(y^\star(x)\big) = \nabla\phi(x).$$

Since $x \in N_\star$ was arbitrary and $\mu(N_\star) = 1$, the convergence holds for $\mu$-a.e. $x \in \mathcal{M}$, which completes the proof.

$\square$

## H.11. Proof of Lemma H.18

*Proof.* We prove the desired result by contradiction. Specifically, suppose that $z_k \to z^\star$ is not true. Then

$$\exists \varepsilon_0 > 0 \quad \text{s.t.} \quad \forall N \geq \mathbb{N}, \exists n \geq N \quad \text{s.t.} \quad d(z_n, z^\star) \geq \varepsilon. \tag{44}$$

We first show that we can extract a subsequence out of $(z_k)$ that always stays at least $\varepsilon_0$ away from $z^\star$. Then, via compactness we show that such subsequence admits a subsequence converging to a point $\bar{z}$ different from $z^\star$.

**Step 1: Constructing a convergent subsequence.** Let $N_1 := 1$. Then, by (44) there exists $k_1 \geq N_1$ such that

$$d(z_{k_1}, z^\star) \geq \varepsilon_0. \tag{45}$$

Let $N_2 := k_1 + 1$. Again, there exists $k_2 \geq N_2$ such that

$$d(z_{k_2}, z^\star) \geq \varepsilon_0.$$

We continue inductively: having chosen $k_j$, set $N_{j+1} := k_j + 1$. Then there exists $k_{j+1} \geq N_{j+1}$ such that

$$d(z_{k_{j+1}}, z^\star) \geq \varepsilon_0.$$

By construction the indices are strictly increasing $k_1 < k_2 < k_3 < \ldots$ so $(z_{k_j})$ is a subsequence of $(z_k)$. Furthermore, each term in the subsequence stays away from $z^\star$ by at least $\varepsilon_0$, i.e.

$$d(z_{k_j}, z^\star) \geq \varepsilon_0, \quad \text{for all } j.$$

Now $(z_{k_j})$ is a sequence in $K$, and $K$ is compact. Therefore, there exists a further subsequence of $(z_{k_j})$ (which, with a slight abuse of notation, we still denote by $(z_{k_j})$) and a point $\bar{z} \in K$ such that

$$z_{k_j} \to \bar{z} \quad \text{as} \quad j \to \infty.$$

Since every term in the subsequence satisfied $d(z_{k_j}, z^\star) \geq \varepsilon_0$, then it follows that $\bar{z} \neq z^\star$, otherwise the distances would converge to zero.

**Step 2: Comparing $f(\bar{z})$ and $f(z^\star)$.** Since uniform convergence implies pointwise convergence, we have that for any fixed point $z \in K$,

$$f_k(z) \to f(z).$$

Since every subsequence of a convergent sequence is itself convergent and has the same limit as the original,

$$f_{k_j}(z^\star) \to f(z^\star).$$

On the other hand, consider the following inequality

$$|f_{k_j}(z_{k_j}) - f(\bar{z})| \leq |f_{k_j}(z_{k_j}) - f(z_{k_j})| + |f(z_{k_j}) - f(\bar{z})|.$$

The first term goes to zero by uniform convergence while the second term goes to zero by the continuity of $f$ and $z_{k_j} \to \bar{z}$. Thus,

$$f_{k_j}(z_{k_j}) \to f(\bar{z}). \tag{46}$$

Since each $z_{k_j}$ minimizes $f_{k_j}$, then

$$f_{k_j}(z_{k_j}) \leq f_{k_j}(z^\star) \quad \text{for all } j. \tag{47}$$

Now we take the limit as $j \to \infty$. By (46), the left hand side gives

$$f_{k_j}(z_{k_j}) \to f(\bar{z}).$$

By uniform convergence the right hand side gives

$$f_{k_j}(z^\star) \to f(z^\star).$$

Since $f_{k_j}(z_{k_j}) \leq f_{k_j}(z^\star)$ for all $j$, and $f_{k_j}(z_{k_j}) \to f(\bar{z})$ and $f_{k_j}(z^\star) \to f(z^\star)$ then

$$f(\bar{z}) \leq f(z^\star). \tag{48}$$

**Step 3: Reaching a contradiction.** We know that $z^\star$ is the unique minimizer of $f$ on $K$. Therefore,

$$f(z^\star) \leq f(z) \quad \text{for all } z \in K \tag{49}$$

and if $f(z) = f(z^\star)$ then $z = z^\star$. By minimality of $z^\star$ we have that

$$f(z^\star) \leq f(\bar{z}). \tag{50}$$

Combining (48) and (50) we obtain that

$$f(\bar{z}) = f(z^\star). \tag{51}$$

Therefore, $\bar{z} = z^\star$ by uniqueness of the minimizer. Hence we reach a contradiction: the subsequence $(z_{k_j})$ is always at least $\varepsilon_0$ away from $z^\star$, therefore its limit $\bar{z}$ cannot be equal to $z^\star$. Thus we conclude that the sequence of minimizers $z_k$ must converge to the unique minimizer $z^\star$: $z_k \to z^\star$. $\qquad\square$

### H.12. Proof of Theorem H.9

Our proof is based on Yarotsky & Zhevnerchuk (2020, Theorem 3.3), which we state below for convenience.

**Theorem H.19** (Theorem 3.3 in Yarotsky & Zhevnerchuk (2020))**.** *For any $r > 0$, any rate $\beta \in \left(\frac{r}{n}, \frac{2r}{n}\right]$ can be achieved with deep ReLU networks with $L \leq c_{r,n} W^{\beta n/r - 1}$ layers.*

*Proof of Theorem H.9.* We show how Theorem H.9 follows from Theorem H.19 under the complexity conventions of Definitions H.7-H.8.

**Step 1: The quantitative statement from Yarotsky & Zhevnerchuk (2020).** By Theorem H.19, for every $r > 0$ and every $\beta \in (r/n, 2r/n]$ there exist constants $C_{r,n,\beta} > 0$ and $C_{r,n} > 0$ such that for every integer $W \geq 1$ there exists a

ReLU approximation scheme (consisting of an architecture $\eta_W$ with at most $W$ weights and a parameter selection map $G_W$) satisfying

$$\sup_{f \in \mathcal{H}_{r,n}} \left\| f - \tilde{f}_{\eta_W, G_W}(f) \right\|_\infty \leq C_{r,n,\beta} W^{-\beta},$$

and whose depth (number of layers, i.e. our Definition H.8) satisfies

$$L_W \leq C_{r,n} W^{\beta n/r - 1}.$$

Fix $f \in \mathcal{H}_{r,n}$ and define the corresponding network

$$\hat{f}_W := \tilde{f}_{\eta_W, G_W}(f).$$

Then $\hat{f}_W \in \mathcal{NN}^{\mathrm{ReLU}}_{n, L_W, W}$ and

$$\|f - \hat{f}_W\|_\infty \leq C_{r,n,\beta} W^{-\beta}.$$

**Step 2: Choose $W$ as a function of $\varepsilon$.** For convenience, set

$$C'_{r,n,\beta} := \max\{1, C_{r,n,\beta}\}, \qquad \widetilde{C}_{r,n,\beta} := 2^\beta C'_{r,n,\beta}.$$

Fix $\varepsilon \in (0,1)$ and define the threshold

$$\bar{W}_\varepsilon := \left( \frac{C'_{r,n,\beta}}{\varepsilon} \right)^{1/\beta}.$$

Choose any integer $W_\varepsilon \in \mathbb{N}$ such that $W_\varepsilon \geq \bar{W}_\varepsilon$ (for instance, $W_\varepsilon = \lceil \bar{W}_\varepsilon \rceil$). Since $C_{r,n,\beta} \leq C'_{r,n,\beta}$, we obtain

$$C_{r,n,\beta} W_\varepsilon^{-\beta} \leq C'_{r,n,\beta} W_\varepsilon^{-\beta} \leq C'_{r,n,\beta} \bar{W}_\varepsilon^{-\beta} = \varepsilon.$$

With $\hat{f}_\varepsilon := \hat{f}_{W_\varepsilon}$ we therefore have

$$\|f - \hat{f}_\varepsilon\|_\infty \leq \varepsilon.$$

**Step 3: Weight bound.** Take $W_\varepsilon := \lceil \bar{W}_\varepsilon \rceil$. Since $\varepsilon \in (0,1)$ and $C'_{r,n,\beta} \geq 1$, we have $\bar{W}_\varepsilon \geq 1$, hence $\lceil x \rceil \leq 2x$ for all $x \geq 1$. Therefore,

$$W_\varepsilon \leq 2\bar{W}_\varepsilon = 2\left( \frac{C'_{r,n,\beta}}{\varepsilon} \right)^{1/\beta} = \left( \frac{\widetilde{C}_{r,n,\beta}}{\varepsilon} \right)^{1/\beta}.$$

**Step 4: Depth bound.** From Step 1 and the choice of $W_\varepsilon$,

$$L_\varepsilon := L_{W_\varepsilon} \leq C_{r,n} W_\varepsilon^{\beta n/r - 1}.$$

This is exactly the stated depth estimate in the convention of Definition H.8.

**Step 5: Replacement of ReLU by a general piecewise-linear activation.** Let $\sigma : \mathbb{R} \to \mathbb{R}$ be any non-affine piecewise-linear activation function. By Yarotsky (2017, Prop. 1(b)), on a bounded domain (here $[0,1]^n$) any ReLU neural network $\hat{f}_\varepsilon$ can be converted into a neural network with activation function $\sigma$ that computes the *same* function on $[0,1]^n$. More precisely, there exists a constant $K_\sigma \geq 1$ (depending only on $\sigma$) and a neural network $\hat{f}_\varepsilon^{(\sigma)}$ with activation function $\sigma$ such that

$$\hat{f}_\varepsilon^{(\sigma)}(x) = \hat{f}_\varepsilon(x) \quad \text{for all } x \in [0,1]^n,$$

and whose depth $L_\varepsilon^{(\sigma)}$ and number of parameters $W_\varepsilon^{(\sigma)}$ satisfies

$$L_\varepsilon^{(\sigma)} = L_\varepsilon, \qquad W_\varepsilon^{(\sigma)} \leq K_\sigma W_\varepsilon.$$

In particular,

$$\|f - \hat{f}_\varepsilon^{(\sigma)}\|_\infty = \|f - \hat{f}_\varepsilon\|_\infty \leq \varepsilon.$$

Thus the same approximation statement holds for neural networks with activation function $\sigma$ after adjusting the constant $\widetilde{C}_{r,n,\beta}$ by a factor depending only on $\sigma$ (and without changing the depth order).

$\square$

