# OpenReview forum: "Riemannian Neural Optimal Transport"
_ICML.cc/2026/Conference — ICML 2026 regular_

### Official Review · Reviewer_F1EG · 2026-03-10

**Soundness:** 4
**Presentation:** 3
**Significance:** 3
**Originality:** 3
**Overall Recommendation:** 5
**Confidence:** 3

**Summary:**

This paper studies computational optimal transport on Riemannian manifolds and introduces Riemannian Neural Optimal Transport. The authors show that discrete transport methods such as RCPM suffer from the curse of dimensionality, requiring exponentially many parameters as dimension grows.

RNOT uses landmark distance embeddings and a neural network to parameterize a prepotential, with the c-transform enforcing c-concavity. Theoretically, RNOT achieves approximation guarantees with polynomial parameter complexity and avoids exponential scaling. Empirically, it demonstrates strong high-dimensional stability on spheres, tori, and continental drift datasets.

**Compliance With Llm Reviewing Policy:**

Affirmed.

**Final Justification:**

During the rebuttal phase, the authors provided additional experimental results where RCPMs are trained with the Kantorovich semi-dual loss, establishing a fairer baseline comparison. Furthermore, the inclusion of new computational results on SO(3) and SE(3) further enhances the overall quality and potential impact of this work.

Regarding my initial concern about algorithmic novelty, the authors clarified that their core innovation lies not in the design of the semi-dual objective, but rather in the intrinsic continuous parameterization of manifold Kantorovich potentials. I accept this clarification. Moreover, I appreciate the completeness and solidness of its theoretical analysis, which I consider to be one of the major strength of this manuscript.

In conclusion, the authors have adequately addressed my concerns, and I believe this paper meets the acceptance criteria. Accordingly, I am raising my score from 3 to 5.

**Key Questions For Authors:**

1. The author uses $MLP( \varphi(x) ) $ to parameterize $ \psi(x) $. Have the authors considered directly parameterizing $ \psi(x) $ using the ambient space embedding of the manifold $ M $, thus avoiding the design of $ \varphi(x) $? Specifically: suppose $ M \hookrightarrow \mathbb{R}^n $ is a submanifold embeded in  $\mathbb{R}^n $ , define $ \overline{\text{MLP}}: \mathbb{R}^n \to \mathbb{R} $ and use $ \overline{\text{MLP}} $ to parameterize $ \psi(x) $.

2. Figure 3 shows that RCPM degrades drastically with increasing dimension $ n $, while RNOT performs favorably. The author uses this to support the advantage of continuous, discretization-free transport parameterization. However, the two methods optimize different losses: RNOT uses the Kantorovich semi-dual objective, while RCPM optimizes KL divergence. Could the better performance of RNOT come from the loss formulation rather than the transport parameterization?

3. Compared with existing methods such as RCPM, RNOT appears more promising for manifolds with complex geometry. Has the author evaluated RNOT on more complex manifolds, e.g., the special orthogonal group $ \text{SO}(n) $ or the special Euclidean group $ \text{SE}(3) $?

**Limitations:**

Yes. The authors have adequately discussed the limitations and potential negative societal impact of their work.

**Strengths And Weaknesses:**

Strengths:

1. The paper presents novel and rigorous theoretical foundations with solid mathematical proofs.

2. The paper is well-structured and logically coherent.

Weaknesses:

1. The algorithmic novelty of the proposed method is relatively modest, as it is a brute-force implementation of the Kantorovich semi-dual formulation, leading to heavy computational costs.

2. Training is extremely slow and computationally expensive. RNOT requires inner-loop Riemannian gradient descent for the c-transform, making it 20–30 times slower than RCPM, especially in high dimensions. It would be valuable if the authors could explore potential acceleration strategies for the training process.

3. It would be highly beneficial to include a description of the Kantorovich semi-dual objective (or Algorithm 1) in the main text, as this would significantly aid readers in understanding the proposed algorithm.

---

> ### Author Rebuttal · Authors · 2026-03-31
>
> We thank the reviewer for the assessment.  We address each concern below.
>
> **(1,2) Algorithmic novelty; Training is extremely slow and computationally expensive.** We agree that the current RNOT implementation is slower than RCPM due to the inner-loop optimization for the manifold c-transform. However, we respectfully believe that describing the method as a "brute-force implementation" misses the central contribution. The novelty is not the semi-dual objective per se, but the intrinsic continuous parameterization of manifold Kantorovich potentials, with c-concavity enforced by construction through the manifold c-transform. This continuous formulation allows us to prove a qualitative separation from discrete-output approaches: discrete parameterizations such as RCPM suffer from a curse of dimensionality, whereas the proposed RNOT class achieves polynomial-complexity approximation guarantees. The extra computation is the price of a fundamentally more expressive representation class better aligned with manifold OT geometry.
>
> Comparing only wall-clock time per iteration does not capture the relevant tradeoff. RCPM is faster because it optimizes over a discretized class, but our results show such classes become exponentially inefficient as dimension grows. RNOT incurs a heavier inner step precisely to avoid this exponential barrier, and remains stable in higher-dimensional settings where discrete baselines degrade. We view the runtime not as evidence against the method, but as a natural consequence of targeting a stronger function class with provable high-dimensional advantages.
>
> That said, we agree solver efficiency is an important next step. The current implementation leaves room for acceleration through warm-starting, amortizing the c-transform with an auxiliary network, reducing inner steps early in training, stronger Riemannian optimizers, and exploiting parallel structure. We will add this discussion explicitly.
>
> **(3) Include a description of the Kantorovich semi-dual objective.** See Reviewer F3gs Question (7).
>
> **(4) MLP parameterization.** We considered direct ambient parameterizations $\psi(x)=\mathrm{NN}(i(x))$ in early development. The reason we do not adopt this is that it is fundamentally *extrinsic*: the learned potential depends on the embedding $i:M\rightarrow\mathbb{R}^n$, whereas our goal is to solve OT in the *intrinsic* geometry. On curved manifolds, ambient Euclidean proximity can be a poor proxy for geodesic proximity, so an ambient MLP must implicitly learn to correct embedding distortions. In contrast, the landmark-distance features $d_L(x)$ provide a geometric prior tied to the manifold metric, making the representation geodesic-aware from the start. This design is consistent with prior geometry-aware architectures using intrinsic distances. We will clarify this choice and note that ambient-coordinate networks are a reasonable baseline but not as well aligned with the intrinsic setting.
>
> **(5) Figure 3 shows that RCPM degrades with increasing dimension.** We agree that one should distinguish the effect of transport parameterization from the training objective. As stated in Section 3, one central contribution is a theoretical result showing that the curse-of-dimensionality scaling is a structural property of the discrete class itself, independent of optimization procedure or loss. That said, we agree an empirical control using the same semi-dual objective would strengthen this point. We are currently running additional experiments in which RCPMs are trained with the same Kantorovich semi-dual loss as RNOT, and we expect these results to be available by the end of the discussion period. Our expectation, based on theory, is that the same exponential scaling persists, providing additional evidence that the degradation is due to the discrete parameterization rather than loss choice.
>
> **(6) More complex manifolds such as $SO(n)$ or $SE(n)$.** We agree that $SO(3)$ is a natural benchmark: it is compact and admits geodesic distances and exp/log maps in closed form. We are currently running experiments on $SO(3)$ and plan to include a comparison table in the revision; these results should be available within the discussion period. We are also running experiments on compactly supported measures on $SE(3)$, a natural compromise between our current theory and the fully non-compact case; results should also be available within the discussion period. Since $SE(n)$ is non-compact, the present theory does not directly apply in full generality, but compactly supported measures provide a practically relevant intermediate regime bridging theory and applications. We discuss this scope boundary in Section 7 and identify extensions to non-compact settings as important future work. Our high-dimensional sphere experiments ($S^{10}$ in the main paper, up to $S^{40}$ in the appendix) already rigorously test scaling behavior; $SO(3)$ provides a meaningful additional benchmark with more structured geometry.

---

> > ### Author Rebuttal · Reviewer_F1EG · 2026-04-02
> >
> > Thank you for the detailed response.
> >
> > Regarding the ongoing experiments where RCPMs are trained with the Kantorovich semi-dual loss, I look forward to seeing these results. Such a controlled comparison is essential to demonstrate the performance of RNOT on a fair basis and is, in my view, critical to supporting the paper’s central claims.
> >
> > Furthermore, the authors have committed to providing computational results on SO(3) within the discussion period. Including this would indeed further enhance the quality and impact of the work.
> >
> > **My other concerns have been addressed. I look forward to the authors presenting these numerical results during the remainder of the discussion period.**

---

> > > ### Author Response · Authors · 2026-04-06
> > >
> > > Thank you for the suggestion which allowed us to further showcase the strengths of our contribution, we hope that these experiments have fully resolved the reviewer's remaining open questions.
> > >
> > > **Training RCPM with the Semi-Dual Loss.**
> > > Following the reviewer's suggestion, we investigated training RCPMs with the semi-dual loss. We highlight an important tension: (1) training with the semi-dual loss is only meaningful when the parameterization recovers a true optimal transport potential, and (2) the RCPM implementation of Cohen et al. recovers such a potential only in the limit $\gamma \to 0$, where the $\mathrm{LogSumExp}$ parameterization converges to the hard minimum. We therefore investigate whether the same CoD behavior persists when training RCPMs with the semi-dual loss at small $\gamma$. The table below confirms that it does, consistent with our theoretical results in Theorem 3.1 and Corollary 3.2. We report KL divergence for comparability with existing results, and include our RNOT results for reference.
> > >
> > > | Manifold | Ours (RNOT) | RCPM $\gamma = 0.01$ | RCPM $\gamma = 0.005$ | RCPM $\gamma = 0.001$ |
> > > |---|---:|---:|---:|---:|
> > > | $\mathbb{S}^{10}$ | **0.04 ± 0.00** | 5.29 ± 0.06 | 5.34 ± 0.07 | 5.40 ± 0.06 |
> > > | $\mathbb{S}^{20}$ | **0.02 ± 0.00** | 8.81 ± 0.03 | 8.81 ± 0.03 | 8.82 ± 0.03 |
> > > | $\mathbb{S}^{30}$ | **0.03 ± 0.00** | 13.46 ± 0.01 | 13.47 ± 0.02 | 13.44 ± 0.02 |
> > > | $\mathbb{T}^{10}$ | **0.92 ± 0.02** | 46.15 ± 0.41 | 45.92 ± 0.40 | 45.59 ± 0.38 |
> > > | $\mathbb{T}^{20}$ | **5.25 ± 0.08** | 83.77 ± 0.39 | 83.73 ± 0.40 | 83.70 ± 0.42 |
> > > | $\mathbb{T}^{30}$ | **3.21 ± 0.10** | 126.66 ± 0.44 | 126.74 ± 0.43 | 126.75 ± 0.44 |
> > >
> > > The results clearly show that RCPM performance degrades dramatically with dimension regardless of $\gamma$, while RNOT remains stable. This confirms that the CoD is a structural property of the discrete parameterization class, not an artifact of the training objective.
> > >
> > > **$SO(3)$ and $SE(3)$ Experiments.**
> > > Following the reviewer's suggestion, we evaluated RNOT on the more complex geometries $SO(3)$ and $SE(3)$. On $SO(3)$, we consider the analogous transport problem from a uniform source to a wrapped normal target, consistent with our $\mathbb{S}^2$ and $\mathbb{T}^2$ experiments. On $SE(3)$, we transport from a product measure (uniform on $SO(3)$ $\times$ uniform on a compact Euclidean set) to a product of a wrapped normal on $SO(3)$ and a truncated normal on the Euclidean component. RCPM suffers numerical instability on $SE(3)$ for $\gamma < 1$ — notably, this setting already departs from the original analysis of Cohen et al. — so only $\gamma = 1$ results are reported for $SE(3)$.
> > >
> > > | Manifold | Method | KL ($\downarrow$) | ESS ratio ($\uparrow$) |
> > > |---|---|---:|---:|
> > > | $SO(3)$ | Ours (RNOT) | 2.96 ± 0.01 | 0.225 |
> > > | $SO(3)$ | RCPM ($\gamma = 1.0$) | 2.86 ± 0.03 | 0.391 |
> > > | $SO(3)$ | RCPM ($\gamma = 0.1$) | 3.29 ± 0.10 | 0.002 |
> > > | $SO(3)$ | RCPM ($\gamma = 0.05$) | 3.54 ± 0.06 | 0.006 |
> > > | $SO(3)$ | RCPM ($\gamma = 0.01$) | 2.83 ± 0.10 | 0.003 |
> > > | $SO(3)$ | RCPM ($\gamma = 0.005$) | 3.72 ± 0.10 | 0.004 |
> > > | $SO(3)$ | RCPM ($\gamma = 0.001$) | 5.90 ± 0.06 | 0.004 |
> > > | $SE(3)$ | **Ours (RNOT)** | **2.50 ± 0.01** | **0.683** |
> > > | $SE(3)$ | RCPM ($\gamma = 1.0$) | 14.38 ± 0.09 | 0.007 |
> > > | $SE(3)$ | RCPM ($\gamma < 1$) | Numerically unstable | — |
> > >
> > > On $SO(3)$, RNOT and RCPM ($\gamma = 1$) achieve comparable KL, though RCPM attains a higher ESS at this particular $\gamma$. However, as $\gamma$ decreases — bringing RCPM closer to a true OT potential — its ESS collapses to near zero, indicating that the smoothed solution at $\gamma = 1$ is not approximating the OT map but rather benefiting from heavy regularization. On $SE(3)$, RNOT substantially outperforms RCPM, and RCPM becomes numerically unstable for all $\gamma < 1$.

---

### Official Review · Reviewer_Ffh1 · 2026-03-11

**Soundness:** 3
**Presentation:** 3
**Significance:** 3
**Originality:** 4
**Overall Recommendation:** 5
**Confidence:** 4

**Summary:**

This paper studies neural optimal transport on compact Riemannian manifolds. The main claim is that discretization based manifold OT parameterizations suffer from a curse of dimensionality, while the proposed continuous class, Riemannian Neural Optimal Transport (RNOT), avoids that barrier by parameterizing a continuous prepotential and enforcing $c$ concavity through the $c$ transform. The paper combines a negative approximation result for discrete output transport classes, a universality theorem for the implicit $c$ concave class, polynomial complexity bounds for approximating sufficiently smooth potentials, and a pointwise stability result for the induced transport map on a full measure subset. Empirically, it compares RNOT against RCPM, RCNF, and Moser flows on spheres and tori, plus a continental drift case study on the sphere.

**Compliance With Llm Reviewing Policy:**

Affirmed.

**Final Justification:**

Authors addressed my concerns and I'm pretty satisfied with their responses, hence, I increased my score to 5.

**Key Questions For Authors:**

1. The KL estimator assumes $T_\theta$ is a $C^1$ diffeomorphism. What guarantees this for the learned RNOT map class in practice? If there is no such guarantee, why is the KL computation valid?

2. Can the authors better separate the claim “competitive generative model” from the claim “better scaling mechanism”? The current low dimensional results suggest the second claim is more strongly supported than the first.

4. Can the authors more directly connect the theory to empirical measurements, for example by testing dependence on smoothness or approximation budget in a way that reflects the complexity theorems?

I am positive on the conceptual contribution and the theoretical ambition. I am not fully convinced by the empirical scale, and I think the KL metric issue is a serious point that needs clarification. If the authors can resolve that issue cleanly and tighten the experimental story, this becomes a much stronger paper.

**Limitations:**

yes

**Strengths And Weaknesses:**

## Strengths

The core idea is novel enough. The paper’s main novelty is the discrete versus continuous distinction for manifold OT parameterizations. Theorem 3.1 and Corollary 3.2 formalize a dimension dependent lower bound for discrete output approximations, and the proposed RNOT class is built precisely to avoid that restriction by using continuous prepotentials and the c transform. That is a clean and original framing.

Also, the theoretical narrative is coherent. The paper is mathematically organized in a sensible way. First it defines the c concave framework and the manifold OT setting. Then it proves a lower bound for discrete output maps. After that it proves universality of the implicit c concave class, then polynomial complexity for function approximation and potential approximation, and finally pointwise stability for the induced maps. The theorem flow is logically consistent.

The dimension sweep does support the claim that RCPM degrades with dimension while RNOT remains much flatter over the tested regime. Even though the regime is limited, the trend is visible and aligned with the intended theory motivation.

## Weaknesses

The main synthetic experiments use 5 runs with 1024 samples each, and the main dimension sweep in the paper is only up to dimension 10, with dimension 40 deferred to the appendix because of computational limits. That is enough for a proof of concept, but it is not enough to convincingly support a broad “breaks the curse of dimensionality” headline.

On $S^2$, RCPM is actually better in KL and ESS and much faster. On $T^2$, RNOT is clearly better. So the empirical message is not that RNOT uniformly dominates previous methods, but rather that it is competitive in low dimensions and appears to scale more gracefully as dimension increases. That is still valuable, but it is narrower than the paper’s rhetoric in a few places.

There is a possible math to evaluation mismatch around KL. The appendix derivation of the KL estimator explicitly assumes that the learned map $T_\theta$ is a $C^1$ diffeomorphism so that a manifold change of variables formula can be applied. I do not see, in the main theoretical development, a corresponding guarantee that the learned RNOT map is globally a $C^1$ diffeomorphism. The main map theorem is a pointwise stability statement for minimizers, not a diffeomorphism result. This creates a real concern that one of the main reported metrics may rely on stronger assumptions than the paper actually proves for the model class. This is the most important technical issue I found. It does not necessarily invalidate the paper, but it should be clarified.

---

> ### Author Rebuttal · Authors · 2026-03-31
>
> We thank the reviewer for the assessment. We address the three key concerns below.
>
> **(1) KL estimator and the diffeomorphism assumption.**
> We agree that the current draft should distinguish more clearly between (i) regularity properties of the *true optimal map* under standard OT assumptions, and (ii) what is currently proved for the *learned finite-capacity RNOT map*. Our present theory does *not* prove that every learned RNOT map is globally a $C^1$ diffeomorphism, so the appendix KL derivation should not be read as a consequence of Theorem 5.3 or Corollary 5.2. We will revise the paper to make this explicit.
>
> What the regularity theory does provide is strong justification for the *underlying OT solution* in our experimental regimes. Under standard MTW regularity assumptions for the quadratic geodesic cost $c=\frac{1}{2}d^2$, smooth strictly positive densities yield smooth Kantorovich potentials, and the optimal map
>
> $$
> T_\star(x)=\exp_x(-\nabla \phi_\star(x))
> $$
> is smooth in the regular regime relevant to our sphere/torus experiments.
>
> For the *learned* map used in KL evaluation, since our theory does not establish a global diffeomorphism guarantee for the finite-capacity model class, we will supplement KL evaluation with empirical diagnostics of local Jacobian nonsingularity. Concretely, for test points $x$ we will compute the differential
>
> $$
> J_x = dT_\theta(x)
> $$
> in orthonormal tangent bases and report the distribution of its smallest singular value $\sigma_{\min}(J_x)$, condition number $\kappa(J_x)$, and $\log|\det J_x|$, together with the fraction of samples satisfying $\sigma_{\min}(J_x)>\varepsilon$. Since $\sigma_{\min}(J_x)>0$ is precisely the local inverse-function-theorem condition, these diagnostics directly corroborate that the learned map behaves as a locally invertible $C^1$ map where KL is evaluated.
>
> We will also clarify that using a manifold change-of-variables formula for KL evaluation is standard in prior work, including RCPMs. The RCPM paper similarly formulates density modeling through a diffeomorphism and the Jacobian-based change-of-variables formula, while noting that its hard discrete OT map is not itself a diffeomorphism and the smoothed version was numerically verified. Our revision will make the same distinction: the KL computation is exact under an evaluation-time regularity assumption, motivated by OT regularity theory and supported empirically through Jacobian nonsingularity checks.
>
> **(2) Separating "competitive generative model" from "better scaling mechanism."**
> We agree that the current draft should separate these two claims more clearly. The evidence most strongly supports RNOT as a better-scaling mechanism for manifold OT, rather than uniform superiority as a generative model. Our main contribution is a continuous parameterization whose approximation complexity behaves more favorably than discretization-based alternatives, as supported by theory and higher-dimensional experiments. The low-dimensional experiments serve a different purpose: they show that adopting this parameterization does not sacrifice generative quality, i.e., RNOT remains competitive when the task is relatively easy. The higher-dimensional regime is where the scaling advantage becomes essential. We will revise to make this distinction explicit and soften any wording implying universal generative superiority.
>
> **(3) Connecting theory to empirical measurements.**
> We agree that a budget- or smoothness-controlled ablation would naturally probe the complexity theorems more directly. Our goal in the current paper is more modest: the theory identifies structural factors governing approximation difficulty, regularity and model capacity, while experiments test the main practical implication, namely scaling with dimension under a fixed parameterization budget.
>
> On regularity, the assumptions in Corollary 5.2 are aligned with known OT regularity regimes: under MTW-type hypotheses, smooth positive densities yield smooth potentials in sphere and product-manifold settings, and away from the cut locus the $c$-transform transfers this regularity to the prepotential. Empirically, our results already reflect the theory's message: under a fixed budget, RNOT remains effective as dimension grows while discretization-based alternatives degrade. We agree a dedicated smoothness/budget ablation would further illuminate the quantitative dependence, but view it as a valuable extension rather than necessary for the main claim. We will revise to make this theory/empirics correspondence explicit.

---

> > ### Author Rebuttal · Reviewer_Ffh1 · 2026-04-02
> >
> > I thank the authors for their clarifications. They have addressed my concerns, and I believe this is a valuable piece of work that fills an important gap. Therefore, I increase my score to 5.

---

> > > ### Author Response · Authors · 2026-04-06
> > >
> > > We thank the reviewer for acknowledging that their concerns have been fully addressed. We appreciate the reviewer’s engagement and increase of the score.

---

### Official Review · Reviewer_pRr1 · 2026-03-12

**Soundness:** 3
**Presentation:** 3
**Significance:** 3
**Originality:** 3
**Overall Recommendation:** 4
**Confidence:** 4

**Summary:**

This paper studies optimal transport on Riemannian manifolds. The authors propose Riemannian Neural Optimal Transport. The paper provides approximation and stability guarantees under regularity assumptions. Also, this paper shows promising performance.

**Compliance With Llm Reviewing Policy:**

Affirmed.

**Final Justification:**

I thank the authors for addressing my questions. I understand the authors' main points regarding the paper, and I maintain my score.

**Key Questions For Authors:**

1. The method relies on geodesic distance. If we do not have explicit information about the data manifold, can this method still be applied? In particular, if we only have a graph representation or point cloud approximation of the manifold, do the authors expect RNOT to still work?
2. Have the authors tested the method in higher-dimensional settings beyond those shown in the main text? How does it compare with discrete OT or the other methods?
3. In Corollary 5.2, can the assumption on the prepotential be guaranteed under natural conditions? It would be helpful if the authors could clarify whether this follows from known regularity results.

**Limitations:**

Yes

**Strengths And Weaknesses:**

This is a solid paper with a meaningful theoretical contribution. The authors provide explicit approximation guarantees for the prepotential class and connect them to the stability of the induced OT map.
My main concern is practical applicability. The method requires access to geometric quantities such as geodesic distance, and in some places also manifold operations like exponential/logarithm maps. This makes it less clear how broadly the method applies when the manifold is not explicitly known and only data samples are available.

---

> ### Author Rebuttal · Authors · 2026-03-31
>
> We thank the reviewer for the  assessment. We address the three questions below.
>
> **(1) Applicability when the manifold is not explicitly known.**
> We agree that access to intrinsic geometric quantities is a substantive modeling assumption. However, this is **not specific to RNOT**; it is the standard setting for most intrinsic generative models on Riemannian manifolds. Riemannian score-based models use geodesic random walks via the exponential map, Riemannian CNFs solve flows using manifold-aware vector fields, and RCPM/IRCPM are built from intrinsic squared-distance costs and exponential-map constructions. RNOT does not assume stronger geometric access than prior work in intrinsic manifold generative modeling.
>
> When the manifold is not given analytically and one only has a graph or point cloud, RNOT can still be applied *after* a geometry-estimation step: one first approximates the intrinsic metric and local geometric operators from data, then runs RNOT on this estimated geometry. Our current theory is formulated where intrinsic geometry is available, so extending guarantees to approximate-geometry settings is an important future direction rather than a claim of the present paper. We will make this assumption explicit in the revised problem setup and clarify that handling point-cloud manifolds requires an additional geometry-learning layer complementary to our contribution.
>
> We view this primarily as a question about the **scope of intrinsic manifold generative modeling as a whole**, rather than a limitation unique to RNOT.
>
> **(2) Higher-dimensional experiments.**
> Yes — beyond the main-text dimension sweep (up to $p=10$), Appendix Figure 5 extends the comparison to $p=40$ on both $S^p$ and $T^p$. The conclusion becomes even clearer: **RNOT remains stable as dimension increases**, with KL staying comparatively low, whereas **RCPM deteriorates sharply**. RCPM results are only shown up to $p=10$ because it became computationally intractable at higher dimensions, while RNOT runs up to $p=40$ and maintains favorable performance.
>
> To directly answer the reviewer's question: **yes, we tested substantially higher-dimensional settings, and the qualitative conclusion remains the same.** Relative to discrete OT via RCPM, our method is markedly more robust in high dimension. In low dimensions RCPM can be competitive, but as dimension grows its performance degrades rapidly and computation becomes impractical, while RNOT performs reliably. In the revision, we will move Figure 5 from the appendix to the main paper so that this comparison is visible more directly.
>
> **(3) Regularity assumption in Corollary 5.2.**
> The $C^{kp,1}$ assumption enters because Theorem 5.1 is a neural approximation result for Holder-smooth scalar functions. This is not an additional modeling assumption specific to RNOT; it is the regularity regime under which polynomial approximation rates are available.
>
> Under standard OT regularity hypotheses of MTW type, smooth positive densities yield smooth Kantorovich potentials. For the quadratic geodesic cost in the round sphere, global smooth solutions are known for smooth positive data, and for products of spheres one has a stay-away from the cut locus together with higher regularity ($C^{2,\alpha}/C^\infty$). Since our torus experiments use a flat torus viewed as a product manifold, these results are directly relevant.
>
> Moreover, away from the cut locus the cost is smooth, so the $c$-transform relation transfers this regularity to the prepotential. Hence, in the regimes covered by known OT regularity theory, the prepotential indeed belongs to $C^{kp,1}$ for any finite $k$. We will add a remark after Corollary 5.2 making this connection explicit and citing the relevant results.

---

> > ### Author Rebuttal · Reviewer_pRr1 · 2026-04-04
> >
> > Thank you to the authors for their response. I have read both the rebuttal and the reviews carefully. While I agree with the authors’ statement that the issue regarding geodesic distances is not specific to RNOT, I believe it still has an important impact on the practical applicability of the method. For this reason, I would like to keep my score.

---

> > > ### Author Response · Authors · 2026-04-07
> > >
> > > We thank the reviewer for the clarification and appreciate the distinction. We agree that requiring access to manifold geometry defines the intended regime of applicability of our method, and we will revise the paper to make this scope more explicit.
> > >
> > > As the reviewer notes, this assumption is shared across geometry-aware generative modeling approaches, and our goal is to provide a theoretically grounded and scalable method within this setting. We hope this clarification helps contextualize the contribution.

---

### Official Review · Reviewer_F3gs · 2026-03-12

**Soundness:** 3
**Presentation:** 3
**Significance:** 4
**Originality:** 3
**Overall Recommendation:** 3
**Confidence:** 3

**Summary:**

In this paper, the authors study the optimal transport problem on a manifold, and it is important that they solve it in the intrinsic geometry of the manifold. They propose to parametrize the Kantorovich potential by a neural network on a compact Riemannian manifold without boundary, and then recover the optimal map through the $c$-transform and the exponential map. On the theory side, the authors prove that constructing an optimal map with constraints on its values at $m$ points leads to the curse of dimensionality, and also that the sequence of empirical solutions of their problem converges pointwise to the optimal map. In the experiments, the method is tested mainly on spheres and tori, as well as on one continental drift example on $S^2$.

**Compliance With Llm Reviewing Policy:**

Affirmed.

**Final Justification:**

I am grateful to the authors for providing a more constructive response than in the previous round.

I would like to note that the addition of more practically motivated examples is a positive signal for me. I am glad that the authors are continuing to develop their paper. At the same time, I must admit that the authors' intention to compare themselves with the results of RCPM, both theoretically and practically, seems to me a negative aspect, since it narrows the positioning of the paper. Instead of presenting the work as a broad and general improvement of the theory of optimal transport on manifolds, it shifts the focus toward explicitly showing that the authors' method outperforms one particular paper.

From the latest response, I now better understand why it is important for the authors not to parameterize the map $T$ with a neural network. Nevertheless, I still view the method as a rather direct adaptation of basic theorems from the theory of optimal transport on manifolds. In practice, this directness is reflected in the noticeably high running time of the method. For this reason, I consider the engineering contribution of the paper to be fairly limited.

The theoretical aspects of the paper remain quite ambiguous. On the one hand, the authors obtain a pointwise convergence estimate toward the optimal transport map and present this as an improvement over RCPM. However, one should not forget that this improvement is achieved under additional smoothness assumptions on the optimal potential. Generally speaking, this smoothness does not follow from the assumptions made on the basic objects of the problem, such as the manifold and the measures. This assumption is therefore implicit at the level of the problem formulation, which makes it a problematic assumption. The authors could have worked out the assumptions on the measures more carefully, so that the smoothness of the potential would follow from the structure of the problem. In addition, stronger and more explicit assumptions on the measures might also help them obtain convergence estimates in $L^2$, which are currently missing.

I thank the authors for their work, but in view of the issues described above, my evaluation of the paper remains weak reject.

**Key Questions For Authors:**

- Can the authors explain in more detail in which real applications optimal transport exactly on compact manifolds without boundary is truly needed?
- Can the authors generalize the theory, at least partially, to manifolds with boundary?
- How do the authors plan to remove the need to solve an inner optimization problem every time $T(x)$ is computed?
- Can the authors strengthen the result for maps to uniform convergence, and not only pointwise convergence?

**Limitations:**

Yes

**Strengths And Weaknesses:**

**Strengths.**

- The theoretical part looks correct overall, and I do not see any clear fundamental error in the main results. Some proofs are written somewhat loosely, but the statements themselves seem correct overall.
- The authors give a clear theoretical motivation for moving from discrete $c$-concave parametrizations to continuous ones. Even if the results do not look surprising, the analysis still seems to be done quite carefully.
- The results on approximation of functions and potentials are stated clearly enough. In this sense, the theory looks stronger than the practical part of the paper.

**Weaknesses.**

- The motivation of the problem remains weak. The authors present the method in a broad way, but the meaningful examples are in fact mostly limited to the sphere and the torus, so the practical need for this exact setting is not convincing.
- The class of spaces considered is too narrow. The whole theory is built only for compact smooth manifolds without boundary, which further weakens the practical value of the work.
- The method does not give an explicit neural network approximation of the map $T$ itself. To compute $T(x)$ at a new point, one still has to solve an inner optimization problem each time.
- The method is computationally expensive. The cost comes both from the inner minimization in the $c$-transform and from computing features through distances to landmarks, so in runtime it is clearly slower than alternative methods.
- The experimental part is limited and does not convincingly show the advantage of the method. On simple tasks, the picture is mixed: the quality is sometimes slightly better, but the runtime is much worse, and the set of examples is too narrow.
- The theory for maps looks weaker than the theory for potentials. The authors only obtain pointwise convergence, which looks weak on a compact manifold.
- In the algorithmic part, there is no clear reference to a dual/semi-dual theorem that justifies the objective used. For a theoretically oriented paper, this looks careless.

At this point, the work is perceived mainly as a theoretical paper. Its most important structural part relies on already known OT theory on manifolds, mainly the description of the optimal map through a $c$-concave potential and the exponential map. The results on the curse of dimensionality for discrete schemes look natural and not especially surprising. The result on convergence of maps looks weak because it gives only pointwise convergence. The motivation and the practical part are clearly weaker than the theory. In its current form, the paper gives the impression of an interesting but not sufficiently convincing work. I recommend a weak reject.

---

> ### Author Rebuttal · Authors · 2026-03-31
>
> We thank the reviewer for the assessment. We address each concern below.
>
> **(1) Motivation and practical need for intrinsic OT on compact manifolds.** Our goal is to develop an intrinsic OT-based generative modeling framework for manifold-valued data broadly. Spheres and tori serve as canonical controlled testbeds with explicit geometry and reliable evaluation, not the limit of applicability. Intrinsic geometry matters for directional/pose data, periodic variables, and molecular modeling (e.g., docking/conformation). We will revise to make this motivation more explicit.
>
> **(2) Why compact manifolds without boundary?** The restriction is deliberate: this is the cleanest setting in which the c-transform, intrinsic exp/log maps, Monge solutions, and regularity theory can be developed without boundary effects. Manifolds with boundary introduce technically distinct regularity issues for both the potential and the transport map. We agree this is a limitation and will make the scope and importance of boundary extensions more explicit.
>
> **(3) The learned object is not yet a fully explicit feed-forward map.** We agree that RNOT does not parameterize the transport map as a single explicit feed-forward network $x \mapsto T_\theta(x)$; instead, it learns the OT potential and recovers the map implicitly through the $c$-transform / argmin structure. This is a deliberate design choice: the goal is to preserve the intrinsic OT structure and enforce $c$-concavity by construction, which underlies the theoretical guarantees. A direct black-box parameterization of $T$ would not provide these properties. We agree that evaluating $T_\theta(x)$ requires an inner solve at test time, and we will make this tradeoff clearer. However, this is best viewed as the cost of an implicit-layer formulation rather than a lack of out-of-sample capability: once the potential has been learned, the same model applies to arbitrary new inputs $x$ without retraining or re-solving the global OT problem. The method generalizes out of sample through an implicit OT-consistent representation rather than an explicit unconstrained map network. More broadly, our aim is to construct a continuous intrinsic OT parameterization with geometric structure and approximation guarantees; the price is the inner optimization needed to recover the map. Acceleration or amortization is an important future direction.
>
> **(4,5) Experimental scope.** We agree that on low-dimensional benchmarks the picture is mixed: RNOT is not uniformly superior and incurs higher wall-clock cost. The main experimental goal is testing whether continuous intrinsic parameterization scales more favorably than discretization-based alternatives **as dimension increases**. The higher-dimensional experiments support our theory: discretization-based baselines degrade beyond $p=10$, whereas RNOT remains stable up to $p=40$. We will better distinguish low-dimensional runtime from high-dimensional scalability in the revision.
>
> **(6) Why is the map result pointwise rather than uniform?** The map is obtained not by a stable linear operation on the potential, but through a minimizer selection,
> $$
> T(x)\in \arg\min_{y\in M}\\{\tfrac{1}{2} d(x,y)^2-\psi(y)\\}.
> $$
> Uniform approximation of the potential does not in general imply uniform approximation of the selected minimizer: singular geometric effects such as minimizer degeneracy and cut-locus phenomena prevent global uniform map control from potential control alone. This is why Theorem 5.3 is formulated on a full-$\mu$-measure subset.
>
> Importantly, our result is not merely qualitative. Theorem 5.3 shows that if $\|\psi_\epsilon-\psi_\star\|_\infty\le \epsilon$, then for every regular point $x\in\Omega$,
>
> $$
> d(T_\epsilon(x),T_\star(x))\le 2\sqrt{\epsilon/C(x)},
> $$
> yielding an explicit $O(\sqrt{\epsilon})$ pointwise rate on a full-measure set. Combined with Corollary 5.2, this gives polynomial $\epsilon^{-1}$-complexity for approximating the transport map pointwise. This mode of convergence is also consistent with prior work: RCPM's map theorem (Theorem 2 therein) states convergence in probability via pointwise convergence for $\mu$-a.e. $x$. Our contribution strengthens this result by providing an explicit stability rate. We will revise the text to emphasize the quantitative nature of this result.
>
> **(7) Semi-dual justification.** We agree that the algorithmic section should explicitly reference the result justifying the objective. The justification is the standard Kantorovich duality together with the $c$-transform reduction to the semi-dual: for $c(x,y)=\tfrac{1}{2} d(x,y)^2$,
>
> $$
> \text{KP}(\mu,\nu) = \sup_{\psi\in C(M)} \left\\{ \int_M \psi^c(x)d\mu(x) + \int_M \psi(y)d\nu(y) \right\\}.
> $$
> Our training objective is precisely this semi-dual, parameterized by $\psi_\theta$. In the revision we will move the derivation from Appendix B.2 into the main text and add an explicit discussion in the algorithmic section linking the optimization to this formulation.

---

> > ### Author Rebuttal · Reviewer_F3gs · 2026-04-04
> >
> > I thank the authors for the comments. Below I give an answer to each point.
> >
> > >Motivation
> >
> > (1) The motivation of the paper still remains unclear, because the authors state that their goal is to use it in generative modeling, and not simply as another OT solver on manifolds. I believe a major weakness is that the method is not tested on real tasks of the kind mentioned by the authors to give the paper more practical relevance (e.g. molecular modeling).
> >
> > >Theory
> >
> > (2-3) It is not very clear what the authors mean by “the goal is to preserve the intrinsic OT structure and enforce c-concavity by construction, which underlies the theoretical guarantees.” What exactly does this mean? Why would a “direct black-box parameterization of $T$” fail to provide some properties? The authors’ reply on why their method does not work for general manifolds, and why they do not parameterize the map directly, sounds very unclear and unconvincing to me. Kantorovich duality holds on any Polish space, an optimal map exists  if the measure $\mu$ gives zero mass to $(D-1)$-dimensional submanifolds [1, Theorem 4.3], and the inclusion $T(x)\in \arg\min_y \lbrace \frac{1}{2} d(x,y)-\psi^*(y)\rbrace$ follows directly from the definition of an optimal dual potential.
> >
> > (6) The authors’ argument here is also not convincing to me. I view this as a weak theoretical result for compact manifolds, and I believe one should expect an estimate of the form $ \int d(T_\theta, T^*)^2 \, d\mu < \delta(\varepsilon)$ under reasonable assumptions on the distributions $\mu$ and $\nu$ [2, Theorem 3.6], [3, Corollary 4.9].
> >
> > >Method
> >
> > (4-5) I see the running time of the authors’ method as its main drawback, and this is quite natural given its structure. One can conclude that the method loses to competing approaches in low dimensions. As for scalability, I would like to see how the method performs not only on toy examples on high-dimensional spheres and tori, but also on real tasks. At the moment, I find the experimental part of this paper very weak, because the parameterization through the dual objective and the need to compute $\min_y \lbrace\frac{1}{2} d(x,y)-\psi^*(y)\rbrace$ for every $x$ does not look like a “deliberate design choice,” but rather like a serious engineering weakness.
> >
> > >Conclusion
> >
> > For these reasons, I recommend that the paper not be accepted in its current form.
> >
> > **References**
> >
> > [1] Albert Fathi and Alessio Figalli, *Optimal Transportation on Non-Compact Manifolds*, *Israel Journal of Mathematics*, 175 (2010), 1–59.
> >
> > [2] Ashok Vardhan Makkuva, Amirhossein Taghvaei, Sewoong Oh, and Jason D. Lee, *Optimal Transport Mapping via Input Convex Neural Networks*, ICML 2020.
> >
> > [3] Roman Tarasov, Petr Mokrov, Milena Gazdieva, Evgeny Burnaev, and Alexander Korotin, *A Statistical Learning Perspective on Semi-Dual Adversarial Neural Optimal Transport Solvers*, ICLR, 2026.

---

> > > ### Author Response · Authors · 2026-04-07
> > >
> > > We thank the reviewer for the continued engagement.
> > >
> > > **(1) Motivation.** This paper primarily focuses on intrinsic manifold OT rather than domain-application. Our evaluation strategy follows the similar prior work as RCPM, which is likewise framed for arbitrary compact Riemannian manifolds but experiments mainly on spheres and tori. We have now added $SO(3)$ and $SE(3)$ experiments (see response to F1EG), directly relevant to molecular conformation [6] and rotational alignment [7]. On $SE(3)$, RNOT achieves KL $2.50$ vs. $14.38$ for RCPM ($\gamma=1$). We observe that RCPM becomes numerically unstable for all $\gamma < 1$.
> > >
> > > **(2--3, 4--5) Why not black-box $T$, and why the inner solve?** These concerns share a common answer, and the reviewer's reference [2] (Makkuva et al.) is the key (in [3] the implementation also uses ICNN from [2]). In the Euclidean setting, $c$-concavity for $c = \frac{1}{2}\|x-y\|^2$ reduces to convexity, so [2] enforces it via ICNNs where the constrained architectures (non-negative weights, convex activations) also have limit in expressiveness and training flexibility. **This is precisely what we do on manifolds**, where $c$-concavity does *not* reduce to convexity and is much harder to enforce.
> > >
> > > OT maps on manifolds must take the form $T(x) = \exp_x(-\nabla \phi(x))$ for a $c$-concave potential $\phi$. A black-box $T_\theta: \mathcal{M} \to \mathcal{M}$ ignores this. RNOT enforces all of this by construction: $\phi_\theta = \psi_\theta^c$ is $c$-concave for *any* $\theta$, which is what makes Theorems 5.1–5.3 go through. The inner $c$-transform solve is the mechanism that achieves this. In this sense, it is the manifold analogue of ICNN's architectural constraints. Removing it would break the structural guarantee. The computational cost is the price of structural guarantees.
> > >
> > > **General manifolds.** Assumptions on manifolds primarily affect the theoretical guarantees rather than the practical method. The distinction between manifold classes is as follows: (i) **Compact without boundary** is the cleanest setting, as well as the most common setting for manifolds, where all our pipelines go through. (ii) **Non-compact without boundary:** As the reviewer notes, several OT results on compact manifolds generalize with additional technical assumptions, for example, the moment conditions in Theorem 1.1 [1]; measure $\mu$ gives zero mass to countably $(n-1)$-Lipschitz sets in Theorem 3.2 [1]. Notice that points on non-compact manifold can also have nonempty cut locus. Extending our results here is entirely possible but would substantially lengthen the paper without conceptual novelty. (iii) **Manifolds with boundary** introduce fundamental difficulties regardless of compactness. A compact manifold with boundary is metrically complete but *not geodesically complete*, so $\exp_x$ is not globally defined, and the Hopf–Rinow equivalence between metric and geodesic completeness breaks down. Moreover, boundary regularity of OT maps is a longstanding open problem even in Euclidean domains: consider a closed set $\Omega \subseteq \mathbb{R}^m$ with boundary $\partial\Omega$, the simplest example of a manifold with boundary, and different assumptions on the boundary geometry lead to qualitatively different regularity regimes for the optimal transport map [4, 5].
> > >
> > > **(6) Pointwise rate.** Our $\mu$-a.e. pointwise rate is stronger than the $L^2$-type estimate the reviewer expects. Concretely: (i) on compact $\mathcal{M}$, $d$ is bounded so, by dominated convergence, Theorem 5.3 yields $\int d^2(T_\varepsilon(x), T_\star(x)) d\mu(x) \to 0$ as a corollary. Assuming $C(x)^{-1}$ is $\mu$-integrable, we also recover the quantitative rate in $\varepsilon$ for $L_2$ convergence. Thus our pointwise convergence *implies* the $L^2$ convergence, not the other way around; (ii) our pointwise rate further implies *uniform convergence with high probability*: for any $\delta > 0$, choosing $\eta$ so that $\mu(\{x : C(x) < \eta\}) < \delta$ gives a uniform $O(\sqrt{\varepsilon})$ rate on a set of measure $\ge 1 - \delta$; (iii) What would be strictly stronger is a global uniform rate on all of $M$, which would require $C(x)$ to be bounded away from zero everywhere. This fails at the cut locus, and to our knowledge such a global uniform result is not available in the literature. We will include a discussion of these points in the revised version.
> > >
> > > (numbered after reviewer's references)
> > >
> > > [4] S. Chen, A. Figalli. Boundary $\varepsilon$-regularity in optimal transportation. *Adv. Math.* 273 (2015), 540–567.
> > >
> > > [5] S. Chen, A. Figalli, E. Indrei. $C^{2,\alpha}$ regularity of free boundaries in optimal transportation. *CPAM* 76 (2023), 2099–2138.
> > >
> > > [6] J. Bose et al. SE(3)-Stochastic Flow Matching for Protein Backbone Generation. ICLR 2024.
> > >
> > > [7] A. Leach et al. Denoising Diffusion Probabilistic Models on SO(3) for Rotational Alignment. ICLR 2022.

---

### Decision · Program_Chairs · 2026-04-30

**Decision:**

Accept (regular)

**Comment:**

The paper introduces a method to avoid the curse of dimensionality in optimal transport  on Riemannian manifolds.
Unlike discrete OT methods (e.g., RCPM), which require exponentially many parameters as dimension grows,
the proposed method uses implicit neural potentials to parameterize OT maps continuously, achieving sub-exponential complexity.

Most reviewers have found that the paper have  good merits and after discussions, they found that despite some issues,
the work makes a significant contribution to the field